# Identification of BiP as a temperature sensor mediating temperature-induced germline sex reversal in *C. elegans*

Jing Shi[1,2,3,4,5], Danli Sheng[1,2,3,4,5], Jie Guo [2,3,4,5], Fangyuan Zhou[2], Shaofeng Wu[2] & Hongyun Tang [1,2,3,4✉]

## Abstract

**Sex determination in animals is not only determined by karyotype but can also be modulated by environmental cues like temperature via unclear transduction mechanisms. Moreover, in contrast to earlier views that sex may exclusively be determined by either karyotype or temperature, recent observations suggest that these factors rather co-regulate sex, posing another mechanistic mystery. Here, we discovered that certain wild-isolated and mutant *C. elegans* strains displayed genotypic germline sex determination (GGSD), but with a temperature-override mechanism. Further, we found that BiP, an ER chaperone, transduces temperature information into a germline sex-governing signal, thereby enabling the coexistence of GGSD and temperature-dependent germline sex determination (TGSD). At the molecular level, increased ER protein-folding requirements upon increased temperatures lead to BiP sequestration, resulting in ERAD-dependent degradation of the oocyte fate-driving factor, TRA-2, thus promoting male germline fate. Remarkably, experimentally manipulating BiP or TRA-2 expression allows to switch between GGSD and TGSD. Physiologically, TGSD allows *C. elegans* hermaphrodites to maintain brood size at warmer temperatures. Moreover, BiP can also influence germline sex determination in a different, non-hermaphroditic nematode species. Collectively, our findings identify thermosensitive BiP as a conserved temperature sensor in TGSD, and provide mechanistic insights into the transition between GGSD and TGSD.**

**Keywords** BiP; Temperature-dependent Sex Determination; TRA-2; Endoplasmic Reticulum-Associated Degradation (ERAD); Temperature Sensor
**Subject Categories** Development; Evolution & Ecology

## Introduction

Sex determination governs the initiation of the sexual identity of an organism and thereby acts as one of the most fundamental developmental processes. It consists of two main components: germline sex determination, which specifies the fate of germ cells into sperm or oocytes, and somatic sex determination, which determines the somatic sexual characteristics of the organism. The mechanisms governing sex determination are amazingly diversified across organisms (Bachtrog et al, 2014; Marin and Baker, 1998). In many animals, sex is primarily determined by the karyotype, such as the ratio of X chromosomes to the autosomes in worms and flies, the presence of the Y chromosome in mammals, and the ZZ/ZW system in birds, which is known as genotypic sex determination (GSD) (Bachtrog et al, 2014; Marin and Baker, 1998). Intriguingly, across a wide range of animal species from invertebrates to vertebrates, environmental cues have also evolved to regulate sex determination (Bull, 1981; Capel, 2017; Tang and Han, 2017; Weber et al, 2020), known as environment-dependent sex determination (ESD). Among ESD mechanisms, temperature-dependent sex determination (TSD) is the most prevalent, especially in reptiles and fish. The ESD mechanisms, such as TSD, were proposed to modulate the male/female ratio in dioecious species or to enable adjustment of the number of sperm and oocytes produced in hermaphrodites in response to the corresponding environmental changes like temperature, thereby potentially serving as a strategy for enhancing organism fitness (Bull, 1981; Crews and Bull, 2009; Warner and Shine, 2008).

The understanding of the mechanisms of TSD has only recently expanded, largely due to the limited genetic manipulations available in species traditionally known to exhibit TSD. Key findings in the red-eared slider turtle, *Trachemys scripta*, indicate that STAT3 phosphorylation and epigenetic regulation are involved in TSD (Ge et al, 2018; Weber et al, 2020). However, the initial perception and translation of temperature cues into sex-reversal signals remain unclear in any species. Previous hypotheses have proposed various proteins, such as heat shock response proteins, enzymes, transcriptional regulators and ion channels, as potential mediators of the temperature to sex-reversal signal transduction (Capel, 2017; Kohno et al, 2010; Weber et al, 2020; Yatsu et al, 2015). However,

[1]Fudan University, 200433 Shanghai, China. [2]Westlake Laboratory of Life Sciences and Biomedicine, 310024 Hangzhou, China. [3]Research Center for Industries of the Future, Key Laboratory of Growth Regulation and Translational Research of Zhejiang Province, School of Life Sciences, Westlake University, Hangzhou, China. [4]Institute of Biology, Westlake Institute for Advanced Study, 310024 Hangzhou, Zhejiang, China. [5]These authors contributed equally: Jing Shi, Danli Sheng, Jie Guo. ✉E-mail: tanghongyun@westlake.edu.cn

no causality has been established between these proteins and TSD yet, leaving the identity of the temperature sensor that mediates TSD still a mystery.

Strikingly, sex determination mechanisms exhibit remarkable plasticity and transitions between GSD and ESD can occur in some animals (Ezaz et al, 2006; Holleley et al, 2015). For example, certain lizard species, such as Alpine skinks or *Pogona vitticeps*, have shown a higher likelihood of transitioning from GSD to TSD with increased temperature (Dissanayake et al, 2021; Holleley et al, 2015). Comparative chromosome mapping analyses in temperature-sensitive animals further support frequent evolutionary transitions between GSD and TSD (Dissanayake et al, 2021). However, the molecular mechanisms enabling these switches between different sex determination modes remain enigmatic. Intriguingly, several studies have revealed the simultaneous operation of GSD and TSD within the same individual in various species (Chen et al, 2014; Ezaz et al, 2006; Holleley et al, 2015; Luckenbach et al, 2009; Yamamoto et al, 2014), which is in striking contrast to the traditional view that the sex determination pattern is exclusively dominated by either TSD or GSD. This phenomenon, where sex is determined by karyotype but with a temperature override within the same organism, potentially represents an intermediate state during the transition between GSD and TSD, which may provide a unique circumstance for studying the mechanisms underlying the switch between different modes of sex determination and the transduction of temperature into a sex cue. However, conducting mechanistic studies in animals such as fish, in which these phenomena have been observed, presents a significant challenge. As a result, the mechanism that permits the switch of GSD and TSD remains a mystery.

Moreover, the exploration of the capability of hermaphrodites to exhibit temperature-dependent somatic and/or germline sex determination has been limited. Germline sex determination, modulating germ cell specification into sperm or oocytes, is a critical aspect of sex determination. In hermaphrodites, the ability to adjust germline sex determination in response to temperature change could result in alterations in the quantity of sperm and oocytes and thereby may impact brood size, potentially serving as a strategy for hermaphrodites to adapt to temperature fluctuations. Exploring this phenomenon could not only complete our understanding of the mechanism underlying TSD, but also shed light on how species with distinct reproductive modes adapt to environmental temperature fluctuations. Therefore, this study uses *C. elegans* hermaphrodites to elucidate the molecular basis underlying the transduction of environmental temperature into an intracellular signal for germline sex determination, as well as the regulation of the potential transition between Genotypic Germline Sex Determination (GGSD) and Temperature-dependent Germline Sex Determination (TGSD).

Germline sex determination in *C. elegans* was primarily regulated by X chromosomes to autosomes ratio (Ellis and Schedl, 2007; Madl and Herman, 1979), but with a potential override by environmental factors (Tang and Han, 2017), which suggests that *C. elegans* with the proper genetic background may show TGSD. Thus, we investigated this idea by exploring whether any wild-isolated or laboratory-used strain may gain the ability to modulate germline sex determination in response to temperature alterations. Remarkably, our investigations have led to the discovery of two wild-isolated and several loss-of-function mutant *C. elegans* strains,

in which germline sex determination in hermaphrodites can be co-regulated by both temperature and karyotype, and thereby exhibits the coexistence of TGSD and GGSD. Further analyses have identified BiP as a temperature sensor mediating TGSD. Specifically, available BiP is reduced in responding to warmer temperature-induced ER protein-folding burden and, in turn, triggers the reduction of TRA-2, a critical driver of female germline sex, via the process of ER-associated protein degradation (ERAD). This modulation partially overrides the activity of the karyotype-controlled germline sex determination pathway, thus enabling the coexistence of GGSD and TGSD within the same worm. Strikingly, we have been able to experimentally induce the switch between GGSD and TGSD by properly manipulating the basal activity of the germline sex determination pathway. Interestingly, BiP is also required for driving oocyte fate in a dioecious nematode species, suggesting a possible conserved role of BiP in germline sex determination. In summary, our findings identify the thermo-sensitive BiP as a temperature sensor that transduces ambient temperature into a signal-modifying sperm/oocyte fate and provide mechanistic insights into the intricate interplay between environmental cues and the karyotype that enable the switch of GGSD and TGSD within the same organism.

## Results

### The discovery of TGSD in *C. elegans*

The germline sex determination by karyotype in *C. elegans* has been extensively examined in previous studies (Goodwin and Ellis, 2002), showing that when *C. elegans* hermaphrodites are cultured at 20 °C, ~160 sperm are made in each germline arm before switching from sperm production to oogenesis for the rest of the lifespan. It has been indicated that the change in the germline sex determination in *C. elegans* hermaphrodites could result in alterations in sperm production, such as excess sperm production (>160 sperm per arm) indicating the masculinization of the germline (Mog) or the reduction in germ cell specification into sperm indicating the feminization of the germline (Fog) (Ellis, 2008; Tang and Han, 2017). Considering that unlike dioecious species, *C. elegans* hermaphrodites possess the ability to regulate the brood size by adjusting the number of sperm produced through altering the germline sex and that *C. elegans* germline showed sensitivity to environmental changes, we explored whether *C. elegans* could potentially have the ability to change germline sex in response to temperature. The high temperature was observed to induce masculinization in other animals (Honeycutt et al, 2019; Ribas et al, 2017), thus we set out to investigate wild-isolated *C. elegans* strains for their potential to display TGSD by analyzing whether a warmer temperature treatment might masculinize the germline sex of these *C. elegans* hermaphrodites. The late L3 stage is when the hermaphrodite germline switches from spermatogenesis to oogenesis (Ellis and Schedl, 2007), thus we first tested whether shifting from a standard lab culture temperature (20 °C) to a commonly used warmer temperature (30 °C) during the late L3 stage (Fig. 1A) could lead to germline sex change by counting the number of sperm via the widely-used DAPI staining assay (Appendix Fig. S1A; Tang and Han, 2017). By randomly picking a total of eleven wild-isolated strains available from the

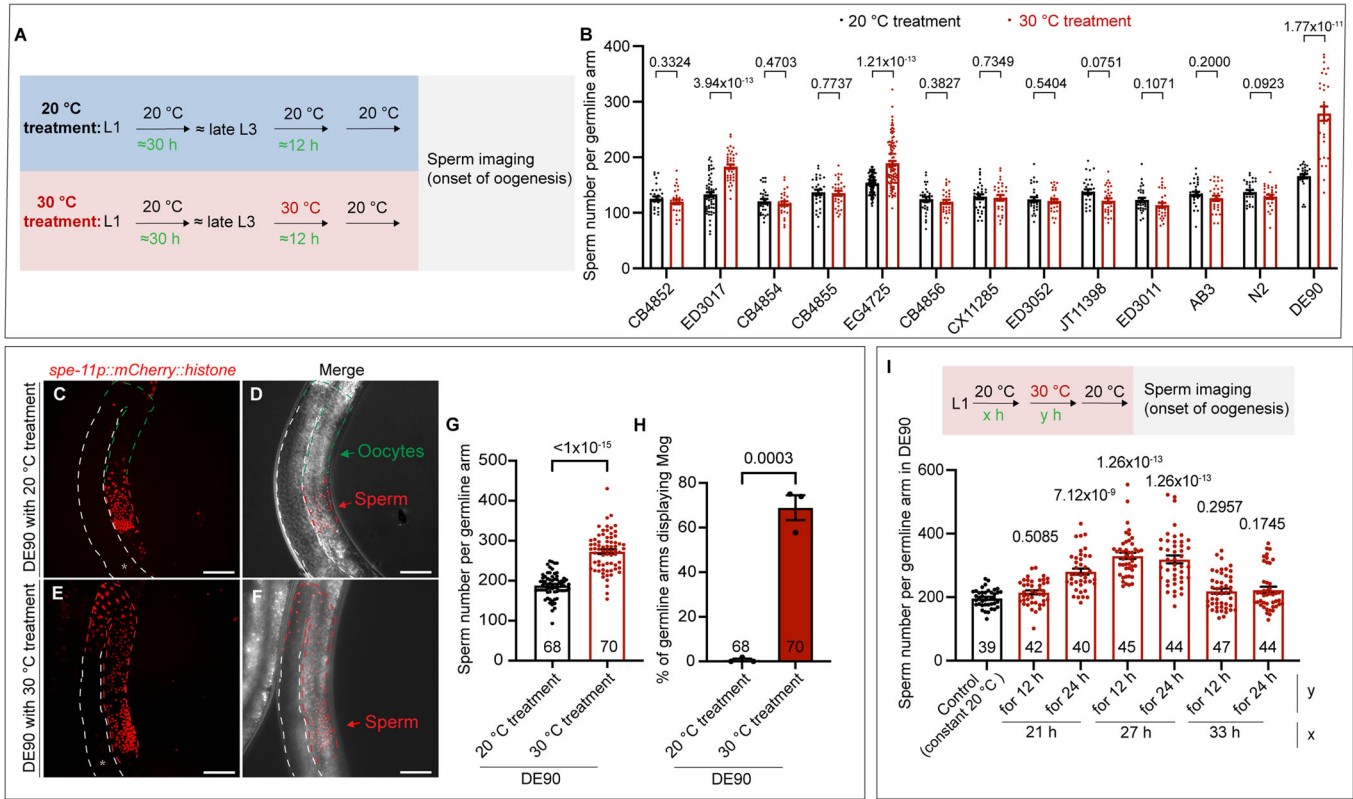

**Figure 1. Warmer temperature drives male germline fate in *C. elegans*.**

(**A**) Schematic diagram of the experimental workflow for analyzing the effect of temperature on germline sex determination in *C. elegans*. *C. elegans* hermaphrodites were cultured at the indicated temperature, and imaging of the germline was performed to count sperm at the first appearance of oocytes. In this study, all 20 °C treatment and 30 °C treatment were performed as shown in this diagram unless otherwise indicated. (**B**) Bar graph showing that the ED3017, EG4725 and DE90 *C. elegans* strains display partial Mog and produce excess sperm in response to warmer temperatures. The laboratory wild-type N2 strain, the N2-derived DE90 [*oxIs318 (spe-11p::mCherry::histone);ruIs32 (pie-1p::GFP::histone H2B);ddIs6 (tbg-1::GFP); dnIs17 (pie-1p::GFP::hPLCIII-delta PH domain)*] strain carrying gamete markers, and 11 other wild-isolated strains, including ED3017 and EG4725, were analyzed for their sensitivity to change germline sex in response to the 30 °C treatment. The sperm number in each germline arm of the listed strains treated with the indicated temperatures was counted by analyzing 3D germline images. For the 20 °C and 30 °C treatments, 61 and 42 germline arms were analyzed for ED3017, and 100 and 100 germline arms were analyzed for EG4725, respectively. For the other strains, 30 germline arms were analyzed for each treatment. *P* value = 0.3324, *P* value = 3.94 × 10$^{-13}$, *P* value = 0.4703, *P* value = 0.7737, *P* value = 1.21 × 10$^{-13}$, *P* value = 0.3827, *P* value = 0.7349, *P* value = 0.5404, *P* value = 0.0751, *P* value = 0.1071, *P* value = 0.2000, *P* value = 0.0923, *P* value = 1.77 × 10$^{-11}$ (from left to right). (**C–H**) Representative images and corresponding quantitative data indicating that warmer temperature partially masculinizes the germline of DE90 *C. elegans* hermaphrodites. The DE90 worms carried the transgene *spe-11p::mCherry::histone*, which specifically labels nuclei in spermatids and spermatocytes. This characteristic was utilized to determine the quantity of sperm in the DE90 strain. (**C–F**) The germline arm is outlined with dashed lines, and the green arrow indicates the oocytes, while the red arrow indicates the sperm. Quantitative analyses of the sperm number and the percentage of germline arms showing Mog in DE90 hermaphrodites with the indicated temperature treatments are shown in (**G, H**), respectively. A total of 68.9% of DE90 hermaphrodites with 30 °C treatment displayed Mog (**H**). *P* value < 1 × 10$^{-15}$ (**G**), *P* value = 0.0003 (**H**). Scale bar: 30 μm. (**I**) Bar graph indicating the developmental stage at which the *C. elegans* DE90 strain responds to the warmer temperature (30 °C treatment) to produce excess sperm. "x" and "y" indicate the time for which the worms were cultured post-hatching at 20 °C and the time for which the animals were incubated at 30 °C, respectively. Temperature treatment was performed as the schematic diagram shown. *P* value = 0.5085, *P* value = 7.12 × 10$^{-9}$, *P* value = 1.26 × 10$^{-13}$, *P* value = 1.26 × 10$^{-13}$, *P* value = 0.2957, *P* value = 0.1745 (from left to right). For all panels, the asterisk (*) in the microscopic images indicates the distal end of the germline arm. Each dot represents the number of sperm in a germline arm in (**B, G, I**) and the percentage of germline showing Mog in each replicate in (**H**). Statistics were performed by unpaired *t* test (**B, G, H**) or one-way ANOVA (**I**). The data in the graphs are the means ± SEMs. All experiments were conducted with at least three biologically independent replicates. Source data are available online for this figure.

Caenorhabditis Genetics Center (CGC) for testing TGSD, we found that in the ED3017 and EG4725 strains, part of the germ cells that would have been oocyte-fated at 20 °C were masculinized and became sperm under the warmer temperature (30 °C), as indicated by the excess sperm production (Fig. 1A,B). Our findings revealed that germline arms displaying Mog were 52.5% for ED3017 and 65.3% for EG4725 with 30 °C treatment, respectively (Appendix Fig. S1B,C). Therefore, the ED3017 and EG4725 wild-type *C. elegans* are responsive to the temperature

increase to override the karyotype to partially masculinize the germline (Fig. 1B; Appendix Fig. S1B,C), displaying the TGSD.

We employed the same testing procedure on two other *C. elegans* strains: a laboratory-used wild-type N2 strain, whose germline sex determination has been previously demonstrated to be dictated by karyotype (Ellis and Schedl, 2007); and a N2-derived strain (DE90) carrying transgenes, including *Is[spe-11p::mCherry::histone]*, which is commonly utilized for specifically labeling sperm. These strains were supposed to serve as negative controls for testing

the effect of temperature on germline sex in the wild-isolated *C. elegans* strains. Consistent with previous findings (Ellis and Schedl, 2007), we found that the number of sperm produced by the domesticated N2 wild-type strain with the warmer temperature treatment was similar to that produced by the ones without a shift to the warmer temperature (Fig. 1B), confirming that N2 strain does not exhibit TGSD under this condition. Surprisingly, we found that the transgenic reporter strain DE90 produced a significantly higher number of sperm in response to warmer temperature (Fig. 1B), and ~70% of the scored germline arms exhibited Mog (Fig. 1C–H). In contrast to the N2 strain, where sex determination is solely influenced by the karyotype, the notable response of the N2-derived DE90 strains to warmer temperature to drive male germline sex fate signifies a transition from GGSD to TGSD in *C. elegans*. Only part of the germ cells in the ED3017, EG4725, and DE90 strains were masculinized in response to temperature elevations (Fig. 1B); therefore, the ED3017, EG4725, and DE90 strains displayed genotypic sex determination but with a temperature override, where both genotypic germline sex determination and temperature-dependent germline sex determination mechanisms coexist.

## The L3-L4 stage is the thermosensitive period (TSP) during which temperature elevation can masculinize the *C. elegans* germline sex

Given the established sperm marker in the DE90 strain and the most obvious warmer temperature-induced Mog phenotype that it exhibited among these three strains displaying TGSD, the DE90 *C. elegans* strain was employed to further investigate the mechanism by which temperature is sensed to modulate germline sex. Then, we further analyzed the developmental stages, at which the DE90 hermaphrodites are most responsive to warmer temperature to masculinize the germline sex, and the suitable duration of warmer temperature exposure to induce Mog phenotype. Our results indicated that a temperature shift to 30 °C for 12 h or 24 h after the worms had been cultured at 20 °C for 27 h post-hatching induced the most obvious increase in sperm production (Fig. 1I). This result corroborates that the L3-L4 stage is the thermosensitive period (TSP) during which *C. elegans* responds to warmer temperature to alter germline sex determination.

Furthermore, we tested a range of temperatures to determine the temperatures that exhibit masculinizing effects. We found that 30 °C treatment exerted a strong masculinizing effect as indicated by the obvious increase in the production of the sperm, and in contrast, treatments with temperatures of 25 °C, 28 °C and 32 °C did not cause noticeable increase in sperm production (Appendix Fig. S2). In subsequent experiments, unless otherwise specified, hermaphrodites were initially cultured at 20 °C for 27–30 h, followed by a shift to 30 °C for 12–24 h, and then returned to 20 °C before analyzing the impact of warmer temperature on germline sex. This procedure is referred to as the "warmer temperature (30 °C) treatment". In contrast, "20 °C treatment" indicates that the worms were cultured at a constant 20 °C before sperm counting. Together, these findings further characterize the ability of DE90 strains to modulate germline sex in response to warmer temperatures, facilitating the use of this *C. elegans* strain for further studying the mechanism underlying TGSD.

## Warmer temperature overrides karyotype to drive sperm fate through modulating the germline sex determination pathway in *C. elegans*

The germline sex determination pathway regulates sperm/oocyte fate in *C. elegans* (Fig. EV1A) and the transcripts of *fog-3*, one of the farthest downstream regulators of germline sex in this pathway, was thus previously used to indicate changes in the germline sex determination pathway (Ellis and Schedl, 2007; Tang and Han, 2017). We next sought to investigate whether temperature may act through this pathway to modulate germline sex. Thus, we first evaluated the effect of temperature on the mRNA level of *fog-3*, that drives sperm fate (Ellis and Schedl, 2007; Tang and Han, 2017). Our qPCR analysis indicated that compared to DE90 worms with 20 °C treatment, DE90 worms with 30 °C treatment displayed an obvious increase in *fog-3* transcription (Fig. 2A), consistent with the excess sperm production observed in these worms (Fig. 1C–H). Moreover, the *fog-3(RNAi)* almost completely suppressed warmer temperature-induced Mog in DE90 worms (Fig. 2B), confirming the role of increased *fog-3* transcription in mediating Mog resulted from warmer temperature treatment. In addition, we compared *fog-3* transcription between N2 that did not acquire TGSD and DE90 strains displaying TGSD after warmer temperature treatment and consistently, we found that upon 30 °C treatment, the *fog-3* level in DE90 is much higher than that in the N2 (Fig. 2A), further supporting the notion that *fog-3* transcription increase in DE90 is responsible for the warmer temperature-induced Mog. Although not as high as that in DE90, N2 also showed an increase in *fog-3* transcription after warmer temperature treatment (Fig. 2A); however, it did not result in Mog (Fig. 1B), likely due to not reaching the threshold for driving sperm fate. Together, these results demonstrate that temperature influences germ cell fate by affecting the activity of the germline sex determination pathway in *C. elegans*.

Next, we performed epistasis analyses to determine the position where the warmer temperature acts on the germline sex determination pathway. Since we did not know the exact genetic change in the background of the DE90 strain that enables TGSD, throughout this study we performed RNAi or directly introduced mutations by using the CRISPR–Cas9 method into the DE90 strain instead of by crossing to analyze the functional relationship between temperature and the corresponding genes. Our further analyses indicated that the masculinizing effect of warmer temperatures is dependent on *fem-3*, as evidenced by that knocking down *fem-3*, acting upstream of *fog-3*, totally abolished the masculinizing effect of the warmer temperature on the germline in the DE90 (Fig. 2B). Taken together, these results support the idea that temperature acts on the genes upstream of *fem-3* to alter germline sex.

Interestingly, we found that *fog-3* transcription was also induced by the warmer temperature in N2 (Fig. 2A), which supports the notion that similar to DE90, the N2 wild-type *C. elegans* strain might also have already acquired the ability to perceive temperature fluctuations to influence the activity of the sex determination pathway, but not strong enough to manifest as a change in germline sex phenotype. Moreover, we found that when cultured at a constant 20 °C, the basal level of *fog-3* expression in N2 was much lower compared to DE90 (Fig. 2A), which may explain the much higher level of *fog-3* in DE90 compared to N2 upon warmer

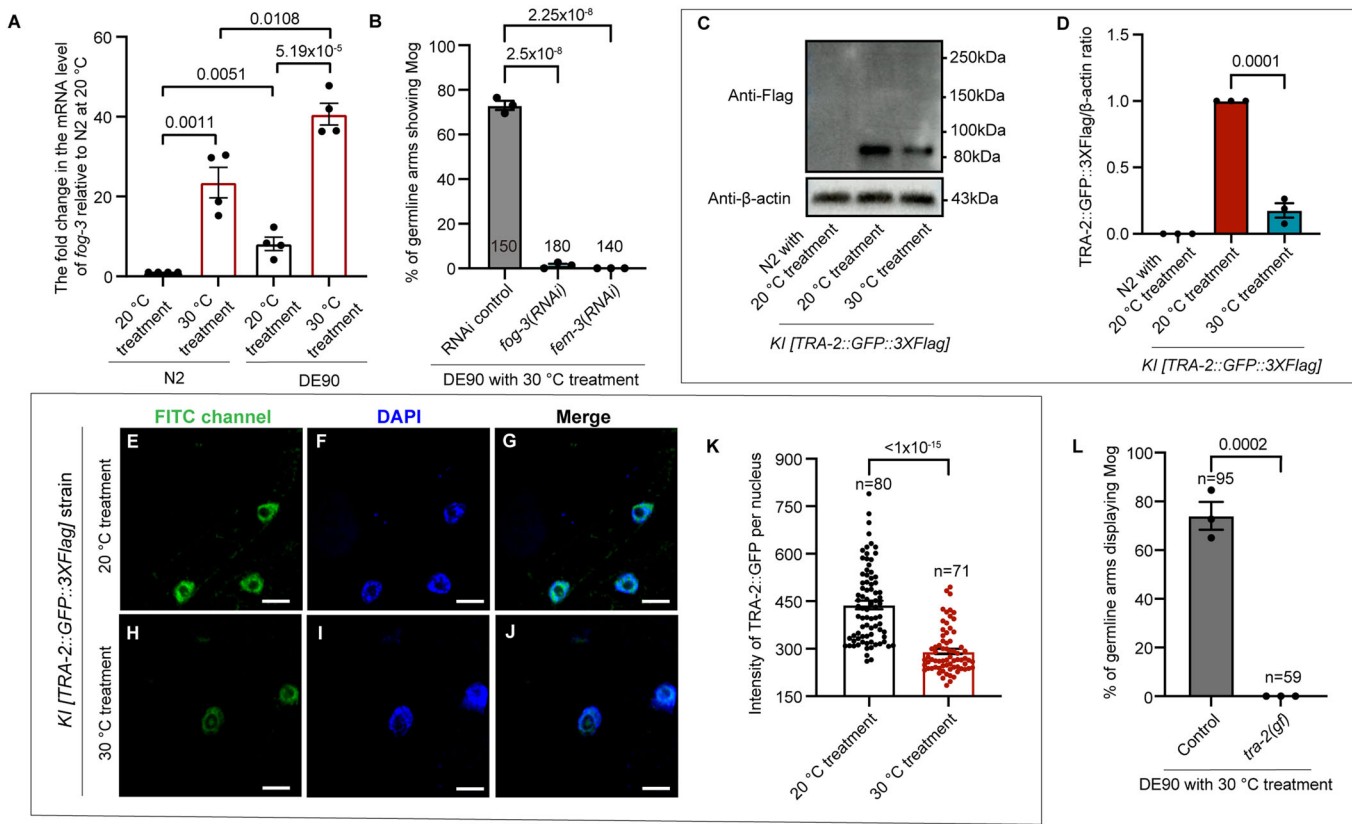

**Figure 2.  Temperature acts upstream of TRA-2, a key sex determination regulator, to modulate sperm/oocyte fate.**

(A) qPCR analysis indicating that the level of *fog-3* transcript in DE90 is higher than N2, both with the 20 °C treatment and the 30 °C treatment. An induction of *fog-3* mRNA level by warmer temperature (30 °C) was also observed in both N2 and DE90. The day-1 adult N2 and DE90 worms receiving the indicated temperature treatment were collected for qPCR analyses. Each dot represents the fold change in *fog-3* mRNA in each replicate. *P* value = 0.0011, *P* value = 0.0051, *P* value = 0.0108, *P* value = $5.19 \times 10^{-5}$ (from left to right). (B) Bar graph showing the suppression of warmer temperature-induced Mog by the knockdown of *fog-3* and *fem-3*. The percentage of germline arms displaying Mog in the DE90 worms with 30 °C treatment, receiving the indicated feeding RNAi treatment, was evaluated. In the RNAi control animals, a total of 73.0% of germline arms were masculinized with 30 °C treatment, while the percentage of germline arms displaying Mog in worms with *fog-3*(RNAi) and *fem-3*(RNAi) treatment is nearly 0.0%. Each dot represents the percentage of germline arms showing Mog in each replicate. *P* value = $2.5 \times 10^{-8}$, *P* value = $2.25 \times 10^{-8}$ (from left to right). (C, D) Representative western blot images and corresponding quantitative analysis showing a reduction in TRA-2 level after the warmer temperature treatment. The *KI [TRA-2::GFP::3×Flag]* strain, with GFP::3×Flag knocked in to the C-terminus of the native TRA-2, was analyzed with western blot using an anti-Flag antibody after the indicated treatment. The molecular weight of the TRA-2(ic)::GFP::3×Flag band is 83 kDa. The size of the full-length TRA-2 fused with GFP::3×Flag is approximately 200 kDa, but it could not be detected. The TRA-2::GFP::3×Flag/β-actin ratio in the 30 °C treatment group was reduced to 17.6% of that in worms with the 20 °C treatment. N2 worms cultured at a constant 20 °C served as a control to determine the specificity of the antibody. Each dot represents the value of TRA-2::GFP::3×Flag/β-actin ratio in each replicate. *P* value = 0.0001 (D). (E–K) Micrographs and a bar graph showing that warmer temperature treatment triggers a reduction in TRA-2 expression in intestinal cells. The GFP fluorescence intensity of the native-expressed TRA-2::GFP::3×Flag in the nuclei of the intestinal cells was measured to determine the TRA-2 level. DAPI was used to outline the nuclei. Each dot in the bar graph represents the mean intensity of TRA-2::GFP in one nucleus. *P* value $< 1 \times 10^{-15}$ (K). Scale bar: 10 μm. (L) Bar graph showing that the induction of Mog by the 30 °C treatment was suppressed by *tra-2(gf)*. The *tra-2(gf)* mutation, mimicking *tra-2(e2020, gf)* that causes constitutively high expression of the TRA-2 protein (Doniach, 1986), was created in the DE90 strain background by using the CRISPR–Cas9 method to delete part of the 3′UTR of *tra-2*. Each dot represents the percentage of masculinized germline arms in each replicate. *P* value = 0.0002. For all the quantitative analyses, the data are shown as the means ± SEMs. Statistics were analyzed by one-way ANOVA (B) and unpaired *t* test (A, D, K, L). At least three biological replicates were carried out. Source data are available online for this figure.

temperature treatment (Fig. 2A) and the germline sex phenotype differences between these two strains with warmer temperature treatment. Additionally, whole-genome sequencing analyses of DE90 revealed no mutations presented in the known sex determination genes in this strain (Table 1), suggesting that certain other genetic modification(s) introduced during constructing this transgenic strain may sensitize the *C. elegans* to show TGSD. Collectively, these data suggest that both DE90 and N2 strains can respond to temperature changes to modulate sex determination pathway activity, but the much higher basal level of *fog-3*

expression in DE90 makes it a more sensitized background for analyzing the impact of temperature on germline sex reversal.

## Warmer temperature reverses karyotype-controlled female germline sex through reducing the level of an oocyte-driving factor TRA-2

Next, we sought to determine whether warmer temperature may promote male germline sex by decreasing the level of TRA-2, a critical factor functioning upstream of FEM-3 to drive oocyte fate

**Table 1. No mutations were detected in sex determination genes in the DE90 strain.**

| Gene | Analysis of SNP and indel | | | |
| | Upstream (5 kb) | Exonic | Intronic | Downstream (5 kb) |
|---|---|---|---|---|
| her-1 | / | / | / | / |
| gld-1 | / | / | / | / |
| gld-3 | / | / | / | / |
| tra-2 | / | / | / | / |
| tra-3 | / | / | / | / |
| fem-1 | / | / | / | / |
| fem-2 | / | / | / | / |
| fem-3 | / | / | / | / |
| nos-1 | / | / | / | / |
| fbf-1 | / | / | / | / |
| fbf-2 | / | / | / | / |
| mog-1 | / | / | / | / |
| mog-2 | / | / | / | / |
| mog-3 | / | / | / | / |
| mog-4 | / | / | / | / |
| mog-5 | / | / | / | / |
| mog-6 | / | / | / | / |
| tra-1 | / | / | / | / |
| fog-1 | / | / | / | / |
| fog-2 | / | / | / | / |
| fog-3 | / | / | / | / |

"/" represents that there is no difference in the sequence of the indicated region of these sex determination-related genes between DE90 and the reference sequence (WS288 release). The mutation analyses of DE90 worms were conducted through whole-genome sequencing.

(Fig. EV1A) (Ellis and Schedl, 2007). Full-length TRA-2 has been previously suggested to be a transmembrane receptor localized on the plasma membrane (Kuwabara et al, 1992; Sokol and Kuwabara, 2000), where a cytosolic fragment of TRA-2 [TRA-2(ic)] is formed by cleavage and subsequently, translocate into the nucleus to drive female fate (Kuwabara and Kimble, 1995; Mapes et al, 2010b; Shimada et al, 2006; Sokol and Kuwabara, 2000) (Fig. EV1B). Given that TRA-2 fused to GFP was previously shown to be functional (Mapes et al, 2010b), for studying the impact of temperature on the TRA-2 level, we established the KI [TRA-2::GFP::3×Flag] strain by using the CRISPR–Cas9 strategy to knock in GFP::3×Flag to the C-terminus of the native TRA-2 protein. Aligning with previous findings, we found that the native TRA-2::GFP signal was observed in the nuclei (Fig. EV1C–F), instead of plasma membrane or perinuclear region where full-length TRA-2 is suggested to localize (Kuwabara et al, 1992; Sokol and Kuwabara, 2000). This result suggests that the full-length transmembrane TRA-2 is rapidly cleaved in worms to generate the fragment of TRA-2 [TRA-2(ic)] localized mainly in the nuclei, as previously implicated (Hubert and Anderson, 2009; Mapes et al, 2010b; Shimada et al, 2006). In further support of this point, our western blot analysis of the KI [TRA-2::GFP::3×Flag] worms indicated that the size of the obtained band corresponded to TRA-2 [TRA-2(ic)], and no full-length

TRA-2 was detected (Fig. 2C). Moreover, when overexpressing TRA-2::mCherry in the Sf9 insect cells, the localization of TRA-2 at the perinuclear region and the plasma membrane was clearly detected (Fig. EV1G–I), which supports that TRA-2 is indeed synthesized in the perinuclear ER and then translocated onto the plasma membrane (Spiess et al, 2019). Together, the transmembrane TRA-2 is synthesized in the ER and trafficked onto the plasma membrane, where it undergoes rapid cleavage and then proceeds to translocate into the nucleus to modulate sex determination.

Before investigating the influence of temperature on TRA-2, we first need to understand in which tissue TRA-2 acts to modulate germline sex. It has been previously shown that genetic analyses indicated tra-2 functioning in the germline to regulate germline sex (Clifford et al, 2000; Jan et al, 1999), but the expression of tra-2 could not been detected in the germ cells with assays such as antibody staining (Mapes et al, 2010b; Shimada et al, 2006). We thus probed the expression of endogenous TRA-2 by microscopic analyses of the KI [TRA-2::GFP::3×Flag] worm. In good agreement with previous reports (Mapes et al, 2010b; Shimada et al, 2006), we found that endogenous TRA-2::GFP was readily observable in somatic cells but undetectable in germ cells (Fig. EV1C–F). This observation implied that tra-2 might be expressed at an undetectable level in the germline, which prompted us to further investigate whether tra-2 actually acts in the germline to regulate germ cell fate as previously reported. Thus, we performed tissue-specific genetic analyses of the role of tra-2, and the obtained data showed that germline-specific or global knockdown of tra-2 but not intestine-specific knockdown of tra-2 caused Mog (Appendix Fig. S3A). These genetic analyses indicated that tra-2 acts in the germline to modulate sperm/oocyte fate and thus support that tra-2 is indeed expressed in germ cells, but at an extremely low level that is beyond current detection limits. Furthermore, performing germline-specific RNAi against gfp in the KI [TRA-2::GFP::3×Flag] strain resulted in a dramatic Mog phenotype (Appendix Fig. S3B), which indicated that the native TRA-2::GFP is expressed and functions in the germline in this knock-in strain, though the GFP signal is undetectable (Fig. EV1D). Therefore, these findings provide compelling evidence that TRA-2 is indeed expressed and acts within the germline to modulate germ cell fate, despite its expression levels being undetectable by conventional means. Collectively, the tra-2 gene exhibits an incredibly low level of expression within the germline but is sufficient to promote female germline sex.

We next investigated the possible role of a decrease in TRA-2 level in mediating the impact of warmer temperature treatment on driving male germline fate. Given that the level of TRA-2 in the germline is undetectable and the possibility that temperature could potentially affect TRA-2 level throughout the entire worm body, we first analyzed the impact of high temperature on TRA-2 expression by detecting the global level of native TRA-2::GFP in the whole worm with western blot analyses, which was also used as a strategy to tackle this issue in previous studies (Mapes et al, 2010a). Western blot analysis of the KI [TRA-2::GFP::3×Flag] worms indicated that warmer temperature treatment indeed reduced tra-2 expression compared to that in worms with 20 °C treatment (Fig. 2C,D), and this finding was further supported by the microscopic analyses showing a decrease in the intensity of the TRA-2::GFP signal in the nuclei of intestinal cells after warmer temperature treatment (Fig. 2E–K). Moreover, warmer temperature exposure phenocopied

the sex determination phenotype caused by the reduction in *tra-2*. Specifically, global knockdown of the TRA-2 expression induced excess sperm production (Appendix Fig. S3A) but no somatic masculinization in the hermaphrodites, as evidenced by the lack of a male-type tail (Appendix Fig. S3C–F), which was also observed in the warmer temperature-treated DE90 worms that displayed Mog but no masculinization of the soma (Appendix Fig. S3G–J). The masculinization in the germline but not in the soma by a partial reduction in the global TRA-2 level is further supported by previous findings that *tra-2(rf, e1875)* and *tra-3(rf, e2333)* induced excess sperm production but no somatic masculinization in the hermaphrodites (Hodgkin and Barnes, 1991). The observation that global depletion of TRA-2 by *tra-2(null)* results in masculinization of both the soma and the germline (Hodgkin and Brenner, 1977), while the global reduction in TRA-2 levels caused by *tra-2(RNAi)*, *tra-2(rf, e1875)* (Hodgkin and Barnes, 1991), *tra-3(rf, e2333)* (Hodgkin and Barnes, 1991), or warmer temperature exposure primarily leads to masculinization of the germline rather than the soma, may be attributed to the significantly lower expression of *tra-2* in the germline than in the soma (Appendix Fig. S3), which renders the germline more sensitive to reductions in TRA-2 levels, thereby allowing the threshold for inducing male germline fate to be more readily surpassed. These results support the notion that warmer temperature promotes male germline fate by reducing TRA-2 levels.

To further demonstrate the causal role of this decrease in TRA-2 level in mediating warmer temperature-induced excess sperm production, we generated a *tra-2(gf)* mutation by deleting part of the 3'UTR of *tra-2*, mimicking *tra-2(e2020, gf)* (Doniach, 1986), in the DE90 strain using the CRISPR–Cas9 method; this dominant-active mutation is known to increase *tra-2* expression specifically in the germline, causing the transformation from hermaphrodite into a fertile female (Clifford et al, 2000; Hubert and Anderson, 2009; Jan et al, 1999). This *tra-2(gf)* mutation completely suppressed the Mog caused by warmer temperature treatment in DE90 (Fig. 2L), indicating that increasing TRA-2 expression in the germline can suppress the masculinizing effect of a warmer temperature on germline sex. Our findings, including the warmer temperature-caused reduction in TRA-2 in the worm (Fig. 2C–K) and the suppression of warmer temperature-induced Mog by the germline-specific increase of TRA-2 level (Fig. 2L), demonstrate that warmer temperature treatment decreases TRA-2 expression to promote male germline sex.

## ER chaperone BiPs is required for driving the female germline sex in *C. elegans*

After determining the impact of temperature on TRA-2 expression, we next aimed to illustrate how temperature changes are sensed to modulate TRA-2 in the germline sex determination pathway. According to the proposal that a protein functioning in the sex determination pathway, whose activity/level is influenced by temperature, has the potential to directly respond to a temperature change to modulate sex (Capel, 2017), we thus attempted to identify such a protein in *C. elegans*. The currently known regulators of sex determination are not inherently thermosensitive; thus, we performed a feeding RNAi screen encompassing 82 genes to find temperature-responsive gene(s) involved in germline sex determination. These genes were selected based on their previous

association with temperature fluctuations or their potential involvement in TSD (Jaffe, 1995; Kohno et al, 2010; Sarge and Cullen, 1997; Weber et al, 2020), including genes encoding TRP channels, heat shock proteins (HSPs), and genes previously suggested to be involved in TSD in turtles (Fig. EV2A and Table 2). Based on the hypothesis described above, the inactivation of such genes could affect germline sex irrespective of the ambient culture temperature, thus the RNAi screen was performed at 20 °C in DE90. Interestingly, our RNAi screen revealed that feeding RNAi against *hsp-4*, which encodes one of the two thermosensitive BiP ER chaperone proteins in *C. elegans*, caused masculinization of the germline (Fig. EV2A,B; Table 2). Further analyses indicated that enhancing *hsp-4(RNAi)* efficacy by injecting corresponding dsRNA resulted in excess sperm production in 88% of the DE90 germlines at the nonmasculinizing temperature (20 °C), in contrast to the RNAi control animals displaying no masculinized germlines (Fig. 3A–D). Moreover, additional data indicated that *hsp-4(RNAi)* also caused Mog in the wild-type ED3017 and EG4725 worms at 20 °C (Fig. EV2C–F), thus the function of *hsp-4* in germline sex determination is not limited to DE90. Therefore, HSP-4 is a critical regulator of driving female sex determination.

To further confirm the role of HSP-4 in germline sex regulation, we analyzed whether the *hsp-4(lf)* mutation might also induce excess sperm production as *hsp-4(RNAi)*. Surprisingly, unlike the *hsp-4(RNAi)*-treated hermaphrodites, the *hsp-4(lf)* mutant hermaphrodites did not exhibit the Mog phenotype (Fig. EV2G). Since BiP proteins in *C. elegans* are encoded by two highly homologous genes, *hsp-3* and *hsp-4*, we postulated that *hsp-4(RNAi)* may knock down both *hsp-3* and *hsp-4* to cause Mog. Consistent with our prediction, *hsp-4(RNAi)* reduced the signals of both native HSP-4::wrmScarlet and HSP-3::wrmScarlet (Fig. EV2H–L). Next, we performed double mutants' analyses to assess whether the roles of *hsp-3* and *hsp-4* in germline sex determination are redundant. Similar to the *hsp-4(lf)* single mutant, the *hsp-3(lf)* single mutant produced a normal number of sperm (Fig. EV2G); however, the *hsp-3(lf);hsp-4(lf)* double mutation was lethal, preventing scoring the Mog phenotype. Thus, we evaluated the germline phenotype of other combinations of *hsp-3(lf)* and *hsp-4(lf)* mutations by scoring F1 progenies of *hsp-3(lf/+);hsp-4(lf/+)* mother, and the obtained results indicated that a subset of hermaphrodites displayed Mog, among which >90% were *hsp-4(lf);hsp-3(lf/+)* mutants (Fig. EV2G). These data indicate that the two BiPs act redundantly to modulate germline sex. Notably, these results show that BiP levels need to be precisely regulated and that Mog can be induced only when the BiP level is reduced to a proper level as that in the *hsp-4(lf);hsp-3(lf/+)* mutants, consistent with the multifaceted roles of BiPs. Taken together, these findings indicate that *hsp-4(RNAi)* knocked down both the HSP-3 and HSP-4 BiPs, which are required for driving oocyte fate, to result in germline masculinization.

## BiPs sense and translate temperature change into the germline sex determination signal in *C. elegans*

We next investigated whether BiP is thermosensitive by evaluating whether warmer temperatures may decrease BiP levels to drive male germline sex. BiP is a well-established major ER-resident chaperone that binds protein clients to facilitate their folding (Walter and Ron, 2011). High temperatures directly promote the protein unfolding process (Lapidus, 2017), which causes more BiP

**Table 2.** List of 82 genes used for RNAi screen for identifying the potential thermosensitive genes that regulate germline sex.

| Gene symbol | WormBase ID | Germline Mog phenotype (%) | Germline Fog phenotype (%) | n |
|---|---|---|---|---|
| xbp-1 | R74.3 | 0 | 0 | 66 |
| ubql-1 | F15C11.2 | 0 | 0 | 57 |
| hsf-1 | Y53C10A.12 | 0 | 0 | 55 |
| pqm-1 | F40F8.7 | 0 | 0 | 84 |
| ddl-1 | F59E12.10 | 0 | 0 | 68 |
| sti-1 | R09E12.3 | 0 | 0 | 78 |
| daf-16 | R13H8.1 | 0 | 0 | 76 |
| eya-1 | C49A1.4 | 0 | 0 | 76 |
| hda-1 | C53A5.3 | 0 | 0 | 54 |
| atx-3 | F28F8.6 | 0 | 0 | 64 |
| ddl-2 | Y48E1B.1 | 0 | 0 | 56 |
| hsp-1 | F26D10.3 | 0 | 0 | 100 |
| hsp-4 | F43E2.8 | 9.2 | 0 | 76 |
| ubxn-4 | ZK353.8 | 0 | 0 | 68 |
| atgl-1 | C05D11.7 | 0 | 0 | 66 |
| pdi-2 | C07A12.4 | 0 | 0 | 58 |
| daf-21 | C47E8.5 | 0 | 0 | 60 |
| icd-1 | C56C10.8 | 0 | 0 | 62 |
| apy-1 | F08C6.6 | 0 | 0 | 58 |
| sel-11 hrd-1 | F55A11.3 | 0 | 0 | 76 |
| atfs-1 | ZC376.7 | 0 | 0 | 70 |
| fic-1 | ZK593.8 | 0 | 0 | 60 |
| gale-1 | C47B2.6 | 0 | 0 | 86 |
| dve-1 | ZK1193.5 | 0 | 0 | 42 |
| manf-1 | Y54G2A.23 | 0 | 0 | 57 |
| spg-7 | Y47G6A.10 | 0 | 0 | 26 |
| pat-4 | C29F9.7 | 0 | 0 | 55 |
| hsp-6 | C37H5.8 | 0 | 0 | 100 |
| hsp-3 | C15H9.6 | 0 | 0 | 96 |
| ubl-5 | F46F11.4 | 0 | 0 | 105 |
| tomm-40 | C18E9.6 | 0 | 0 | 84 |
| tomm-20 | F23H12.2 | 0 | 0 | 68 |
| dnj-10 | F22B7.5 | 0 | 0 | 60 |
| mrp-7 | Y43F8C.12 | 0 | 0 | 76 |
| cyl-1 | C52E4.6 | 0 | 0 | 24 |
| tomm-22 | W10D9.5 | 0 | 0 | 67 |
| hsp-70 | C12C8.1 | 0 | 0 | 60 |
| hsp-110 | C30C11.4 | 0 | 0 | 66 |
| stc-1 | F54C9.2 | 0 | 0 | 58 |
| F11F1.1 | F11F1.1 | 0 | 0 | 65 |
| hsp-60 | Y22D7AL.5 | 0 | 0 | 64 |
| dnj-19 | T05C3.5 | 0 | 0 | 57 |

**Table 2.** (continued)

| Gene symbol | WormBase ID | Germline Mog phenotype (%) | Germline Fog phenotype (%) | n |
|---|---|---|---|---|
| dnj-12 | F39B2.10 | 0 | 0 | 58 |
| hsp-16.48 | T27E4.3 | 0 | 0 | 50 |
| hsp-16.41 | Y46H3A.2 | 0 | 0 | 48 |
| hsp-16.2 | Y46H3A.3 | 0 | 0 | 112 |
| hsp-12.2 | C14B9.1 | 0 | 0 | 72 |
| hsp-16.1 | T27E4.8 | 0 | 0 | 82 |
| hsp-25 | C09B8.6 | 0 | 0 | 90 |
| T14G8.3 | T14G8.3 | 0 | 0 | 72 |
| T24H7.2 | T24H7.2 | 0 | 0 | 60 |
| hsp-75 | R151.7 | 0 | 0 | 62 |
| hsp-12.6 | F38E11.2 | 0 | 0 | 80 |
| hsp-12.1 | T22A3.2 | 0 | 0 | 83 |
| hsp-43 | C14F11.5 | 0 | 0 | 68 |
| hsp-12.3 | F38E11.1 | 0 | 0 | 65 |
| hsp-17 | F52E1.7 | 0 | 0 | 80 |
| trpa-1 | C29E6.2 | 0 | 0 | 88 |
| TAX-2 | F36F2.5 | 0 | 0 | 70 |
| TAX-4 | ZC84.2 | 0 | 0 | 46 |
| GCY-8 | C49H3.1 | 0 | 0 | 40 |
| GCY-18 | ZK896.8 | 0 | 0 | 60 |
| GCY-23 | T26C12.4 | 0 | 0 | 59 |
| PDE-1 | T04D3.3 | 0 | 0 | 55 |
| PDE-2 | R08D7.6 | 0 | 0 | 60 |
| PDE-5 | C32E12.2 | 0 | 0 | 64 |
| DGK-3 | F54G8.2 | 0 | 0 | 45 |
| PKC-1 | F57F5.5 | 0 | 0 | 78 |
| osm-9 | B0212.5 | 0 | 0 | 59 |
| kqt-1 | C25B8.1 | 0 | 0 | 84 |
| kqt-3 | Y54G9A.3 | 0 | 0 | 70 |
| trp-1 | ZC21.2 | 0 | 0 | 46 |
| trp-2 | R06B10.4 | 0 | 0 | 110 |
| trp-3 | K01A11.4 | 0 | 0 | 72 |
| trp-4 | Y71A12B.4 | 0 | 0 | 68 |
| trpa-2 | M05B5.6 | 0 | 0 | 66 |
| ced-11 | ZK512.3 | 0 | 0 | 82 |
| sta-1 | Y51H4A.17 | 0 | 0 | 65 |
| sta-2 | F58E6.1 | 0 | 0 | 84 |
| jmjd-3.1 | F18E9.5 | 0 | 0 | 64 |
| jmjd-3.2 | F23D12.5 | 0 | 0 | 67 |
| jmjd-3.3 | C29F7.6 | 0 | 0 | 50 |

The knockdown was performed in the DE90 strain by feeding RNAi methods. The germline sex phenotypes, Mog and Fog, were scored and are shown in the table. 'n' represents the number of worms used for analysis.

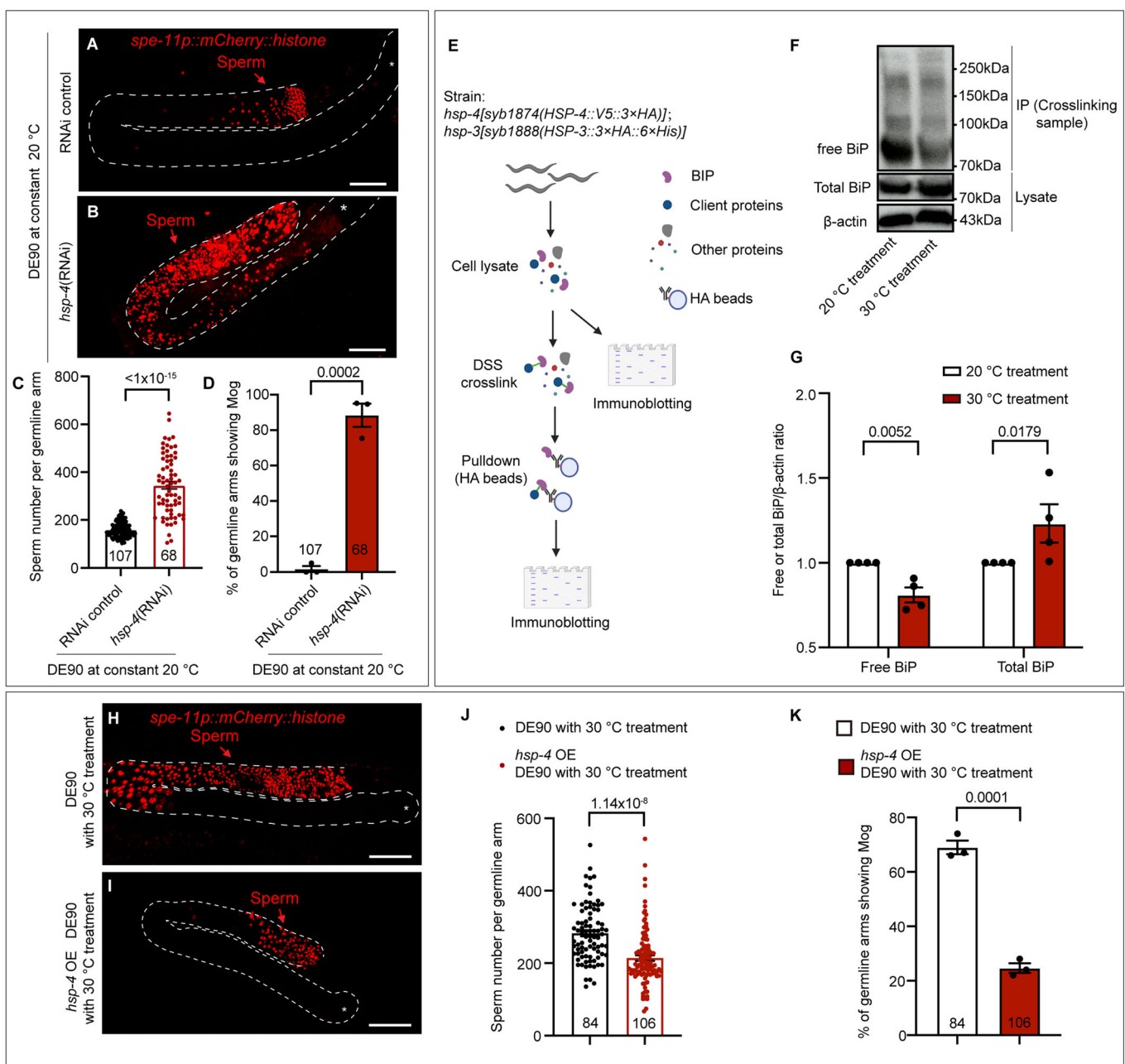

to be allocated for maintaining protein folding and thereby reduces the amount of available BiP (Vitale et al, 2019). Since *hsp-4(RNAi)* caused Mog, we postulated that the amount of available BiP might be reduced to mediate the masculinizing effect of the warmer temperature. Therefore, we performed a gel crosslinking analysis to evaluate the amount of available BiP (Fig. 3E). As expected, both western blot analysis of BiPs (Fig. 3F,G) and microscopic analyses of native HSP-4::wrmScarlet and HSP-3::wrmScarlet (Fig. EV3A–F) indicated that the level of total BiP, including both HSP-3 and HSP-4, were induced by warmer temperature treatment, due to an increased demand of BiP for facilitating ER protein folding. In contrast, the free BiP level was obviously decreased in response to the increase in temperature (Fig. 3F,G), indicating the temperature

responsiveness of BiP. Moreover, according to previous findings that a reduction in available BiP could induce BiP transcription through negative feedback regulation (Carrara et al, 2015), we further confirm the decrease in the amount of available BiP by analyzing the transcription of BiP in response to a temperature increase. Similar to the finding in the positive control that a decrease in BiP resulted from the *hsp-4(RNAi)* induced the expression of *hsp-4p::GFP* (Fig. EV3G–I), treatment with the masculinizing temperature (30 °C) also increased *hsp-4p::GFP* expression (Fig. EV3J–L). Next, we further evaluated the effect of warmer temperature treatment on the transcription of BiP in the germline. qPCR analyses of the *hsp-4* transcripts extracted from the dissected germline of DE90 indicated a two-fold increase in *hsp-4*

◀

**Figure 3. The thermosensitive ER chaperone BiP mediates the influence of temperature on germline sex.**

(A–D) Representative images and bar graphs showing strong Mog caused by *hsp-4*(RNAi) in the DE90 strain cultured at constant 20 °C. The *hsp-4*(RNAi) experiment was performed by microinjecting the corresponding dsRNA into DE90 worms. A quantitative analysis of the sperm number is shown in (C). After the *hsp-4*(RNAi) treatment, 88.4% of germline arms displayed Mog (D). Each dot represents the sperm number in one germline arm in (C) and the percentage of germline arms showing Mog in each replicate in (D). $P$ value $<1 \times 10^{-15}$ (C), $P$ value $= 0.0002$ (D). Scale bar: 30 μm. (E) Workflow for the detection of free BiP levels by using the disuccinimidyl suberate (DSS)-mediated protein crosslinking method. An equal number of KI [HSP-3::3×HA::6×His];KI [HSP-4::V5::3×HA] worms with indicated temperature treatments were homogenized in lysis buffer. A small portion of the cell lysate was used for SDS–PAGE to measure the total BiP level. The remaining cell lysate was then subjected to DSS crosslinking, and BiP was pulled down using HA beads for western blot analysis of free BiP levels. (F, G) Western blot analysis and corresponding quantitative analysis indicating an obvious reduction in free BiP and an increase in total BiP levels after warmer temperature (30 °C). Free BiP and total BiP were detected as described in (E) and β-actin was used to indicate equal loading of samples. BiP is encoded by *hsp-3* and *hsp-4* in *C. elegans*, and KI [HSP-3::3×HA::6×His];KI [HSP-4::V5::3×HA] worms with the indicated temperature treatment were analyzed to determine the BiP level. The corresponding quantitative analysis of free and total BiP/β-actin ratio was presented in (G). Each dot represents the value of free BiP/β-actin or total BiP/β-actin ratio in each replicate. $P$ value $= 0.0052$, $P$ value $= 0.0179$ (G) (from left to right). (H–K) Micrographs and bar graphs showing the suppression of warmer temperature-induced Mog by the overexpression of HSP-4. Sperm number in DE90 with or without the Is [hsp-16.41p::HSP-4::Flag::HA] transgene were scored after the 30 °C treatment. The heat shock promoter *hsp-16.41* was used to drive the overexpression of *hsp-4* at 30 °C. The corresponding quantitative analysis of sperm number is shown in (J). The percentage of germline arms displaying Mog was reduced from 69 to 24.6% after the overexpression of *hsp-4* in DE90 with warmer temperature treatment (K). Each dot represents the sperm number in one germline arm in (J) and the percentage of germline arms showing Mog in each replicate in (K). $P$ value $= 1.14 \times 10^{-8}$ (J), $P$ value $= 0.0001$ (K). Scale bar: 30 μm. For all the panels, the asterisk (*) in the microscopic images indicates the distal end of the germline and white dashed lines outline the germline arms. The data are shown as the means ± SEMs. Statistics were performed by unpaired *t* test. At least three biological independent experiments were performed. Source data are available online for this figure.

mRNA levels in the 30 °C treatment group compared to 20 °C treatment group (Fig. EV3M), further supporting the idea that warmer temperature triggered a decrease in available BiP in the germline of *C. elegans*. Our findings, including the warmer temperature-caused reduction in the amount of available BiP (Figs. 3F,G and EV3J–L) as well as the *hsp-4(RNAi)*-induced excess sperm production (Fig. 3A–D), support the idea that a decrease in the level of available BiP mediates the role of warmer temperature in promoting male germline fate.

To further demonstrate that the decrease in the amount of available BiP is responsible for warmer temperature-induced Mog, we next investigated whether overexpressing BiP could suppress the excess sperm production caused by warmer temperature. For this purpose, we constructed an integrated transgene carrying *hsp-16.41p::HSP-4::Flag::HA* in the DE90 strain, and this *hsp-16.41* heat shock promoter increased HSP-4 expression in response to warmer temperature (Fig. EV3N,O). The overexpression of HSP-4 BiP indeed suppressed warmer temperature-induced Mog, as indicated by the reduction in the percentage of masculinized germline at 30 °C from 69 to 24.6% (Fig. 3H–K). Together with the observations that *hsp-4(RNAi)* induced Mog (Fig. 3A–D) and that warmer temperature triggered a reduction in the amount of available BiP (Figs. 3F–G and EV3J–L), these results demonstrate that the amount of available BiP is reduced in response to the increased temperature to drive sperm fate.

In addition, we explored whether previously known process that regulates BiP expression might affect worms' sensitivity in displaying TGSD. The ER unfolded protein response regulator, X-box binding protein 1 (XBP-1), is required for the induction of HSP-4 upon heat stress (Calfon et al, 2002; Xu et al, 2024), thus it is reasonable to postulate that the *xbp-1(lf)* mutation, which was unable to induce BiP levels when exposed to warmer temperatures, may make the worms more prone to adopt male germline sex. Indeed, *xbp-1(lf)* mutants displayed masculinized germlines upon 30 °C treatment, in contrast to the ones maintained at 20 °C (Fig. EV3P), which thus further supports that decreased BiP levels mediate the warmer temperature-induced Mog effect. Given that high temperature directly promotes ER protein unfolding that sequesters BiP (Cooley et al, 2014; Nehls et al, 2000) (Fig. 3F,G),

our findings support that BiP acts as a temperature sensor likely through detecting temperature-caused change in ER folding dynamics to mediate TGSD and the reduction in available BiP translates warmer temperature stimulus into the germline sex determination signal to drive sperm fate in *C. elegans*.

## BiP drives female-fated germline sex by preventing the reduction of TRA-2 levels

After determining the role of BiP in mediating the impact of temperature on germline sex, we next aimed to illustrate the underlying mechanism. Given our earlier experiments showing that temperature affects TRA-2 level to influence germline sex, we hypothesized that BiP acts upstream of TRA-2 to modulate its expression. Consistent with our prediction, *hsp-4(RNAi)*, resulting in Mog at nonmasculinizing temperature (20 °C), induced an increase in *fog-3* transcription in DE90, and *fog-3(lf)* suppressed the *hsp-4(RNAi)*-induced excess sperm production (Fig. 4A,B), indicating that BiP acts upstream of *fog-3* to regulate germline sex. Further epistatic analyses showed that both *fem-3(lf)* and *tra-2(gf)* were able to suppress *hsp-4(RNAi)*-induced Mog (Fig. 4B,C), indicating that BiP functions upstream of *tra-2* to affect germ cell fate, similar to the effect of warmer temperature (Fig. 2B,L). These genetic analyses support the idea that BiP acts upstream of TRA-2 to translate temperature cues into activities of the germline sex determination pathway to modulate germ cell fate.

We then aimed to understand the mechanism by which BiP, a well-known ER lumen chaperone molecular (Hamman et al, 1998), functions upstream of TRA-2. Since the relative ratio of TRA-2 to FEM-3 is known to be crucial for germline sex determination (Ellis and Schedl, 2007), we investigated whether ER lumen-resident BiP physically interacts with TRA-2 or FEM-3. Given that ER-localized TRA-2 was too low to be detected in *C. elegans* (Figs. 2E–J and EV1C–F), we co-expressed the corresponding worm proteins in HEK293T cells to detect their interactions. Interestingly, coprecipitation of full-length transmembrane TRA-2 with HSP-4::GFP was clearly observable when GFP-Trap beads were used to pull down the HSP-4::GFP protein (Fig. 4D), which is in line with our earlier observation that TRA-2 is synthesized in the ER where a regulation

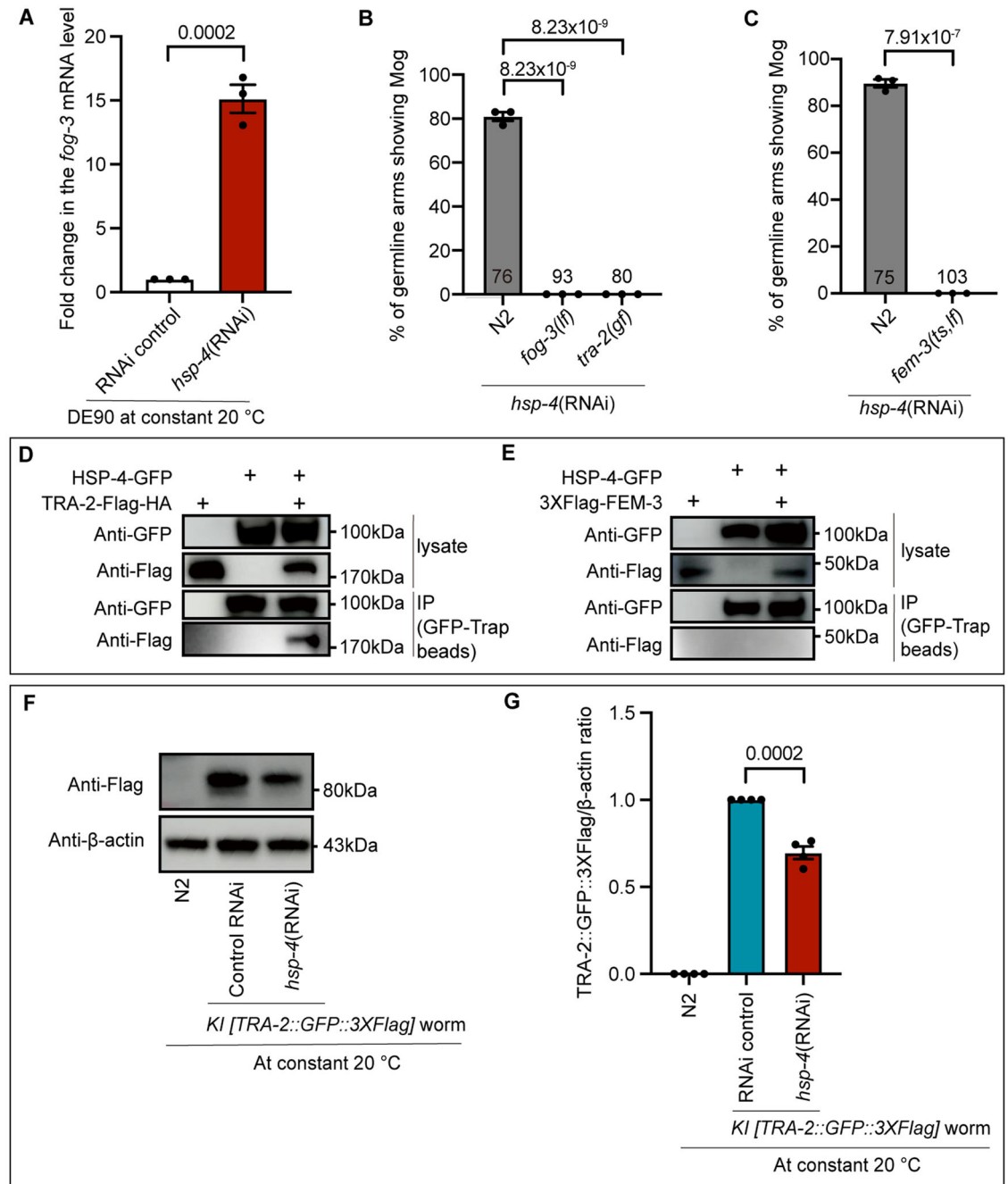

of TRA-2 expression could occur (Fig. EV1G–I). In contrast, the coprecipitation of FEM-3 with HSP-4 was not observed; therefore, HSP-4 and cytosolic FEM-3 do not physically interact (Fig. 4E). Collectively, these findings demonstrate that before trafficking to the plasma membrane, transmembrane TRA-2 in the ER interacts with the ER lumen chaperone BiP.

The interaction of ER-resident BiP with TRA-2 might act as a posttranslational mechanism for regulating the TRA-2 level at the ER before its translocation to the plasma membrane; therefore, we investigated whether knockdown BiP would cause a reduction in the level of oocyte-driving TRA-2. Same as the warmer temperature treatment, *hsp-4(RNAi)* also resulted in a reduction

in TRA-2 level in the worms cultured at a constant 20 °C (Fig. 4F,G), indicating that BiP critically modulates the TRA-2 expression. Together with the result that *tra-2(gf)*, which increased TRA-2 expression in the germline, suppressed *hsp-4(RNAi)*-induced Mog (Fig. 4B), these findings demonstrate that a decreased amount of available BiP induces a reduction in TRA-2 level to drive male germline fate. Moreover, same as TRA-2, BiP also functions in the germline to modulate sperm/oocyte fate, as evidenced by the induction of Mog by *hsp-4(global RNAi)* and *hsp-4(germline-RNAi)* but not *hsp-4(intestine-RNAi)* (Appendix Fig. S4), which aligns with the idea that BiP modulates TRA-2 expression to alter germ cell fate. In summary, after the 30 °C

**Figure 4. BiP promotes female germline sex by positively regulating the level of TRA-2.**

(A) qPCR analysis indicating that *hsp-4*(RNAi) caused an increase in the *fog-3* mRNA level. The *fog-3* transcript in the DE90 hermaphrodites cultured at a constant 20 °C, receiving the indicated RNAi treatment, were analyzed using qPCR. Each dot represents the fold change in the *fog-3* mRNA level in each replicate. P value = 0.0002. (B) Bar graph showing the suppression of *hsp-4*(RNAi)-induced Mog by the *fog-3*(lf) and *tra-2*(e2020, gf) mutations. The worms with the indicated genotypes and treatments, cultured at a constant 20 °C, was scored for the presence of Mog. After *hsp-4*(RNAi) treatment, the percentages of germline arms showing Mog were 79.8%, 0% and 0% in the N2, *fog-3*(lf), and *tra-2*(gf) worms, respectively. Each dot represents the percentage of germline arms showing Mog in each replicate. P value = $8.23 \times 10^{-9}$, P value = $8.23 \times 10^{-9}$ (from left to right). (C) Bar graph indicating that the *fem-3*(ts, lf) mutation suppresses Mog caused by *hsp-4*(RNAi). RNAi was performed in *fem-3*(ts, lf) and N2 worms at 25 °C, at which temperature *fem-3*(ts, lf) worms showed a mutant phenotype. The percentage of germline arms displaying Mog reduced to 0% in *fem-3*(ts, lf);*hsp-4*(RNAi) worms, while it was 90.8% in *hsp-4*(RNAi)-treated N2 hermaphrodites. Each dot represents the percentage of germline arms showing Mog in each replicate. P value = $7.91 \times 10^{-7}$. (D, E) Western blot images show the coprecipitation of HSP-4 and TRA-2, but no coprecipitation of HSP-4 and FEM-3. HSP-4::GFP was co-expressed with TRA-2::Flag::HA or 3×Flag::FEM-3 in 293T cells. GFP-Trap beads were used to pull down HSP-4::GFP, and an anti-Flag antibody was then used to detect the potential coprecipitation of TRA-2 (D) and FEM-3 (E). (F, G) Western blot images and the corresponding quantitative analyses showing a reduction of TRA-2 by BiP knockdown. The KI [TRA-2::GFP::3×Flag] strain receiving the indicated RNAi treatment was analyzed by western blot using an anti-Flag antibody to determine the levels of TRA-2. N2 worms served as a negative control to determine the specificity of the antibody, and β-actin was used to indicate equal loading of samples. The fold change in the TRA-2::GFP::3×Flag/β-actin ratio was calculated by normalizing to the RNAi control group (G) and each dot represents the value from the calculation in each replicate. P value = 0.0002 (G). The *hsp-4* RNAi experiments were conducted by microinjection with the corresponding dsRNA in (A–C) and by feeding in (F, G). The data are shown as the means ± SEMs. Statistics were performed by unpaired *t* test (A, C, G) or one-way ANOVA (B). Three biological independent experiments were performed. Source data are available online for this figure.

treatment, the availability of BiP decreases, leading to a decrease in the expression of TRA-2 and consequently resulting in Mog in the hermaphrodites.

## BiP promotes oocyte fate by preventing ERAD-mediated degradation of functional TRA-2

Next, we aimed to understand the mechanism by which BiP regulates TRA-2 levels. The *hsp-4(RNAi)*-induced reduction in TRA-2 expression (Fig. 4F,G) suggests that this ER lumen-localized molecular chaperone may prevent TRA-2 degradation at the ER before TRA-2 translocates to the plasma membrane. ER-associated protein degradation (ERAD) is a major mechanism for targeting both misfolded and functional ER proteins to the proteasome for degradation (Ruggiano et al, 2014). In *C. elegans*, the critical ERAD factor P97, responsible for shuttling and remodeling ER proteins for proteasome degradation in the cytosol, is encoded by *cdc-48.1* and *cdc-48.2*, which play crucial roles in the organism's development. Specifically, disrupting the functions of both *cdc-48.1* and *cdc-48.2* leads to developmental arrest (Fig. EV4A–E) and intriguingly, *cdc-48.1* was implicated in the regulation of germline sex in *C. elegans* with unclear mechanisms (Sasagawa et al, 2009). Moreover, it was previously shown that coupling with the P97 complex, CUL-2, a ubiquitin ligase that governs proteasomal degradation, also modulates the switch from spermatogenesis to oogenesis (Sasagawa et al, 2009). Thus, we investigated whether these two ERAD regulators, CDC-48.1 and CUL-2, may mediate the *hsp-4(RNAi)*-induced reduction in TRA-2 levels by analyzing the intensity of native-expressed TRA-2::GFP. Interestingly, we found that after the *hsp-4(RNAi)* treatment, *cul-2(RNAi)* was able to partially enhance the TRA-2::GFP level and *cdc-48.1(lf)* showed no obvious effect on reversing the decrease in TRA-2::GFP intensity; however, *cdc-48.1(lf);cul-2(RNAi)* was able to drastically reverse the TRA-2::GFP level reduced by the *hsp-4(RNAi)* treatment (Fig. 5A). These results support the idea that the ERAD pathway is required for the degradation of TRA-2 when the amount of available BiP is decreased. Moreover, consistent with the synergistic effect of *cdc-48.1(lf)* and *cul-2(RNAi)* on recovering TRA-2 level, *cdc-48.1(lf);cul-2(RNAi)* suppressed the *hsp-4(RNAi)*-caused Mog

phenotype (Fig. 5B). This suppression of the *hsp-4(RNAi)*-induced Mog by the inhibition of ERAD indicates that the functional TRA-2, rather than the misfolded TRA-2, is degraded by ERAD in the *hsp-4(RNAi)-treated* worms. These results support the idea that the decrease in BiP is unlikely to cause TRA-2 misfolding and may instead predispose ER-localized functional TRA-2 to undergo degradation by ERAD. This ERAD-mediated degradation of functional TRA-2 is consistent with previous studies showing that the ERAD-mediated removal of functional proteins is an important posttranslational mechanism for regulating gene expression (Ruggiano et al, 2014).

To determine the direct role of CDC-48.1 in mediating the reduction of TRA-2, we investigated a potential interaction between these two proteins by coprecipitation. Since the biosynthesis of TRA-2 at ER is transient and thereby under detectable level in *C. elegans* (Fig. EV1C–F), TRA-2::Flag::HA and P97(CDC-48.1)::GFP::HA::6×His were co-expressed in HEK293T cells to facilitate the detection of TRA-2 and P97 interaction. Indeed, after P97 pulldown with GFP-Trap beads, we observed the coprecipitation of TRA-2 with P97 but at a low level (Fig. 5C), which may be due to the rapid transfer of TRA-2 to the proteasome for degradation once the P97 unfolds TRA-2 substrate. Therefore, we investigated whether inhibiting proteasome activity could enhance the level of coprecipitated TRA-2 during the pulldown of CDC-48.1. Indeed, this coprecipitation of TRA-2 with P97 was dramatically increased after adding the proteasome inhibitor MG132 (Fig. 5C), which further demonstrates the role of P97 in targeting ER-localized TRA-2 for subsequent proteasomal degradation. Therefore, our data support the idea that BiP prevents the ERAD-mediated degradation of functional TRA-2 and thereby drives oocyte fate.

To further confirm the function of ERAD in mediating *hsp-4(RNAi)*-induced TRA-2 degradation, we next investigated the role of RPN-10, a ubiquitin receptor that delivers client proteins of ERAD from ER to the 26 S proteasome, in mediating the involvement of BiP in germline sex determination. Interestingly, the *rpn-10(lf)* mutation suppressed the decrease in the TRA-2 level as well as the excess sperm production caused by BiP knockdown (Fig. EV4F–G), thus further supporting that TRA-2 is degraded by ERAD when the amount of BiP is decreased. This result is

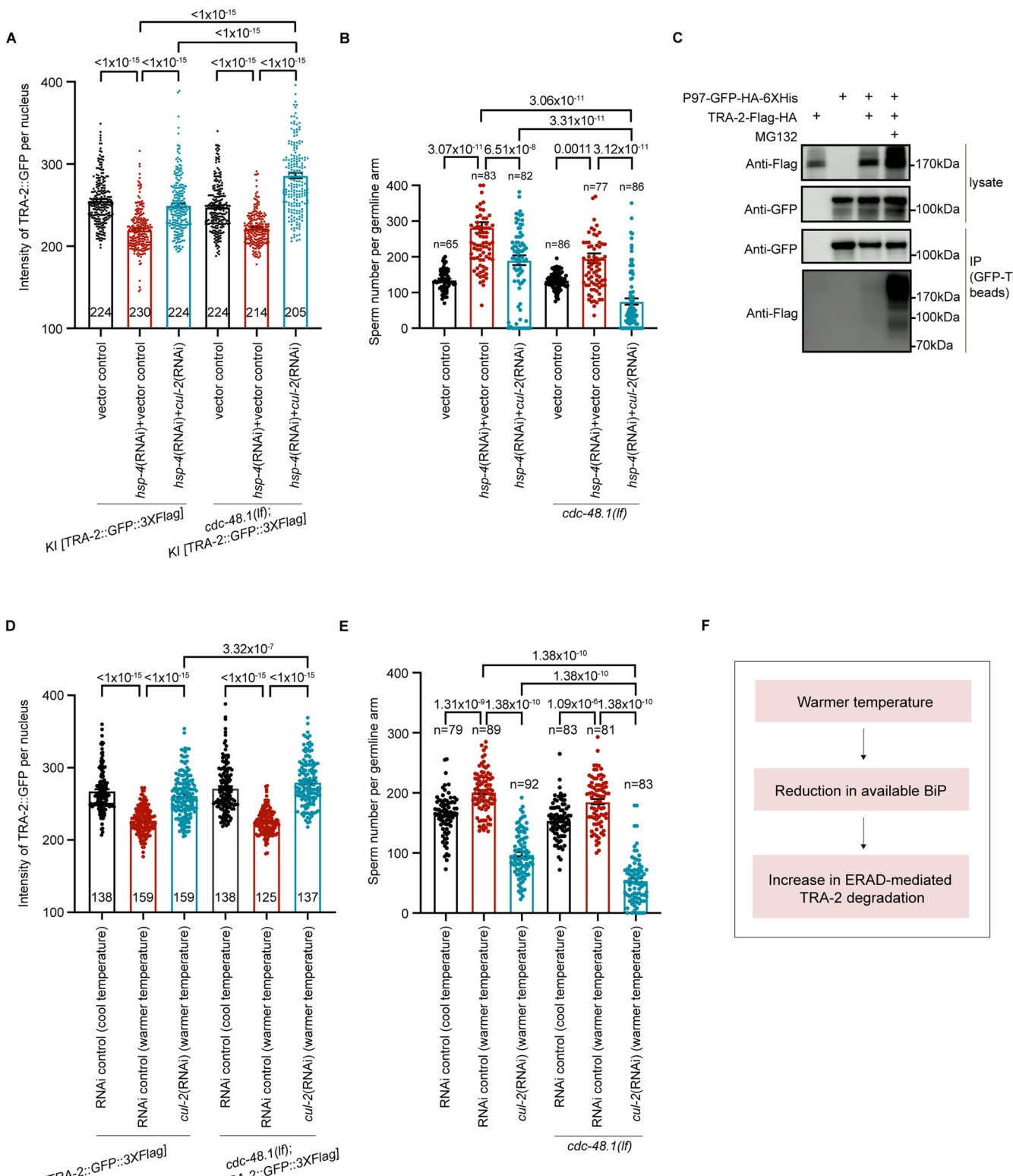

consistent with previous observations showing that RPN-10 functions in germline sex determination by affecting TRA-2 expression (Shimada et al, 2006). Together, BiP prevents the ERAD-mediated degradation of functional TRA-2 to drive female germline sex.

Given the role of BiP in TGSD, our above results suggest that ERAD should also be involved in the warmer temperature-induced degradation of TRA-2 and resultant Mog. Indeed, *cdc-48.1(lf);cul-2(RNAi)* and *rpn-10(lf)* suppressed the warmer temperature-induced decrease in TRA-2::GFP level (Figs. 5D and EV4H),

◀ Figure 5. BiP drives oocyte fate by preventing P97-mediated degradation of TRA-2 and thereby mediates temperature-dependent germline sex change.

(A) Bar graph showing that the decrease in TRA-2 levels induced by *hsp-4*(RNAi) was inhibited by the *cdc-48.1(lf);cul-2(RNAi)*. The *cdc-48.1(lf)* mutation was generated in the *KI [TRA-2::GFP::3×Flag]* strain using the CRISPR–Cas9 method. RNAi experiments were conducted by feeding. The vector control refers to the worms treated with HT115 containing the empty PL4440 vector. For double genes' RNAi, the indicated RNAi bacteria were mixed at a 1:1 ratio. The worms with indicated treatment were cultured at a constant 25 °C, at which *cdc-48.1(lf)* display mutant phenotype as previously shown (Sasagawa et al, 2009). The fluorescence intensity of native TRA-2::GFP::3×Flag in the nuclei of the intestinal cells was measured to determine the TRA-2 level. All *P* value < 1 × 10⁻¹⁵. (B) Bar graphs indicating the suppression of *hsp-4*(RNAi)-induced excess sperm production by the *cdc-48.1(lf);cul-2(RNAi)*. The feeding RNAi and culture temperature information is the same as in (A). The fluorescence of *spe-11p::his-58::mCherry* was analyzed to determine the sperm number. The mean sperm number in *cdc-48.1(lf);cul-2(RNAi)* worms decreased to 74, compared to 282 in the ones treated with *hsp-4(RNAi);vector control*. *P* value = $3.07 × 10^{-11}$, *P* value = $6.51 × 10^{-8}$, *P* value = $3.06 × 10^{-11}$, *P* value = $3.31 × 10^{-11}$, *P* value = 0.0011, *P* value = $3.12 × 10^{-11}$ (from left to right). (C) Western blot images showing the coprecipitation of P97 with TRA-2. P97::GFP::HA::6×His and TRA-2::Flag::HA were co-expressed in 293 T cells and then treated with/without MG132. GFP-Trap magnetic beads were used to pull down P97, and an anti-Flag antibody was used to detect the coprecipitation of TRA-2. (D) Bar graph showing the suppression of the decrease in TRA-2 levels caused by the warmer temperature (30 °C) treatment through the *cdc-48.1(lf);cul-2(RNAi)*. Due to the temperature sensitivity of *cdc-48.1(lf)*, the worms were cultured at a constant 25 °C (cool temperature) to observe the mutant phenotype (Sasagawa et al, 2009). RNAi was performed by feeding. *P* value < 1 × 10⁻¹⁵, *P* value < 1 × 10⁻¹⁵, *P* value = $3.32 × 10^{-7}$, *P* value < 1 × 10⁻¹⁵, *P* value < 1 × 10⁻¹⁵ (from left to right). (E) Bar graph indicating that the *cdc-48.1(lf);cul-2(RNAi)* suppresses the excess sperm production induced by the warmer temperature (30 °C) treatment. As mentioned in (D), the worms with indicated genotypes and feeding RNAi treatment were maintained at a constant 25 °C (cool temperature). Each dot represents the sperm number in one germline arm. *P* value = $1.31 × 10^{-9}$, *P* value = $1.38 × 10^{-10}$, *P* value = $1.38 × 10^{-10}$, *P* value = $1.09 × 10^{-6}$, *P* value = $1.38 × 10^{-10}$, *P* value = $1.38 × 10^{-10}$ (from left to right). (F) A schematic summarizes that warmer temperatures induce insufficient amount of available BiP and then leads to the increased degradation of TRA-2 through ERAD, according to our results from Figs. 3 and 5. The data are shown as the means ± SEMs. Each dot in the bar graph represents the mean intensity of TRA-2::GFP in one nucleus in (A, D) and the number of sperm per germline arm in (B, E). Statistical significance was performed by one-way ANOVA. Three biological independent replicates were performed. Source data are available online for this figure.

indicating that the reduction in TRA-2 levels resulting from warmer temperature exposure was dependent on ERAD. Furthermore, in accordance with their impacts on reversing the reduction of TRA-2, *cdc-48.1(lf);cul-2(RNAi)* and *rpn-10(lf)* were able to reverse warmer temperature-triggered Mog in the DE90 strain background as evidenced by the suppression of the excess sperm production (Figs. 5E and EV4I). If warmer temperature treatment induces the misfolding of TRA-2 to target it for ERAD, then the accumulation of misfolded TRA-2 resulting from *cdc-48.1(lf);cul-2(RNAi)* or *rpn-10(lf)* would not be expected to reverse warmer temperature-induced Mog; therefore, our results support that similar to the knockdown of BiP, warmer temperature promotes the degradation of functional TRA-2 through ERAD. Additionally, our data indicated that the decrease in TRA-2 protein levels resulting from warmer temperature treatment was not due to a reduction in *tra-2* transcription (Fig. EV4J), thus further supporting the crucial role of posttranslational regulation of TRA-2 through ERAD in the observed TRA-2 reduction. In summary, in response to an increase in temperature, the amount of available BiP is reduced, which induces the ERAD-mediated degradation of functional TRA-2 and thereby results in male germline fate (Fig. 5F).

## Manipulating basal activity of germline sex determination pathway confers the transition between GGSD and TGSD in *C. elegans*

We next aimed to understand the mechanism that permits the transition of the germline sex determination mode from GGSD to TGSD and thus allows sex to be regulated by both karyotype and temperature in *C. elegans*. Our aforementioned results found that the basal level of *fog-3* was much lower in N2 (displaying GGSD) than in DE90 (exhibiting TGSD) (Fig. 2A), which thus prompted us to speculate that reducing the basal level of the oocyte fate-driving activity of this pathway, such as decreasing TRA-2 level, might enable N2 to show germline masculinization in response to warmer temperatures. We thus established a transgene carrying *Is [spe-11p::his-58::mCherry]* sperm marker in N2 background to facilitate

counting sperm and as expected, this N2-derived transgenic strain showed no Mog after warmer temperature treatment (Fig. 6A,B,G). Next, we generate a *tra-2* loss-of-function allele in this *Is [spe-11p::his-58::mCherry] C. elegans* and then used the *tra-2(lf/+)* heterozygotes, which exhibited a decrease in the oocyte fate-driving activity of the germline sex determination pathway, to test this hypothesis. When compared to the 20 °C treatment control, a temperature shift to 30 °C for 12 h at the late L3 stage induced an obvious increase in sperm production in this *tra-2(lf/+)* heterozygous mutant (Fig. 6C,D,G), indicating that hermaphrodites with low female-fated activity in the sex determination pathway are sensitive to the warmer temperature and that TGSD is thus permitted. Conversely, recovering the female fate-driving activity of the germline sex determination pathway conferred the switch back from TGSD to GGSD (Fig. 6H–J). Specifically, we found that the *rpn-10(lf)*, capable of increasing the TRA-2 level (Shimada et al, 2006) (Fig. EV4H), desensitized *tra-2(lf/+)* worms to the effect of the warmer temperature, as indicated by the suppressive effect of the *rpn-10(lf)* mutation on Mog in the high-temperature-treated *tra-2(lf/+)* worms (Fig. 6H–J). Therefore, manipulating the basal activity of the germline sex determination pathway confers the transition between GGSD and TGSD, thus revealing a potential mechanism by which animals may acquire the coexistence of GGSD and TGSD during evolution.

Moreover, we used the *hsp-4(lf)* mutant to further test the hypothesis that decreasing the female fate-driving activity sensitizes the responses of the worms to the warmer temperature. HSP-3/-4 BiPs both drive oocyte fate and hermaphrodites with the *hsp-4(lf)* mutation displayed no obvious Mog at 20 °C (Fig. 6A,B,E–G), due to the functional redundancy from HSP-3. In striking contrast, in response to warmer temperature treatment, *hsp-4(lf)* single-mutant hermaphrodites produced excess sperm (Fig. 6E–G). Collectively, these results further support the idea that reducing the basal level of the sex determination pathway activity of driving oocyte fate permits the *C. elegans* germline to proceed to male-fated development in response to warmer temperature and allows the genotypic sex determination with a temperature override; conversely, increasing the basal activity of the oocyte fate-driving

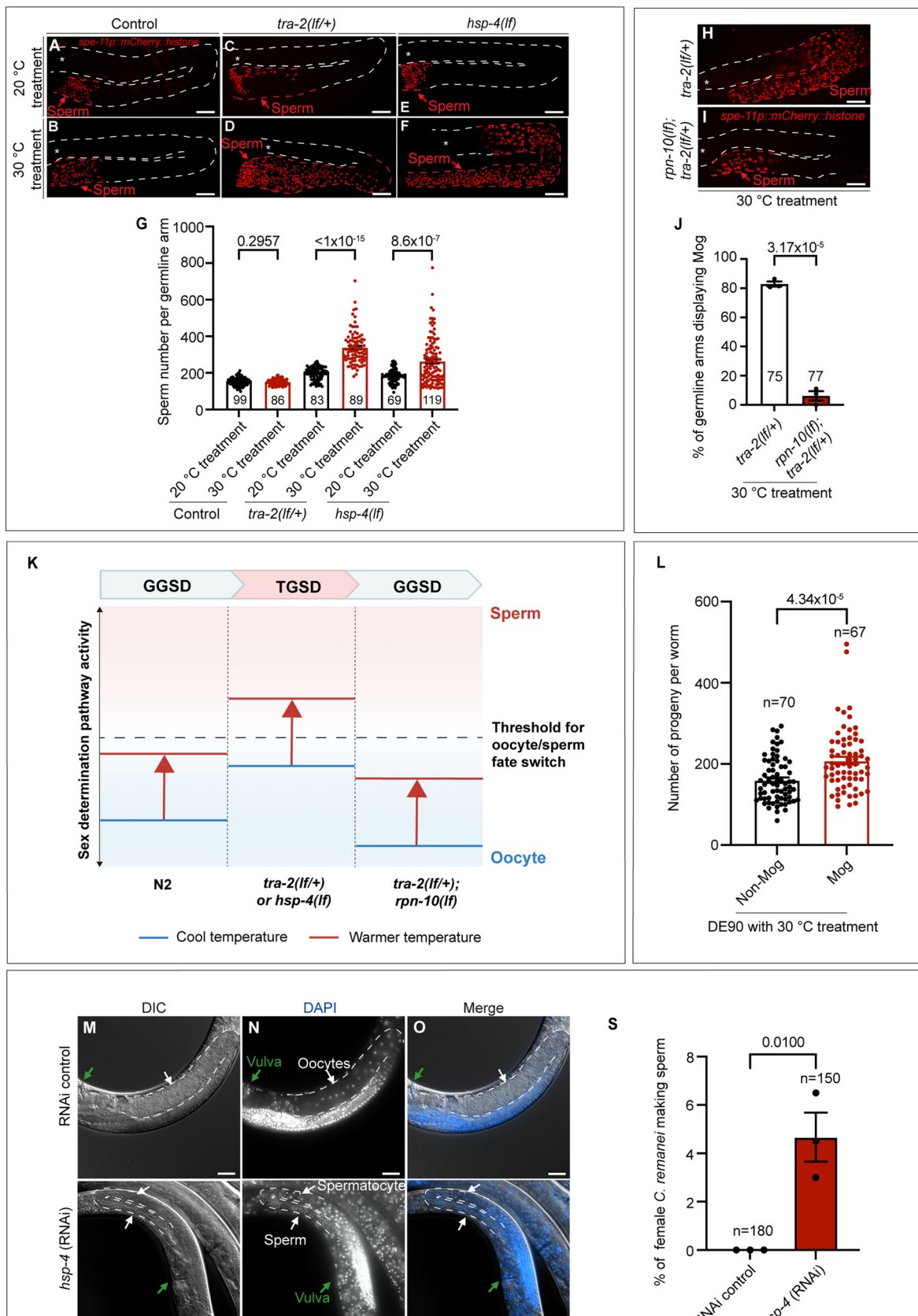

**Figure 6. Manipulating the activity of GGSD pathway permits the transition between GGSD and TGSD in *C. elegans*, and BiP is also required for driving female germline sex in a dioecious nematode.**

(A–G) Representative fluorescence images and corresponding quantitative analysis showing that downregulating the oocyte fate-driving activity of the germline sex determination pathway by introducing *tra-2(lf/+)* heterozygous mutation or *hsp-4(lf)* homozygous mutation permits the animals to show increased sperm production after the 30 °C treatment. The *Is [spe-11p::his-58::mCherry]* transgene was created in the N2 strain, and this transgenic strain did not show Mog in response to the 30 °C treatment, thereby serving as a negative control. A *tra-2(lf/+)* was further introduced into this *Is [spe-11p::his-58::mCherry]* strain using CRISPER-Cas9 method and the *tra-2(lf/+)* carrying this transgene was used for the analyses. The *hsp-4(lf)* mutants carrying the *Is [spe-11p::his-58::mCherry]* transgene were obtained by crossing. The quantification of sperm number in one germline arm is illustrated in (G). P value = 0.2957, P value < $1 \times 10^{-15}$, P value = $8.6 \times 10^{-7}$ (G) (from left to right). Scale bar: 20 μm. (H–J) Images and a bar graph showing that upregulating the oocyte fate-driving activity of the germline sex determination pathway by *rpn-10(lf)* reverses the ability of the *tra-2(lf/+)* mutant to display Mog in response to the 30 °C treatment. The percentage of germline arms showing Mog in *tra-2(lf/+)* worms with 30 °C treatment decreased from 82.8 to 6.1% due to the *rpn-10(lf)* mutation (J). P value = $3.17 \times 10^{-5}$ in (J). Scale bar: 20 μm. (K) The diagram illustrating the proposed rationale to account for the transition between GGSD and TGSD based on the results in (A–L). The blue line represents the activity of the germline sex determination pathway at cool temperature, while the red line represents it at a warmer temperature. In N2 wild-type worms, the increase in the sperm-fated activity of the germline sex determination pathway induced by the warmer temperature does not reach the threshold required for switch to the sperm fate, and consequently, the Mog phenotype could not be observed. However, *tra-2(lf/+)* or *hsp-4(lf)* mutations, which upregulate the sperm fate-driving activity of the germline sex determination pathway, allows the threshold to be surpassed by warmer temperature treatment and thereby permits the switch from GGSD to TGSD in *C. elegans*. Conversely, increasing the oocyte fate-driving activity of the germline sex determination pathway can switch from TGSD back to GGSD by introducing the *rpn-10(lf)* mutation into the *tra-2(lf/+)* mutant. (L) Bar graph showing that in DE90 worms with warmer temperature treatment, those displaying Mog can produce an increased number of progenies compared to the ones showing no Mog. The progenies of the DE90 hermaphrodites with/without Mog were scored after the 30 °C treatment. P value = $4.34 \times 10^{-5}$. (M–S) Images and bar graph showing that *hsp-4(RNAi)* induces the production of sperm in a subset of *C. remanei* females. *hsp-4(RNAi)* was performed by injecting corresponding dsRNA into the germline of *C. remanei* grown at a constant 20 °C. DAPI staining was used to determine the presence of sperm and spermatocytes. A total of 4.7% of the females subjected to *hsp-4(RNAi)* produced sperm (S). P value = 0.0100 in (S). Scale bar: 30 μm. The asterisk (*) indicates the distal end of the germline and dashed lines outlined the germline arms. Each dot represents the number of sperm per germline arm in (G), the percentage of germline arms showing Mog in each replicate in (J), the number of progeny per worm in (L) and the percentage of female *C. remanei* making sperm in each replicate in (S). The data are presented as the means ± SEMs. Statistics were performed by unpaired *t* test; three biological independent replicates. Source data are available online for this figure.

pathway reverses the TGSD to GGSD (Fig. 6K). Therefore, our findings provide insights into understanding the mechanism by which an animal with karyotype-dictated sex determination may gain the ability to respond to a temperature change to alter germline sex and thus enable the transition from GGSD to TGSD.

## Under warmer temperature treatment, the *C. elegans* hermaphrodites exhibiting TGSD make more progenies than the ones without TGSD

Environmental cue-induced sex reversal has been theorized as a potential mechanism that may increase the fitness of animals. It is well-known that high temperature impairs sperm activity in various animals, including *C. elegans* (Petrella, 2014), thus it is reasonable to postulate that increasing sperm production in the animals with TGSD may help to maintain reproductive fitness under high-temperature treatment. Taking into account the influence of temperature on the reproductive process and the limitation of hermaphrodite self-progeny to sperm quantity (Hodgkin and Barnes, 1991), we investigated whether DE90 hermaphrodite worms displaying Mog after warmer temperature treatment would exhibit a larger brood size compared to non-Mog individuals. Indeed, in DE90 worms with 30 °C treatment, hermaphrodites displaying Mog produced an obviously increased number of progenies compared to the non-Mog ones (Fig. 6L). Therefore, the TGSD mechanism enables the production of surplus sperm in hermaphrodites in response to elevated temperatures, which culminates in the production of a larger brood size, thereby enhancing reproductive fitness. Given that high temperatures impair sperm activity (Petrella, 2014), our findings support the notion that hermaphrodites obtain the ability of TGSD to increase sperm production for compensating the loss of sperm function caused by high temperature.

## The role of BiP in driving oocyte fate is conserved in a dioecious nematode species

Besides its function in hermaphrodites *C. elegans*, we investigated whether BiP might also modulate sex determination in a dioecious nematode species *C. remanei*, whose sex is determined by the XX(female)/XO (male) system. We attempted to generate an *hsp-4* mutation in *C. remanei* and we found that only *hsp-4(lf/+)* heterozygotes could be recovered, suggesting that *C. remanei hsp-4(lf)* might be largely lethal in homozygosity. Thus, we performed *hsp-4(RNAi)* by microinjecting corresponding dsRNA into the germline and the obtained F1 virgin females were evaluated for the potential presence of sperm. In contrast to that all of the RNAi control females produced only oocytes, the BiP knockdown females showed a reversal of female germline sex, as indicated by the production of sperm in a set of females (Fig. 6M–S). Spermatocytes were observed within the germline arms after *hsp-4* knockdown (Fig. 6P–R), which demonstrates that the sperm were made by these females themselves, rather than having been obtained via undesired mating. The relatively low percentage of *hsp-4(RNAi)*-treated *C. remanei* females exhibiting Mog may be because of only part of the animals with the BiP being knockdown to a proper level or the existence of other pathways redundant with the role of BiP in regulating *C. remanei* germline sex. Taken together, our data supports a possibly conserved role of BiP in modulating germline sex determination across species.

## Discussion

In this study, we aimed to identify the mechanisms underlying the transduction of temperature into a regulatory signal governing germline sexual fate and the molecular bases for the regulation of germline sex by both karyotype and environmental cues in the

same animal. Firstly, we discovered the amazing coexistence of GGSD and TGSD in several *C. elegans* strains. Further analyses uncovered that when *C. elegans* encounters a warmer temperature during the L3-L4 stage, the amount of available BiP is reduced likely due to the increased demand for BiP for protein folding in the ER, which leads to the ERAD-mediated degradation of TRA-2 and thereby translates warmer temperature into a sperm-fate-driving signal (summarized in Fig. EV5). This mechanism facilitates a temperature-induced effect on the karyotype-controlled germline sex determination pathway, thereby enabling the coexistence of GGSD and TGSD within the same animal. Strikingly, we could induce the animals to switch between GGSD and TGSD by manipulating the basal activity of the germline sex determination pathway, which sheds light on understanding how the transition and coexistence between GGSD and TGSD may occur during evolution. Moreover, our results reveal a role of BiP in promoting female germline fate in a dioecious nematode species, suggesting a possible conserved role of BiP in germline sex determination from hermaphrodites to male/female species. In summary, our findings identify BiP as a temperature sensor capable of perceiving and translating temperature into a signal for germline sex reversal and reveal the molecular mechanism underlying the coexistence of TGSD and GGSD in *C. elegans*, which represents the first mechanistic understanding of how temperature is transduced into a germline sex-determining signal and casts light on clarifying the mystery of how the karyotype and environmental cues co-regulate germline sex determination within the same organism.

Our study elucidates that the thermosensitive ER-resident chaperone BiP acts as a temperature sensor that mediates TGSD. It has been proposed that a protein that directly mediates the impact of temperature on the development of sex needs to both function in the sex determination pathway and display thermo-sensitivity so that different sexual fates can be induced at either end of the temperature range (Capel, 2017). BiP is consistent with this definition of a temperature sensor mediating the influence of temperature on germline sex determination. Specifically, at 20 °C, the amount of available BiP is relatively high and accordingly suppresses ERAD-mediated posttranslational repression of TRA-2 to permit the production of normal number of sperm; when worms encounter a warmer temperature, the amount of available BiP is reduced in response to the high-temperature-induced increase in the protein-folding burden in the ER, which leads to an increase in TRA-2 degradation by ERAD and thus promotes excess sperm production. Therefore, the thermosensitive BiP monitors the temperature change by sensing the consequent fluctuations in ER folding and then links temperature to germline sex reversal by interacting with the ERAD-mediated posttranslational mechanism that down-regulates the oocyte fate-driving factor TRA-2. Previous correlative-expression studies implied that heat shock response proteins as well as several other types of temperature-responsive proteins, including TRP channels, might link temperature to sex developmental signaling (Kohno et al, 2010; Weber et al, 2020; Yatsu et al, 2015), whereas whether the changes in the expression of these genes with temperature is a cause of sex determination or simply an accompanying process irrelevant to sex determination is unclear. In contrast, our study established a causal link between the change in heat shock response proteins, BiPs, and the alterations in the germline sex in response to temperature change. The involvement of HSPs that respond to relatively long duration of

warmer temperature, but not ion channels displaying a rapid response to temperature change, is consistent with the requirement of hours-long warmer temperature treatment to be able to induce germline sex reversal (Fig. 1A). By identifying the role of BiP in mediating TGSD, our research clarifies the mystery of how temperature is transduced into the signal determining sexual identity.

BiPs, belonging to the heat shock response HSP70 protein family, are expressed in all eukaryotes, which implies that these proteins may have the potential to act as conserved upstream temperature sensors in different species to mediate TGSD. Firstly, our study found that in addition to functioning in driving oocyte fate in *C. elegans* hermaphrodites, BiP is also required to promote oocyte fate in a male/female nematode species, supporting the role of BiP in modulating germline sex in both hermaphrodites and dioecious animals. Furthermore, it has been previously shown in fish (He et al, 2017), the expression of heat shock responses proteins (HSPs) is highly correlated with sex reversal (Kohno et al, 2010), also suggesting that these HSP proteins such as BiP may act as temperature sensors to mediate TGSD in these animals. Notably, studies in turtles have shown that $Ca^{2+}$ may affect STAT3 expression to mediate TSD, but the molecular mechanism that mediates the effect of temperature on $Ca^{2+}$ release to modulate sex needs further elucidation (Weber et al, 2020). Since BiP has been shown to affect $Ca^{2+}$ release from the ER (Preissler et al, 2020), it will be interesting to investigate whether BiP might also mediate the impact of temperature on sex determination in these reptiles. It is possible that the specific molecular mechanism by which BiP modulates sex determination could differ across species. For instance, in *C. elegans*, BiP may impact TRA-2 levels, whereas in turtles, it could potentially activate $Ca^{2+}$ release or other molecular events to direct sex determination. Therefore, BiP may play a conserved role in transducing temperature into a signal-modifying sexual identity across species.

Our research represents one of the very few studies that provide experimental evidences to illustrate the physiological significance of temperature-induced sexual identity change (Warner and Shine, 2008). Sex reversal in response to alterations in environmental cues has long been hypothesized as a strategy to enhance lifetime reproductive success (Bull, 1981; Crews and Bull, 2009). Theoretically, for hermaphrodites whose sex is responsive to temperature changes, they could have at least two mechanisms to enhance the reproductive fitness. One strategy is to alter the ratio of hermaphrodites to males in their offspring in order to increase genetic diversity through mating. Another strategy is for hermaphrodites to directly produce more sperm, resulting in an increased number of offspring. This latter strategy was demonstrated in this study, and we found that in *C. elegans* hermaphrodites, in which the number of self-progeny is limited by the number of sperm, TGSD mechanism enables animals to elevate sperm production in response to warmer temperature (30 °C) and, thus, to increase brood size. Therefore, our study supports the notion that in hermaphrodites, environmental cues such as temperature modulate only germline sex, which is sufficient to modulate the brood size through altering the number of sperm and oocytes, in contrast to dioecious species such as reptiles in which TSD allows the transformation of sex-related characteristics in both the soma and germline to produce the sex that shows better fitness at the corresponding temperature. Therefore, animals with distinct

reproductive modes, such as hermaphroditic and dioecious organisms, exhibit the ability to modify their germline or overall body sex in response to fluctuations in temperature, further underscoring the remarkable diversity of sex determination mechanisms. It is worth mentioning that the warmer temperature treatment utilized in this study, set at 30 °C, was conducted over a period of several hours and 30 °C also falls within the range of temperatures found in natural habitats where at least some nematodes can reside, suggesting that the TGSD in *C. elegans* observed in this study may also be present in the natural environment. We have not been able to demonstrate the advantages that the increased offspring number might confer for the population under a high temperature in this study, which will be highly intriguing to investigate in the future for understanding the importance of TSD from an evolutionary perspective.

Interestingly, in contrast to the TSP of reptiles, which occurs in the embryonic stage (Ge et al, 2018), *C. elegans* is sensitive to the effect of temperature at the L3-L4 stage, when its germ cells switch from adopting a sperm fate to adopting an oocyte fate. The embryonic-stage TSP of reptiles is consistent with sex reversal throughout the whole organism. In contrast, the sex change observed in *C. elegans* hermaphrodites in response to warmer temperature occurs only in the bipotential germline, and it is thus reasonable for the worms to respond to temperature at this later developmental stage. Similar to *C. elegans*, many fish can also undergo sex alterations at a late developmental stage, even during adulthood, as an adaptive response to environmental changes (Kobayashi et al, 2013). Furthermore, our findings provide evidence account for the limitation of *C. elegans* sex reversal to the germline in response to temperature change. Specifically, in *C. elegans* hermaphrodites, the much lower expression of *tra-2* in the germline than in the soma may confer the sensitivity of the germline to the warmer temperature-induced global reduction in the TRA-2 level such that the threshold for driving male sex is surpassed only in the germline. This mechanism thus permits germline-specific sex alteration to increase sperm production without causing defects in somatic sexual development that might compromise the production of more progeny. Therefore, animals with different sex types, such as hermaphrodites and dioecious animals, may adopt distinct sex-reversal strategies (germline-specific or global sex reversal) to respond to fluctuations in environmental cues to enhance reproductive fitness.

Interestingly, our data support that treatment with 30 °C, but not 25 °C, 28 °C, and 32 °C, significantly induces masculinization of the germline. Notably, previous studies have indicated that *C. elegans* possesses the ability to detect temperature changes as small as 0.1 °C (Ramot et al, 2008). This precise temperature monitoring suggests that animals may employ specific coping strategies for specific levels of temperature. For instance, in response to a warmer temperature of 30 °C, *C. elegans* utilizes a regulated germline sex alteration strategy to enhance reproductive fitness. This change of germline sex is not observed at other warmer temperatures, implying that worms may have evolved alternative strategies to cope with these levels of temperature. In the future, the molecular mechanisms underlying this specific temperature (30 °C) sensitivity, as well as its physiological significance, require further investigation from an ecological perspective.

Our study provides the first mechanistic understanding of how GGSD and TGSD can switch and coexist within the same organism.

The remarkable phenomenon of switching between sex determination modes and the coexistence of GSD and TSD have been observed in various organisms (Capel, 2017; Chen et al, 2014; Ezaz et al, 2006; Holleley et al, 2015), but the underlying mechanisms remain enigmatic. Our study contributes to solving this puzzle by revealing that the temperature sensor BiP can transduce the temperature cues into TRA-2 level and thereby overrides the karyotype to modulate the sex determination pathway (Ellis and Schedl, 2007). Therefore, the effects of both temperature and karyotype converge on the germline sex determination pathway to co-regulate germ cell fate (Fig. EV5). Remarkably, by taking advantage of this newly discovered TGSD mechanism, we have been able to switch sex determination modes between GGSD and TGSD through manipulating the basal activity of the sex determination pathway by introducing proper mutations. Moreover, compared to the N2 that does not acquire TGSD, we observed a significantly higher basal *fog-3* transcription in the DE90 worms at 20 °C. This observation supports the idea that making the germline sex determination pathway activity easier to approach the threshold for adopting a sperm fate enables the worms to acquire TGSD when exposed to elevated temperatures. Therefore, it is possible that during evolution, the accumulation of mutations that promote the basal activity of the sex determination pathway to approach the threshold for driving the corresponding sex fate may favor the acquisition of TSD or ESD in animals.

## Methods

### Worm strains and maintenance

The following wild-isolated *C. elegans* strains AB3, CB4852, CB4854, CB4855, CB4856, CX11285, ED3011, ED3017, ED3052, EG4725, and JT11398 were obtained from the Caenorhabditis Genetics Center (CGC). The domesticated wild-type N2 strain and the following transgenic or mutant *C. elegans* strains were also obtained from the CGC: DE90 *[oxIs318 (spe-11p::mCherry::histone);ruIs32 (pie-1p::GFP::histone H2B);ddIs6 (tbg-1::GFP); dnIs17 (pie-1p::GFP::hPLCIII-delta PH757 domain)]*, VP303 *[rde-1(ne219) V; kbIs7 [nhx-2p::rde-1+rol-6(su1006)]]*, SJ4005 *[zcIs4[hsp-4p::GFP]*, VC1369 *[rpn-10(ok1865) I]*, FX544 *[cdc-48.1(tm544) II]*, FX659 *[cdc-48.2(tm659)]*, RB1104 *[hsp-3(ok1083) X]*, VC1099 *[hsp-4(gk514)]*, CB3844 *[fem-3(e2006) IV]*, CB3778 *[tra-2(e2020) II]* and JK4871 *[fog-3(q520) I;qSi41 II]*, SJ17 *[(zc12) III; zcIs4 (hsp-4p::GFP)]*. The following *C. elegans* strains were generated by SunyBiotech (China): PHX1874 *[hsp-4[syb1874(HSP-4::V5::3× HA)]*, PHX1888 *[hsp-3[syb1888(HSP-3::3×HA::6×His)]]*, PHX1960 *[hsp-3[syb1960(HSP-3:: degron::wrmScarlet)]]*, PHX1966 *[hsp-4 [syb1966(HSP-4:: degron::wrmScarlet)]]*, PHX6709 *[sybSi6709 (hsp-16.41p::HSP-4::Flag-HA-tbb-2 3'UTR;oxIs318 II;unc-119(ed3 or e2498) ruIs32 III;ddIs6 V;dnIs17)]*, PHX1459 *[sybSi36(spe-11p::his-58::mCherry); unc-119(ed3)]*, and PHX1009 *[tra-2 [syb1009(TRA-2::GFP::3×Flag)]]*. The MAT115 *[fog-3(q520)I/hT2]* strain was derived from JK4871. The following *C. elegans* strains were established in our laboratory: MAT142 *[tra-2(jef37)/mIn1 [mIs14 dpy-10(e128)]; sybSi36(spe-11p::his-58::mCherry) I]*, MAT145 *[hsp-4[syb1874(HSP-4::V5::3×HA)]; hsp-3[syb1888(HSP-3::3×HA::6× His)]]*, MAT147 *[mkcSi13 [sun-1p::rde-1::sun-1 3'UTR + unc-119(+)]; rde-1(mkc36) V;sybSi36(spe-11p::his-58::mCherry)I]*, MAT148 *[rpn-10*

(ok1865)I;sybSi36(spe-11p::his-58::mcherry)I], MAT149 [cdc-48.1 (jef39);oxIs318 II;unc-119(ed3 or e2498) ruIs32 III;ddIs6 V;dnIs17], MAT150 [rpn-10(jef40);oxIs318 II;unc-119(ed3 or e2498) ruIs32 III;ddIs6 V;dnIs17], MAT151 [cdc-48.1(jef39);tra-2[syb1009(TRA-2::GFP::3×Flag)]], MAT152 [cdc-48.1(tm544)II;sybSi36(spe-11p::his-58::mCherry) I], MAT153 [rpn-10(ok1865) I;tra-2[syb1009(TRA-2::GFP::3XFlag)]II], MAT191 [tra-2(jef43);oxIs318 II;unc-119(ed3 or e2498) ruIs32 III;ddIs6 V;dnIs17] and MAT192 [rpn-10(ok1865) I;tra-2(jef37)/mIn1[mIs14 dpy-10(e128)]; sybSi36(spe-11p::his-58::mCherry) I]. The MAT204 [hsp-4(gk514);[sybSi36(spe-11p::his-58::mCherry)]] strain was obtained from a cross between PHX1459 and VC1099. The DCL569 [mkcSi13 [sun-1p::rde-1::sun-1 3'UTR + unc-119(+)] II;rde-1(mkc36) V] strain, used for knocking down genes specifically in the *C. elegans* germline (Zou et al, 2019), was kindly provided by Di Chen's laboratory. The wild-type *C. remanei* SB146 strain was purchased from the CGC. Worms were maintained on nematode growth medium (NGM) spotted with OP50 bacteria at 20 °C unless otherwise indicated.

## Construction of transgenic and mutant strains

The integrated transgenic and knock-in strains were established by using the CRISPR–Cas9 method described below. Specifically, for the overexpression of HSP-4 at the warmer temperature, the PHX6709 strain [sybSi6709 (hsp-16.41p::HSP-4::Flag::HA::tbb-2 3'UTR)] was generated by inserting the hsp-16.41p::HSP-4::Flag::HA::tbb-2 3'UTR sequence into the locus immediately downstream of the 3'UTR of srm-1 at the distal end of chromosome III via a highly efficient sgRNA target site in the DE90 strain. To evaluate the expression of endogenous tra-2, we generated the PHX1009 [tra-2[syb1009(TRA-2::GFP::3×Flag)]] strain, in which the GFP::3×Flag sequence was targeted to the locus immediately upstream of the stop codon of the endogenous tra-2 gene. To detect the expression of endogenous BiP, the 3×HA::6×His and V5::3×HA sequences were inserted into the sequence immediately upstream of the ER retention signal HDEL in the endogenous hsp-3 and hsp-4 genes, respectively, to generate the corresponding strains PHX1888 [hsp-3[syb1888(HSP-3::3×HA::6×His)]] and PHX1874 [hsp-4[syb1874(HSP-4::V5::3×HA)]. Moreover, to explore whether hsp-4(RNAi) can simultaneously knock down HSP-3 and HSP-4 in worms, we generated the PHX1960 [hsp-3[syb1960(HSP-3::degron::wrmScarlet)]] and PHX1966 [hsp-4[syb1966(HSP-4::degron::wrmScarlet)]] strains, in which the degron::wrmScarlet sequence was inserted within the sequence immediately upstream of the HDEL-encoding sequence in the endogenous hsp-3 and hsp-4 genes, respectively.

Moreover, the CRISPR–Cas9 method was used to generate the MAT142, MAT149 and MAT151 mutant strains by introducing a premature stop codon. Specifically, the corresponding sgRNA was designed online (https://benchling.com) and was synthesized by using the HiScribe™ T7 RNA Synthesis Kit (NEB, E2050). The purified RNA and Cas9 protein (Cat1081058, IDT) were mixed at 37 °C for 10 min to assemble the RNP complex. Then, the donor DNA and plasmids expressing the coinjection marker were added to the RNP complex, and this mixture was subsequently injected into the indicated worms. The F1 worms expressing the coinjection marker were isolated and genotyped by sequencing. The MAT149 [cdc-48.1(jef39);oxIs318 II;unc-119(ed3 or e2498) ruIs32 III;ddIs6 V;dnIs17] and MAT150 [rpn-10(jef40);oxIs318 II;unc-119(ed3 or

e2498) ruIs32 III;ddIs6 V;dnIs17] strains were generated by introducing premature stop codons at the N-termini of CDC-48.1 and RPN-10 in the DE90 strain, respectively. The MAT151 [cdc-48.1(jef39);tra-2[syb1009(TRA-2::GFP::3×Flag)]] strain was generated by introducing a premature stop codon at the N-terminus of CDC-48.1 in the PHX1009 [tra-2[syb1009(TRA-2::GFP::3×Flag)]] strain. The MAT142 [tra-2(jef37)/mIn1[mIs14 dpy-10(e128)];syb-Si36(spe-11p::his-58::mCherry) I] strain was obtained by introducing a stop codon at the N-terminus of TRA-2 in the PHX1459 [sybSiIs36(spe-11p::his-58::mCherry);unc-119(ed3)] strain and was balanced by mIn1. MAT191 [tra-2(jef43);oxIs318 II;unc-119(ed3 or e2498) ruIs32 III;ddIs6 V;dnIs17] was generated by deleting 108 bp of the 3'UTR of tra-2 in DE90 background with CRISPR–Cas9 method, completely mimicking tra-2(e2020, gf) mutation (Doniach, 1986).

## Evaluating sperm number in *C. elegans* hermaphrodites with warmer temperature treatment

For exploring the effect of warmer temperature treatment on the germline sex of wild-type worms, hermaphrodites of the indicated wild-isolated strains, the domesticated wild-type N2 and the N2-derived DE90 were cultured at 20 °C for 27 h post-hatching. They were then transferred to 30 °C and incubated for 12 h before being shifted back to a temperature of 20 °C. When oogenesis just initiated, these wild-type hermaphrodites were fixed with 100% methanol for 5 min at −20 °C and were then stained with DAPI to visualize sperm by microscopic analyses, as previously described (Tang and Han, 2017). The DE90 strain, which expresses the spe-11p::mCherry::histone transgene to specifically label the nuclei of spermatids and spermatocytes (Merritt et al, 2008), was collected to perform microscopic analyses to count spe-11p::mCherry::histone-labeled sperm.

To count sperm number, ~30–100 Z-stack images (0.5 μm per slice) of the germline were acquired by using spinning-disk confocal microscopy (Nikon) (30–50% of the laser power, 100–300 ms exposure time for imaging DAPI; 100% of the laser power, 300–500 ms exposure time for imaging spe-11p::mCherry::histone). The sperm number was quantified with Imaris 9.3 software by 3D reconstruction of the germline by setting the size of the dots that simulate the nuclei at 1.8 μm. As long as the images of a germline arm were clear, they were used for quantifying the sperm. It is worth mentioning that it could sometimes be challenging to obtain clear images of both germline arms in the same worm due to autofluorescence signal interference from the intestine. Consequently, there are instances where only one germline arm is counted for certain animals. Nonetheless, this does not affect our results, as "n" represents the number of scored germline arms. As previously described (Tang and Han, 2017), when the average number of sperm in the hermaphrodites was greater than that in the control hermaphrodites, then the corresponding animal was scored as showing Mog.

## RNAi by feeding and microinjection

To identify the genetic factors that mediate the impact of temperature on germline sex determination, RNAi feeding experiments targeting 82 genes that were previously shown to be responsive to temperature (Coburn et al, 1998; Jovic et al, 2017;

Weber et al, 2020) were performed (Table 2). Specifically, the bacterial strains for the RNAi knockdown of the target genes were selected from the ORF RNAi (GE Dharmacon) and MRC RNAi (Source BioScience) libraries and cultured on Luria–Bertani (LB) plates containing ampicillin. Then, single colonies of these bacteria were inoculated into LB liquid medium with ampicillin (50 μg/ml) and tetracycline (20 μg/ml) and grown for 12 h. The bacteria in this liquid medium were subsequently spotted onto NGM plates containing IPTG (2 mM) and ampicillin (100 μg/ml) and incubated for 3–5 days at room temperature (RT) to induce the expression of dsRNA before seeding worms. Approximately 5 L1-stage DE90 hermaphrodites were seeded onto each RNAi plate, and the Mog phenotype was evaluated in the F2 worms at the late L4 stage. Hermaphrodites that show Mog still produce sperm at the turn of the germline arm at the late L4 stage, when oocytes have already been produced in the worms with the wild-type phenotype. To facilitate screening, the Mog phenotype was first evaluated based on the ectopic presence of sperm at the turn of the germline arm in late L4-stage hermaphrodites, and these worms displaying ectopic production of sperm were subjected to further rigid counting of the sperm to confirm the excess sperm production. The presence of Fog phenotype was evaluated by the significant reductions or absence of sperm in the germline. Worms fed with HT115 bacteria containing the empty PL4440 vector served as the negative control. In Fig. 5, two generations of *hsp-4* RNAi by feeding was performed to enhance RNAi efficiency. Specifically, L1-staged worms with indicated genotypes were separately seeded onto PL4440 vector control and *hsp-4*(RNAi) plates. Subsequently, the L1-staged F1 worms from those plates were transferred to PL4440 vector control, PL4440 vector control:*hsp-4*(RNAi) (mixed at 1:1 ratio), and *cul-2*(RNAi):*hsp-4*(RNAi) (mixed at 1:1 ratio) plates to grow to Late L4 stage. Finally, the fluorescent signal of TRA-2::GFP and Mog phenotype in these worms were scored using spinning-disk confocal microscopy (Nikon).

*hsp-4(RNAi)* by microinjection was performed as previously described (Tang and Han, 2017). Specifically, the DNA template for *hsp-4* dsRNA synthesis was amplified from *hsp-4(RNAi)* bacteria from the ORF RNAi library by using primers corresponding to the T7 promoter. The *hsp-4* dsRNA was synthesized in vitro with a HiScribe™ T7 RNA Synthesis Kit (NEB, E2050). Subsequently, 1 μg/μl dsRNA was injected into the germline of the mother worm. The progeny produced between 8 and 24 h after injection were grown to the late L4 stage, and the Mog phenotype was evaluated when the first oocyte appeared in the germline.

To evaluate the role of *hsp-4* in germline sex determination in *C. remanei*, knockdown of *hsp-4* was performed by injecting the corresponding dsRNA into the germlines of conceived females. The progeny produced at 8–24 h after injection were grown to the L3 stage, and then, the females were picked to transfer to a new plate to prevent mating. When these virgin female F1 worms reached adulthood, the Mog phenotype was evaluated by determining the presence of sperm by DAPI staining as described above (Tang and Han, 2017).

## Analyses of the fluorescent signal of the transgenic worms

To analyze the native TRA-2::GFP level in the germline and intestine, hermaphrodites expressing *KI [TRA-2::GFP::3×Flag]* with

knock-in of GFP::3×Flag fused to the C-terminus of native TRA-2 were dissected and then fixed with 4% formaldehyde for 5–10 min, followed by three times wash with M9 buffer containing 0.1% Tween-20. Following this, the samples were stained with DAPI for 10 min and placed on a 2% agarose pad for microscopy analysis. The TRA-2::GFP::3×Flag and DAPI signals were imaged by using the Nikon spinning-disk confocal microscope. The intensity analyses of TRA-2::GFP::3×Flag fluorescence as well as the sperm counting based on the DAPI signals were performed by using Imaris 9.3 software.

To analyze the fluorescence of the *Is[hps-4p::GFP]*, *KI [HSP-3::wrmScarlet]* and *KI [HSP-4::wrmScarlet]* animals with indicated treatment, worms were transferred to 5–10 μl of M9 buffer containing levamisole on a standard NGM plate and then analyzed under a dissecting fluorescent microscope. Quantitative analysis of fluorescent images was performed using Imaris software.

## Expressing TRA-2 in Sf9 insect cells

Worm TRA-2 was expressed in the *Spodoptera frugiperda* (Sf9) cells (CRL-1711™) to analyze its localization on the plasma membrane and in the perinuclear region. To generate the *C. elegans* TRA-2::mCherry recombinant Baculovirus, the DNA fragment of TRA-2::mCherry was integrated into the pFastBac vector. The resulting plasmids were then transferred into DH10Bac-competent *E. coli* cells to isolate the recombinant Bacmid DNA, which served as the basis for generating the recombinant Baculovirus. Next, the Baculoviral supernatant, which was used to express TRA-2::mCherry, was employed to infect Sf9 cells cultured in Sf-900TM II SFM medium (Gibco) and at 27 °C. Seventy-two hours post infection, the Sf9 cells were fixed using a 4% paraformaldehyde solution for 10 min. Subsequently, the cells were washed with PBS and permeabilized using 0.2% Triton X-100. After permeabilization, the cells were incubated with a DAPI solution, washed again, and then mounted for microscopic imaging using a Zeiss microscope. Before conducting these assays, the cells were authenticated and tested to ensure that they did not contain any mycoplasma contamination.

## Coimmunoprecipitation and immunoblotting

To detect the interaction between TRA-2 and HSP-4, HEK293 cells (ATCC®CRL-3216™) co-transfected with plasmids expressing TRA-2::Flag::HA or Flag::FEM-3 and HSP-4::GFP were used to perform coimmunoprecipitation assays. Specifically, these cells were harvested and lysed in lysis buffer (150 mM NaCl, 50 mM Tris (pH 7.4), 0.5% Triton X-100) containing protease inhibitors at 4 °C for 20 min. After the lysates were centrifuged at $14{,}000 \times g$ for 15 min at 4 °C, the lysate supernatants were collected and then incubated with GFP-Trap magnetic beads (gtma-20, ChromoTek) with rotation at 4 °C overnight. After five washes, proteins were eluted from the GFP-Trap beads with glycine under acidic conditions. Then, these eluted proteins were analyzed by western blotting with the primary antibody anti-Flag (1:3000, M8823, sigma) and anti-GFP (1:1000, A-11122, Invitrogen) to detect TRA-2::Flag::HA, Flag::FEM-3, and HSP-4::GFP, respectively. The bands were imaged with an Amersham Imager 680 system (AI 680, GE). For the quantification of expression, band intensity was analyzed with ImageJ.

To perform P97 and TRA-2 interaction analysis, HEK293 cells co-expressing TRA-2::GFP::Flag and P97::GFP::HA::6×His with/without MG132 (10 μM) treatment for 8 h were pelleted and lysed in cell lysis buffer at 4 °C for 20 min, and the supernatants were then incubated with GFP-Trap beads to pull down the P97-interacting proteins, and after five washes, the proteins eluted from the beads were analyzed by western blotting with an anti-Flag antibody to detect TRA-2. Before conducting these assays, the cells were authenticated and tested to ensure that they did not contain any mycoplasma contamination.

To investigate the effect of warmer temperature on the amount of available BiP in *C. elegans*, crosslinking protein analysis was performed. Specifically, *KI [HSP-3::3×HA::6×His];KI [HSP-4::V5::3×HA] C. elegans* worms with indicated treatment were pelleted, resuspended in lysis buffer (20 mM HEPES, 150 mM NaCl, 0.5% Triton X-100; pH 7.5) containing protease inhibitors and then homogenized by using a Fastprep-24™ 5 G homogenizer (MP Biomedicals). Following centrifugation, the lysate supernatant was collected, and the protein concentration was determined by a BCA assay kit (Cat E162-101, GenStar). To perform the chemical crosslinking reaction, the cell lysates were incubated with 4 mM DSS for 1 h at room temperature. Subsequently, the crosslinking reaction was quenched by incubating the lysates in 100 mM Tris-HCl (pH 7.4) for 20 min. The crosslinked cell lysate was then mixed with prewashed anti-HA magnetic beads (Cat 88836, Thermo Fisher) to specifically enrich the HA-tagged BiPs, followed by incubation with rotation for 8 h at 4 °C. After rinsing in washing buffer for five times, these beads were boiled in SDS loading buffer, and the supernatants were used for immunoblot analysis of HA-tagged BiPs level with the primary antibody anti-HA (1:3000, HOA012HA01, AbHO).

To analyze the TRA-2 levels in the whole worm, the *KI[TRA-2::GFP::3×Flag]* worms with indicated treatment were subjected to western blot analyses. Specifically, GFP-Trap magnetic beads were used to enrich TRA-2::GFP::3×Flag from worm lysates, and the beads were subsequently resuspended and boiled in SDS loading buffer for 5 min. The endogenous TRA-2 level was measured by immunoblotting TRA-2::GFP::3×Flag with an anti-Flag antibody.

### Analyses of brood size of hermaphrodites with warmer temperature treatment

Following the treatment of warmer temperatures, the sperm in live DE90 worms, whose sperm were labeled with the *Is[spe-11p::mCherry::histone]* transgene, were counted using a fluorescent microscope to assess whether the worms displayed the Mog or non-Mog phenotype. Subsequently, both the Mog and non-Mog worms from the agarose pad were carefully transferred to NGM plates with one hermaphrodite per dish, ensuring minimal harm to the organisms throughout the process. Then, the progenies from each of these hermaphrodites were counted until no further offspring were produced. To facilitate accurate counting, each mother worm was transferred to a new NGM plate every 24 h. The mean value of the progeny counted from the hermaphrodites with indicated phenotypes was determined by dividing the total number of progenies produced by all worms with the indicated phenotype (Mog or non-Mog) with the number of worms displaying the corresponding phenotype.

### qPCR analysis

The transcription of *fog-3* and *tra-2* were analyzed by qPCR as previously described (Tang and Han, 2017). Specifically, young adult worms with indicated genotypes and treatments were collected and mRNA was then extracted from these worms using TRIzol. Reverse transcription was performed with FastKing gDNA Dispelling RT SuperMix (KR118, TIANGEN). qPCR analysis was conducted using Jena Qtower3G (Analytik Jena). As previously indicated, *rpl-26*, which encodes the large ribosomal subunit L26 protein, was used as the internal control (Cui et al, 2006). The *fog-3* qPCR primers, *fog-3*-F: ATGTA-TACCGAAGTCCGCGAGC and *fog-3*-R: GAACATCCCAGGTA-GACGAGAA, were adopted from a previous study (Chen and Ellis, 2000). The *tra-2* qPCR primers, *tra-2*-F: TTATTGGTC GTTGGGACGCA and *tra-2*-R: TGATGCCAACTCAGGGGTTC, were utilized.

To analyze *hsp-4* transcription in the germline, L4-staged worms with the indicated treatment were carefully and rapidly dissected. Approximately 20 intact germline arms were then collected by suction with a pipette and transferred to an Eppendorf tube, as confirmed by microscopic analysis. The mRNA from these germline arms was extracted using the E.Z.N.A.® MicroElute® Total RNA Kit (R6831-02, Omega), which is suitable for isolating total RNA from a small amount of tissue. Subsequently, reverse transcription and qPCR analysis were conducted following the aforementioned protocol. The *hsp-4* qPCR primers, *hsp-4*-F: GCCGTTCAAGATCGTCGACA and *hsp-4*-R: CGTTCTTCA CCTCGTGACCA, were used.

### Statistical analysis

Before analyzing Mog with a complex microscope, worms were collected by using a dissecting microscope to ensure that worms with desired phenotypes in the germline were not preferentially chosen, thereby randomizing the samples. No samples were excluded from the analysis. To avoid potential bias, both the treatment and genotype information were masked during sample evaluation. To ensure robustness and reliability, a minimum of three independent experiments were performed, and the data are presented as means ± standard errors of the mean (SEMs) to illustrate the level of variation. The number showing in the bar graph represents the number of germline arms, cells or worms used for the indicated analysis. The unpaired $t$ test and one-way ANOVA was carried out with GraphPad Prism to evaluate statistical significance.

## Data availability

This study includes no data deposited in external repositories. The source data of this paper are collected in the following database record: https://www.ebi.ac.uk/biostudies/studies/S-BSST1466.

The source data of this paper are collected in the following database record: biostudies:S-SCDT-10_1038-S44318-024-00197-z.

# Peer review information

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

## Acknowledgements

The authors thank the CGC (funded by NIH [P40OD010440]) for strains, Di Chen's laboratory (Zhejiang University) for providing strain DCL569, and Enzhi Shen's laboratory (Westlake University) for providing technical supports and reagents, the Microscopy Core facilities in Westlake University for support, Y. Wang, Y. Gao, F. Xiao, G. Fang, and Z. Yu for technical support, Lianfeng Wu for comments and all the members of the Tang laboratory for comments and suggestions. This research was supported by National Natural Science Foundation of China (No. 32070565, No. 32350015 and No. 31871465), HRHI program (No. 202109007 and No. 202209003) of Westlake Laboratory of Life Sciences and Biomedicine, Zhejiang Provincial Natural Science Foundation of China (No. LQ23C040002 and XHD24C0701), National Key Research and Development Program of China (No. 2019YFA0802900), the Westlake Education Foundation of Westlake University and Zhejiang Provincial Key Laboratory Construction Project.

## Author contributions

**Jing Shi**: Conceptualization; Resources; Data curation; Software; Formal analysis; Validation; Investigation; Visualization; Methodology; Writing—original draft; Project administration; Writing—review and editing.
**Danli Sheng**: Conceptualization; Resources; Data curation; Software; Formal analysis; Validation; Investigation; Visualization; Methodology; Writing—original draft; Project administration; Writing—review and editing. **Jie Guo**: Resources; Data curation; Software; Formal analysis; Validation; Investigation; Visualization; Methodology; Writing—original draft; Project administration; Writing—review and editing. **Fangyuan Zhou**: Resources; Data curation; Software; Formal analysis; Supervision; Validation; Investigation; Visualization; Methodology; Project administration; Writing—review and editing. **Shaofeng Wu**: Resources; Data curation; Software; Formal analysis; Validation; Investigation; Visualization; Methodology; Project administration; Writing—review and editing. **Hongyun Tang**: Conceptualization; Resources; Data curation; Software; Formal analysis; Supervision; Funding acquisition; Validation; Investigation; Visualization; Methodology; Writing—original draft; Project administration; Writing—review and editing.

Source data underlying figure panels in this paper may have individual authorship assigned. Where available, figure panel/source data authorship is listed in the following database record: biostudies:S-SCDT-10_1038-S44318-024-00197-z.

## Disclosure and competing interests statement

The authors declare no competing interests.

# Expanded View Figures

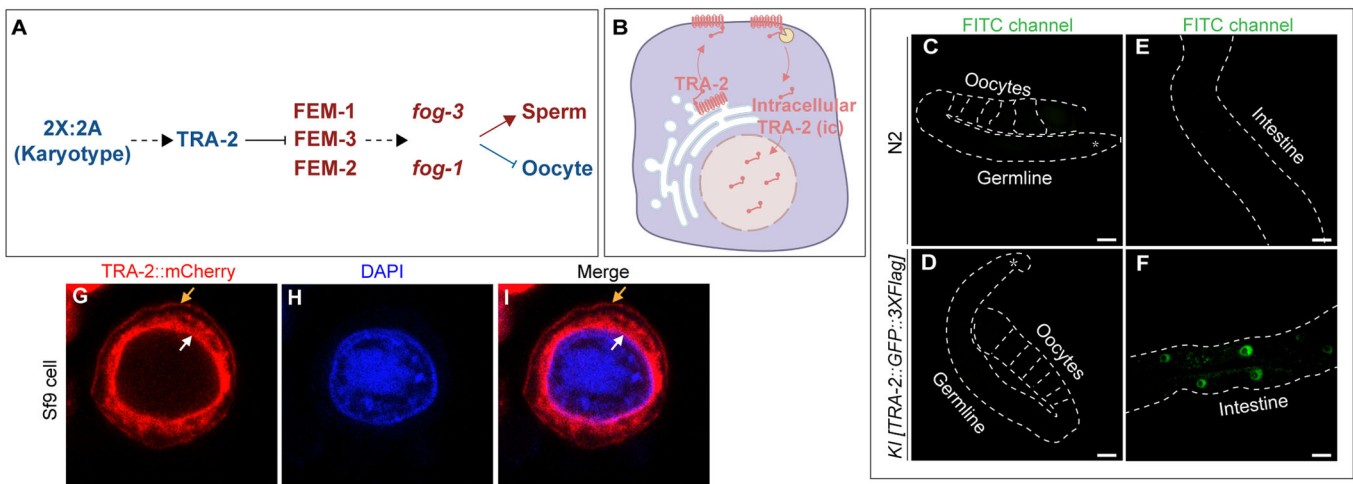

**Figure EV1. The analysis of the TRA-2 expression pattern and subcellular localization.**

(A) Model of the karyotype-mediated germline sex determination pathway in *C. elegans*, adopted from previous studies (Ellis and Schedl, 2007). The karyotype (2X:2A; ratio of X chromosomes to autosomes in *C. elegans* hermaphrodites) is the primary signal that modulates germline sex by regulating *tra-2*, which encodes a transmembrane protein functioning upstream of the FEM proteins. The relative ratio of TRA-2 to FEM-3 expression is critical for determining germline sex. In response to the upstream signal, the transcription of *fog-3*, encoding a member of the Tob protein family, is regulated to modulate germ cell fate. The genes in red promote sperm fate and the genes in blue drive oocyte fate. (B) A diagram illustrating the cellular behavior of TRA-2, summarized from previous studies (Mapes et al, 2010b; Shimada et al, 2006; Sokol and Kuwabara, 2000). The full-length transmembrane TRA-2 is synthesized and folded at the ER membrane. It is then trafficked to the plasma membrane, where it is cleaved. This cleavage releases a short intracellular C-terminal fragment that can translocate to the nucleus. (C–F) Representative images showing the expression pattern of the native-expressed TRA-2::GFP. The *KI [TRA-2::GFP::3×Flag]* strain, where the *GFP::3×Flag* was knocked in to the C-terminus of native *tra-2*, was used to determine the expression pattern of *tra-2*. GFP fluorescence can be observed in the nuclei of intestinal cells (F). In contrast, the TRA-2::GFP signal was not visible in the germline (D). The asterisk (*) indicates the distal end of the germline and white dashed lines outlined the germline arms and intestine. N2 served as a control to exclude interference from autofluorescence. *n* = 60 worms. Scale bar: 20 μm. (G–I) Micrographs showing the perinuclear and cell peripheral TRA-2 in Sf9 cells. The worm TRA-2 fused with mCherry was expressed in Sf9 insect cells to determine the subcellular localization of TRA-2. The white arrow indicates the TRA-2 signal in the perinuclear region, and the orange arrow indicates the plasma membrane-localized TRA-2. *n* = 30 cells were evaluated. Scale bar: 3 μm.

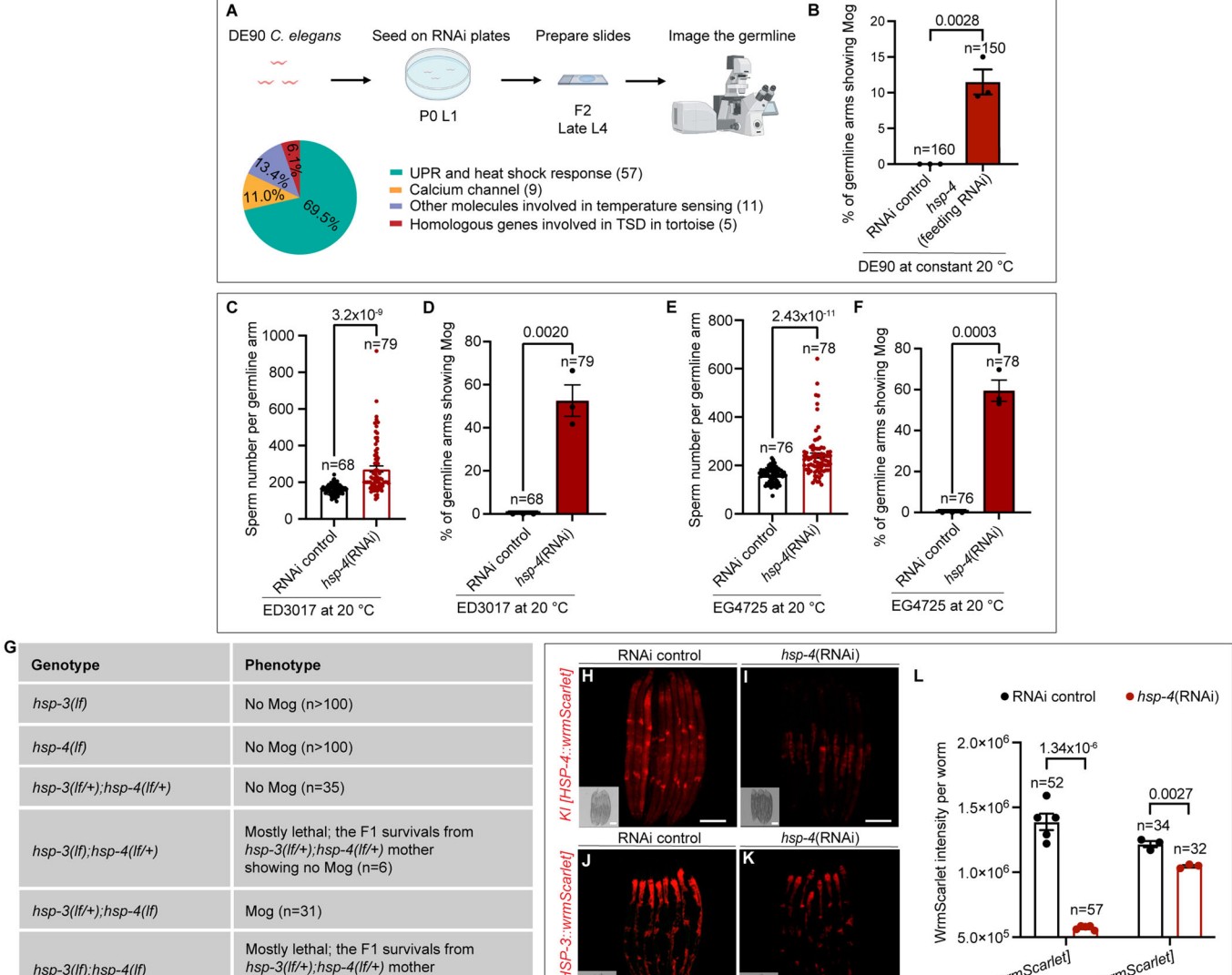

**Figure EV2. Uncovering the role of BiP in germ cell fate determination through a RNAi screen and the redundant role of HSP-3 and HSP-4 BiPs in driving female germline sex.**

(A) Workflow of the RNAi screen for identifying the potential thermosensitive genes that regulate germline sex, and a pie chart showing the categories of the genes included in the screen. The RNAi screen was performed by feeding at a constant 20 °C using DE90 strain. (B) Graph showing that in the screen described in (A), *hsp-4(RNAi)* by feeding caused Mog in a subset of hermaphrodites. The DE90 strain receiving *hsp-4(RNAi)* treatment was analyzed to determine the percentage of germline arms showing Mog. P value = 0.0028. (C–F) Bar graphs showing strong Mog caused by *hsp-4(RNAi)* in the wild-isolated *C. elegans* ED3017 and EG4725 strains cultured at a constant 20 °C. The *hsp-4(RNAi)* experiment was performed by microinjection with corresponding dsRNA. The quantitative analyses of the sperm number are shown in (C, E). With *hsp-4(RNAi)* treatment, 52.6% and 59.5% of germline arms displayed Mog in ED3017 (D) and EG4725 (F), respectively. P value = $3.2 \times 10^{-9}$ (C), P value = 0.0020 (D), P value = $2.43 \times 10^{-11}$ (E), P value = 0.0003 (F). (G) A table showing that the HSP-3 and HSP-4 BiPs function redundantly to regulate germline sex determination. The progeny of *hsp-3(lf/+);hsp-4(lf/+)* worms were scored to analyze Mog in the *hsp-3(lf/+);hsp-4(lf/+)*, *hsp-3(lf/+);hsp-4(lf)*, *hsp-3(lf);hsp-4(lf/+)* and *hsp-3(lf);hsp-4(lf)* worms. "*n*" indicates the number of worms evaluated. (H–L) Micrographs and a corresponding bar graph indicating that *hsp-4(RNAi)* can knock down both *hsp-3* and *hsp-4*. *hsp-4(RNAi)* was performed by the feeding method. The fluorescence signals of the native HSP-4::wrmScarlet and native HSP-3::wrmScarlet proteins in worms were analyzed after control(RNAi) or *hsp-4(RNAi)* treatment. The quantification of the mean fluorescence intensity of HSP-4::wrmScarlet in (H, I) and HSP-3::wrmScarlet per worm in (J, K) is shown in (L). P value = $1.34 \times 10^{-6}$, P value = 0.0027 in (L) (from left to right). Scale bar:100 μm. For the statistics in this Figure, the data are presented as the means ± SEMs. Each dot represents the sperm number in one germline arm in (C, E), the percentage of germline showing Mog each replicate in (B, D, F) and the mean fluorescence intensity of wrmScarlet in a worm in each replicate in (L). Statistical significance was performed by unpaired *t* test, three biological independent replicates.

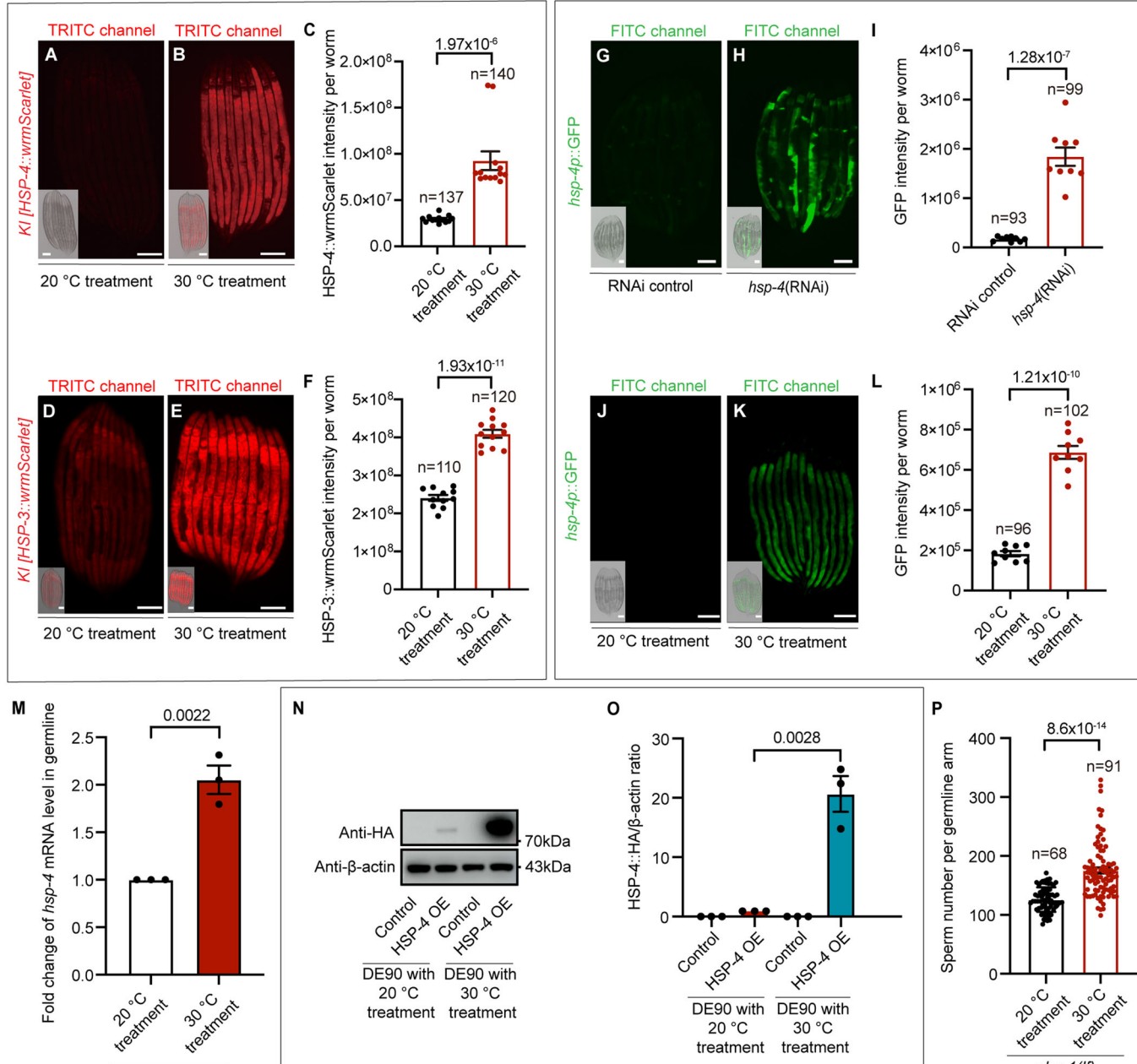

◀

**Figure EV3.  The responsiveness of BiP to the temperature change.**

(A–F) Micrographs and bar graphs showing the induction of BiPs by warmer temperature. After treatment with the indicated temperatures, the BiPs' levels were analyzed by determining the fluorescence intensity of native HSP-4::wrmScarlet and HSP-3::wrmScarlet. The quantification of the mean fluorescence intensity of HSP-4::wrmScarlet and HSP-3::wrmScarlet per worm in (A, B) and (D, E) is shown in (C, F), respectively. *P* value $= 1.97 \times 10^{-6}$ (C), *P* value $= 1.93 \times 10^{-11}$ (F). Scale bar: 100 µm. (G–I) Fluorescent images and a corresponding bar graph showing that *hsp-4*(RNAi) induces *hsp-4p*::GFP expression. The transcription of *hsp-4* was obviously induced in the transcriptional fusion *hsp-4p::GFP* reporter worms treated with *hsp-4(RNAi)* by feeding at a constant 20 °C. The quantification of the mean fluorescence intensity of GFP per worm in (G, H) is shown in (I). *P* value $= 1.28 \times 10^{-7}$ (I). Scale bar: 100 µm. (J–L) Micrographs and corresponding quantitative analyses showing the induction of *hsp-4* transcription in the worms treated with the warmer temperature. The *hsp-4p::GFP* signal was markedly enhanced by 30 °C treatment (J, K), which further support the notion that warmer temperature induces a reduction in the available BiP levels. Quantification of the mean fluorescence intensity of GFP per worm is shown in the bar graph (L). Scale bar: 100 µm. *P* value $= 1.21 \times 10^{-10}$ (L). (M) qPCR analysis indicating the induction of *hsp-4* transcription in the germline after warmer temperature treatment. After the treatment with indicated temperatures, the mRNA of *hsp-4* was isolated from the dissected germline arms and then analyzed by qPCR. *P* value $= 0.0022.$ (N, O) Western blot images and the corresponding quantification showing the successful overexpression of HSP-4. The *Is [hsp-16.41::HSP-4::Flag::HA]* was created in DE90 background. The indicated worms were grown at 20 °C for 30 h, and then the worms in 30 °C treatment group were shifted to the 30 °C for 12 h. The corresponding quantitative analysis was presented in (O). *P* value $= 0.0028$ (O). (P) A bar graph showing that excess sperm were produced in the germline of the *xbp-1(lf)* mutant in response to the 30 °C treatment. The sperm number in each germline arms were counted by DAPI staining. *P* value $= 8.6 \times 10^{-14}.$ The data are presented as the means ± SEMs. Each dot represents the mean fluorescence intensity of wrmScarlet in a worm in each replicate in (C, F), the mean fluorescence intensity of GFP in a worm in each replicate (I, L), the fold change in *hsp-4* mRNA levels in each replicate in (M), the value of HSP-4::Flag::HA/beta-actin ratio in each replicate in (O) and the sperm number in one germline arm in (P). Statistical analyses were performed by unpaired t test; three biological independent replicates.

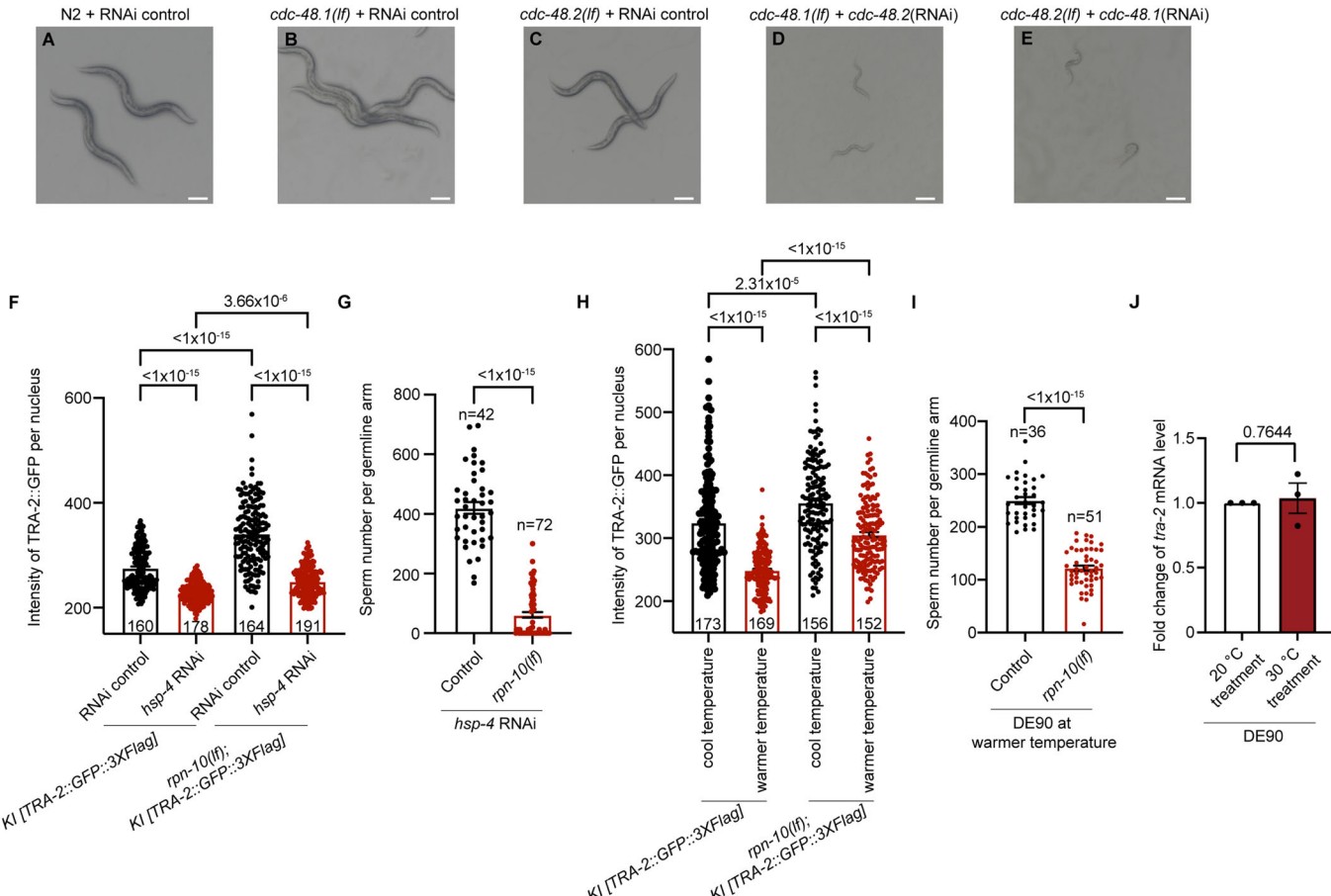

**Figure EV4. CDC-48.1 and CDC-48.2 function redundantly to modulate *C. elegans* development.**

(A–E) Images showing that *cdc-48.1(lf)* with *cdc-48.2*(RNAi) treatment and *cdc-48.2(lf)* with *cdc-48.1*(RNAi) treatment resulted in severe developmental defects. Synchronized N2, *cdc-48.1(lf)* and *cdc-48.2(lf)* worms with indicated feeding RNAi treatment, cultured at a constant 20 °C, were scored for the developmental defects. The number of nematodes assessed in each group exceeds 50. Scale bar: 100 μm. (F) Bar graphs illustrating that the decrease in TRA-2 expression induced by *hsp-4*(RNAi) was suppressed by the *rpn-10(lf)* mutations. The RNAi experiments were performed by feeding. *P* value $<1 \times 10^{-15}$, *P* value $<1 \times 10^{-15}$, *P* value $= 3.66 \times 10^{-6}$, *P* value $<1 \times 10^{-15}$ (from left to right). (G) Bar graphs indicating the suppression of *hsp-4*(RNAi)-induced Mog by the *rpn-10(lf)* mutations. The *rpn-10(lf)* mutations were generated in the DE90 strain by the CRISPR–Cas9 method, and the worms were treated with RNAi against *hsp-4* by microinjection. *P* value $<1 \times 10^{-15}$. (H) A bar graph indicating that warmer temperature (30 °C)-induced decrease in TRA-2 levels was suppressed by the *rpn-10(lf)* mutation. Due to the temperature sensitivity of *rpn-10(lf)* (Shimada et al, 2006), the worms were cultured at a constant 25 °C (cool temperature) to observe the mutant phenotype. *P* value $<1 \times 10^{-15}$, *P* value $= 2.31 \times 10^{-5}$, *P* value $<1 \times 10^{-15}$, *P* value $<1 \times 10^{-15}$ (from left to right). (I) Bar graphs indicating that the *rpn-10(lf)* mutation significantly suppresses the warmer temperature (30 °C) treatment-induced excess sperm production. The *rpn-10(lf)* mutation was generated in the DE90 strain using the CRISPR–Cas9 method. *P* value $<1 \times 10^{-15}$. (J) qPCR analysis indicating that *tra-2* transcription is not affected by the temperature elevation. DE90 worms with the indicated temperature treatments were collected for qPCR analyses. *P* value $= 0.7644$. The data are presented as the means ± SEMs. Each dot in the bar graph corresponds to the mean intensity of native TRA-2::GFP per nucleus in (F, H), the sperm number per germline arm in (G, I) and the fold change in *tra-2* mRNA levels in each replicate in (J). Statistical analyses were performed by unpaired *t* test in (G, I, J) and one-way ANOVA in (F, H). Three biological independent replicates were conducted.

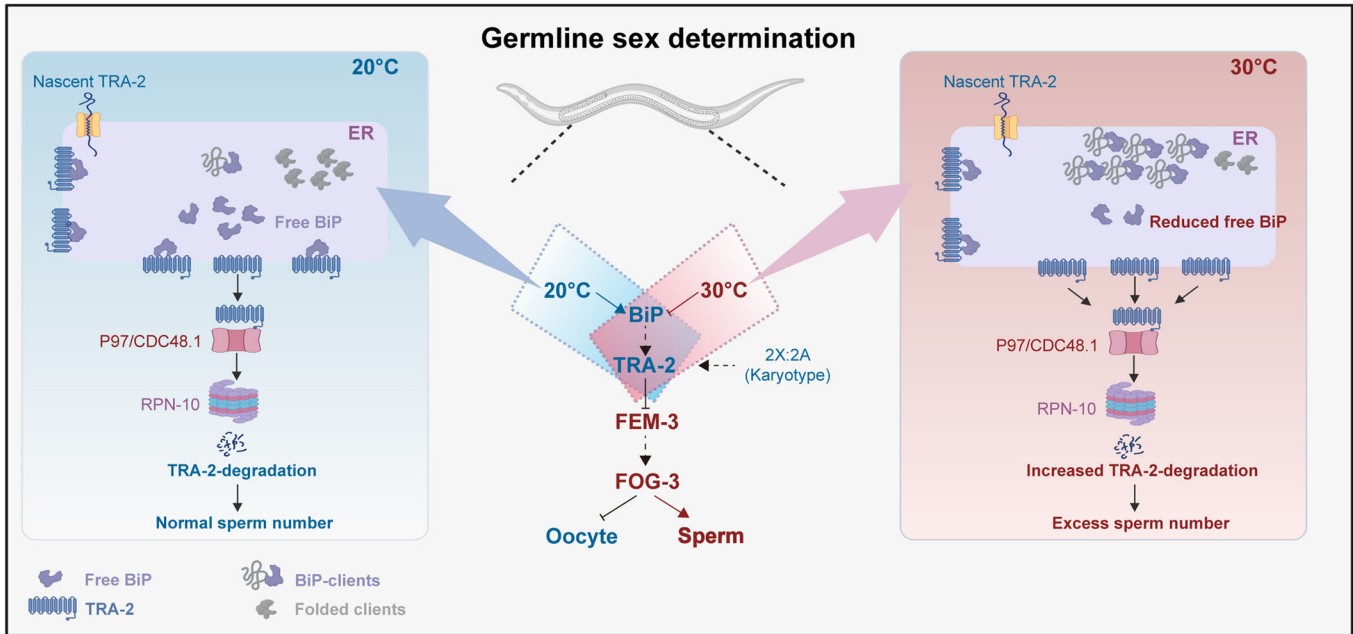

**Figure EV5.   A proposed model of BiP-mediated temperature-dependent germline sex reversal in *C. elegans*.**

Model of the role of the thermosensitive ER chaperone BiP in translating temperature cues into the germline sex determination signal. The karyotype was previously shown to modulate TRA-2 expression to determine germline sex (Ellis and Schedl, 2007). In this study, we found that the thermosensitive regulator BiP senses and translates temperature cues into a germline sex regulatory signal by modulating the level of TRA-2. Specifically, we revealed that BiP is required for driving female germline fate by preventing the ERAD-mediated degradation of TRA-2, which acts as an important posttranslational mechanism to modulate germline sex determination. As temperature has been shown to directly affect protein folding (Day et al, 2002), the amount of available BiP is altered in response to alterations in temperature via detection of the resultant fluctuations in ER protein folding. When worms are exposed to the warmer temperature (30 °C), the amount of available BiP is decreased, which causes a decrease in TRA-2 expression to promote male germline fate. Therefore, BiP acts as a temperature sensor that mediates germline sex determination. Taken together, these findings indicate that the temperature, via a BiP-mediated process, and the karyotype co-regulate the key regulator of the sex determination pathway, TRA-2, to modulate germline sex in *C. elegans*, thus enabling the coexistence of TGSD and GGSD.

