## [Peer Review File · The EMBO Journal]

Identification of BiP as a temperature sensor mediating temperature-induced germline sex reversal in *C. elegans*

Hongyun Tang, Jing Shi, Danli Sheng, Jie Guo, Fangyuan Zhou, and Shaofeng Wu

Corresponding author: Hongyun Tang (tanghongyun@westlake.edu.cn)

Review Timeline:

Submission Date:	25th Feb 24
Editorial Decision:	24th Mar 24
Revision Received:	29th May 24
Editorial Decision:	25th Jun 24
Revision Received:	2nd Jul 24
Accepted:	17th Jul 24

Editor: Kelly Anderson

Transaction Report:

Dear Dr. Tang,

Thank you for submitting your manuscript for consideration by the EMBO Journal. It has now been seen by three referees whose comments are shown below.

Given the referees' positive recommendations, I would like to invite you to submit a revised version of the manuscript, addressing the comments of all three reviewers. I should add that it is EMBO Journal policy to allow only a single round of revision, and acceptance of your manuscript will therefore depend on the completeness of your responses in this revised version. It would be good to discuss your plan to address referee concerns and I am available to do so via zoom or email in the coming weeks.

Thank you for the opportunity to consider your work for publication. I look forward to your revision.

Yours sincerely,

Kelly M Anderson, PhD
Editor, The EMBO Journal
k.anderson@embojournal.org

We realize that it is difficult to revise to a specific deadline. In the interest of protecting the conceptual advance provided by the work, we recommend a revision within 3 months (22nd Jun 2024). Please discuss the revision progress ahead of this time with the editor if you require more time to complete the revisions.

Referee #1:

In the manuscript by Shi and colleagues, the authors address a fundamental aspect of biology that is elegant, yet simple. There is a long standing field of inquiry into the genetic requirements for sex determination, but there is very little about the environmental inputs into sexual dimorphisms. Many vertebrates and invertebrates can override genetic determinants of sex based solely on environmental conditions. One such condition, temperature can be a potent transducer of sex determination.

As an entry into this new field, the authors have taken advantage of the fact that the nematode, *C. elegans* can alter its sexual identity by simply changing the temperature at which it is raised. In this study the authors identify wild isolates and mutants of *C. elegans* that can override genetic determinants of sex determination.

In this study the authors do a remarkable job explaining and detailing their experiments that show how temperature can override genetic determination of sexual identity. The most compelling part of this body of work is that the authors go on to uncover a mechanism by how this can happen.

While it is not entirely clear how the authors chose the 83 genes to screen via RNAi (their explanation in the text is not clear how they narrowed heat responsive genes and sex determination to 82 genes), the authors find that knockdown of hsp-4 (or the ER chaperone hsp70, BiP) is absolutely required. They follow this lead and uncover that their RNAi knockdown of both orthologues of BiP in the worm, hsp-3 and hsp-4, thus explaining the redundancy when mutations of hsp-4 do not recapitulate the phenotype of the RNAi.

Now, why does BiP play a crucial role in thermos-sex determination? Here, the authors uncover that levels of TRA-2, the major genetic player of sex determination, is also regulated by temperature and this regulation is at the protein level, not transcription. The authors find that BiP can directly bind to TRA-2 and BiP is responsible for the changing levels of TRA-2 during gametogenesis. The authors then postulate that TRA-2 is a target of ER associated degradation and loss of cdc-48 (p97) has similar effects as BiP.

While the results are fascinating a few key questions arise:

- 1) Is TRA-2 really in the lumen of the ER to be targeted by BiP and ERAD? Subcellular fractionation experiments, ER - colocalization, references to support claim?
- 2) The authors need to test knockdown of the other HSP70 orthologues to show specificity to BiP.
- 3) BiP expression under heat stress is regulated by xbp-1. IS xbp-1 also involved?
- 4) Cdc-48/p97 is a sledgehammer of a tool to look at ERAD and affects MANY other processes. The authors need to more precisely target ERAD and demonstrate the same effects. Suggestions include HRD1.
- 5) Minor, some of the writing has extra "a, in, the, etc"

Referee #2:

General summary and significance of the study, along with questions and answers

This manuscript presents a novel study investigating the impact of high temperatures on germline sex reversal in the nematode *C. elegans*. The authors demonstrate that exposing animals to heat treatment during their late larval stages triggers a masculinization of the germline (MOG) phenotype, primarily due to an increased tendency of germ cells to adopt sperm fates. This phenomenon is shown to be dependent on the heat shock chaperone hsp-4, which operates in conjunction with hsp-3, both of which are BiP homologs. The study further elucidates that changes in BiP expression influence the expression of key sex determination pathway genes, including tra-2, fem-3, and fog-3, thereby promoting spermatogenesis.

A particularly compelling finding is the demonstration that genes within the sex determination pathway can respond to environmental cues such as temperature changes, affecting germ cell fate. This is the first report in any animal model of specific

genes involved in germ cell fate determination that are modulated by thermal conditions.

Researchers have been studying the relationship between temperature and offspring ratios in different systems for quite some time. We have learned that while sex determination of offspring depends on the genotype of the parent, environment also plays an important role in the process. In this regard, the identification of BiP as a potential thermal sensor offers intriguing insights into how environmental factors intersect with genetic determinants to influence sex determination.

While the manuscript is well-organized and presents data clearly, there are several areas where further clarification or additional data could significantly strengthen the conclusions drawn.

Major concerns to be addressed

Genetic Background and Chromosomal Aberrations:

The reliance on a single transgenic strain (DE90) with multiple integrated transgenes raises concerns about the potential influence of chromosomal aberrations. Incorporating analyses of BiP and sex determination pathway genes in the identified wild isolates could validate the findings from DE90, enhancing the manuscript's robustness.

Selection of Wild Isolates:

The criteria for selecting the wild isolates from among hundreds available are unclear. Authors should clarify whether geographical factors influenced the selection of strains. Do ED3017 and EG4725 come from the same region? Is it expected to find ~20% of isolates in the wild to show a Mog phenotype? That seems rather high!

Temperature Specificity:

The exclusive induction of MOG phenotype at 30°C, with no effects at slightly lower or higher temperatures, requires an explanation. In other words, what is special about the 30°C? A discussion of this specific temperature sensitivity and the absence of a graded response across a wider temperature range is needed.

BiP Expression and Functionality (lines 428-429):

The evidence supporting an increase in total BiP following heat treatment in the Figure 3F is not very convincing. Authors wrote that total BiP was up following warmer temperature treatment. However, there may not be an increase in total BiP, considering that the Actin control band shows higher intensity (Figure 3F). This part needs to be addressed.

Related to this, I am unconvinced by the argument that warmer temperature treatment triggered a decrease in available BiP in worms. While it is true that BiP is more associated with lysate and less free, it does not show there is less 'free' BiP in the germline. The argument presented in the section ('ER chaperone BiPs sense...') should accommodate this fact. Additionally, how could reduced BiP in germ cells trigger the ERAD pathway to specifically degrade TRA-2 and not affect other proteins?

There is some confusion about data presented in Figure 4B,C and 6G. If hsp-4 RNAi causes a massive increase in Mog phenotype in N2 worms, then should we not expect sperm numbers to go up as well without heat treatment?

Mechanism of HSP-4 Response to Temperature:

The finding that hsp-4 responds to temperature increase is exciting and puzzling at the same time. Is it the protein itself or the pathway regulating its expression that senses temperature changes? Mechanistically, how could HSP-4 sense the temperature? Given that it is a chaperone regulated by the ER-UPR, one would expect the pathway to respond to temperature changes. However, none of the other ER-UPR components showed the Mog phenotype in RNAi experiments.

Minor concerns

Lines 112-114: Refine the sentence to more accurately reflect that reduced BiP levels lead to decreased TRA-2 levels.

Since fog-1 functions at the same level as fog-3. Did authors test fog-1 levels following heat treatments? Was that affected as well?

Discussion on the implications of differing fog-3 levels between DE90 and N2 worms would be insightful. What could >50% fog-3 do mechanistically that ~25% fog-3 cannot?

A lot of the details in Section 'Warmer temperature reverses...' are about tra-2 strain construction. Consider moving some of that to a supplementary data file. This would help keep focus on the main story, i.e., the effect of temperature on germline sex determination.

Lines 236-238: While fem-3 RNAi data is supportive, I would have liked to see fem-3 levels in those worms. Was that affected as well?

Line 304: change 'promoted' to 'prompted'

Line 334: Did authors test tra-2 mRNA levels?

Lines 623-624: How were Mog and non-Mog animals identified?

Lines 630-632: Not sure what this means. If you are dealing with adult worms and their germ cells then there is no issue with changing the sex of somatic cells.

Lines 643-644: Were these females able to produce progeny without mating?

Lines 755-756: I don't think that worms are at the L3 stage. A 12 hrs heat shock starting the late-L3 stage will take the animals to early L4 stage. Also, I don't understand what it means by 'when its germ cells switch from sperm fate to oocyte fate'. Germ cells adopt a specific fate!

FIGURES and TABLES

The figure and table legends could be improved. Often, important details have been left out, forcing the reader to go over the results sections carefully to find details.

Figure 1 mentions $n > 29$ germline arms per group. Does this mean > 29 animals? Or, both arms were counted in each animal? A similar issue applies to other figure legends as well.

Figure 2A Y-axis: Is it in percentage or fold change?

Figure 4: It appears that hsp-4 RNAi was done by injection. If so, then it should be made clear in the legend.

Table 1: mention that RNAi was done in DE90 animals.

Additional suggestions, at author/editor's discretion

The manuscript offers valuable insights into the complex interplay between environmental factors and genetic pathways in determining sex in *C. elegans*. Addressing the above concerns and questions would strengthen the evidence for the proposed mechanisms and broaden the impact of the findings.

Referee #3:

The *C. elegans* sex determination pathway has been extensively studied and shown to be driven by a genetic cascade (GSD). Shi et al. identify a layer of temperature control on the sex determination pathway suggesting that GSD can be influenced by the environment. Using a large array of approaches, the authors identify *C. elegans* strains that have a Mog (Masculinization of germline) phenotype upon temperature shift. They systematically examine the germ line sex determination pathway and find decreased levels of the oocyte driving TRA-2 at high temperatures. They go on to show that this is driven by the ER chaperone, BiP, and provide evidence that this may be conserved in male/female worms. This is a very interesting study that provides mechanistic insight into how germ line sex determination can be modulated by the environment (temperature) with broad implications for understanding sex determination controlled at both the genetic and environmental levels.

The manuscript is unnecessarily lengthy and would benefit from extensive editing. The authors should ensure that a non-worm audience doesn't get bogged down in worm terminology.

The following points should also be addressed:

1. If there is no shift back to 20C, will the strains continue to make sperm?
2. I recommend changing EV3 - seems like a lot of space to say that there are no DNA changes in the sex determination genes in the sensitized worm strain.
3. Given that there is WGS available of the sensitized temperature-induced Mog strain, did the authors examine BiP or components of ERAD to see if they are altered?
4. In Figure 3, please quantify the change in total BiP levels.
5. Need to explain the significance of rpn-10.
6. I recommend making *C. remanei* part of Figure 6 rather than in the extended figures. I would also like to know whether the *C. remanei* are self-fertilizing when Mog is induced.

Referee #1:

In the manuscript by Shi and colleagues, the authors address a fundamental aspect of biology that is elegant, yet simple. There is a long standing field of inquiry into the genetic requirements for sex determination, but there is very little about the environmental inputs into sexual dimorphisms. Many vertebrates and invertebrates can override genetic determinants of sex based solely on environmental conditions. One such condition, temperature can be a potent transducer of sex determination.

As an entry into this new field, the authors have taken advantage of the fact that the nematode, *C. elegans* can alter its sexual identity by simply changing the temperature at which it is raised. In this study the authors identify wild isolates and mutants of *c. elegans* that can override genetic determinants of sex determination.

In this study the authors do a remarkable job explaining and detailing their experiments that show how temperature can override genetic determination of sexual identity. The most compelling part of this body of work is that the authors go on to uncover a mechanism by how this can happen.

While it is not entirely clear how the authors chose the 83 genes to screen via RNAi (their explanation in the text is not clear how they narrowed heat responsive genes and sex determination to 82 genes), the authors find that knockdown of *hsp-4* (or the ER chaperone *hsp70*, BiP) is absolutely required. They follow this lead and uncover that their RNAi knockdown of both orthologues of BiP in the worm, *hsp-3* and *hsp-4*, thus explaining the redundancy when mutations of *hsp-4* do not recapitulate the phenotype of the RNAi.

Response 1:

We thank the reviewer for acknowledging the importance of our discovery. We appreciate the reviewer's valuable comments regarding the rationale for selecting these 82 genes for RNAi screening. In response, we have now included an explanation in the revised manuscript (lines 353-357).

Specifically, to identify potential temperature sensors that regulate sex determination in *C. elegans*, our RNAi screen encompassed genes that have been previously suggested to be responsive to temperature fluctuations or implicated in temperature-dependent sex determination (TSD). Based on these previous studies (Jaffe, 1995, Kohno, Katsu et al., 2010, Sarge & Cullen, 1997, Weber, Zhou et al., 2020), we have incorporated a total of 82 genes, including the ones encoding TRP channels, heat shock proteins (HSPs), and genes previously suggested to be involved in TSD in turtles (Fig. EV5A and Table 2).

Now, why does BiP play a crucial role in thermos-sex determination? Here, the authors uncover that levels of TRA-2, the major genetic player of sex determination, is also regulated by temperature and this regulation is at the protein level, not transcription. The authors find that BiP can directly bind to TRA-2 and BiP is responsible for the changing levels of TRA-2 during gametogenesis. The authors then postulate that TRA-2 is a target of ER associated degradation and loss of *cdc-48* (p97) has similar effects as BiP.

While the results are fascinating a few key questions arise:

- 1) Is TRA-2 really in the lumen of the ER to be targeted by BiP and ERAD? Subcellular fractionation experiments, ER -colocalization, references to support claim?

Response 2:

We appreciate the reviewer's inquiry regarding whether there are any references or data supporting the ER localization of TRA-2. Both previous studies and our results presented in the original manuscript support the localization of the transmembrane TRA-2 on the ER membrane, which allows it to be targeted by BiP and ERAD. Additionally, we have included new data in this response letter to further confirm the ER localization of TRA-2.

Firstly, both previous studies and the data in our original manuscript support the localization of transmembrane TRA-2 on the ER membrane. Specifically, it has been previously shown that

TRA-2 is a transmembrane protein on the plasma membrane (Hubert & Anderson, 2009, Mapes, Chen et al., 2010, Sokol & Kuwabara, 2000), which is also supported by our data in the manuscript (Fig. EV3G-I). As the ER is known to be the site where transmembrane proteins are targeted to the plasma membrane (Spiess, Junne et al., 2019), this supports the ER localization for TRA-2. Moreover, both our results (Fig. EV3G-I) and previous studies (Sokol & Kuwabara, 2000) demonstrate the perinuclear localization of TRA-2, providing additional evidence for its localization on the ER.

Furthermore, we have provided new data in this response letter to further confirm the ER localization of TRA-2 (Figure 1 in the response letter). Specifically, when using the split-GFP strategy in 293T cells for another project to identify the topology of TRA-2 on the ER membrane, our data also shows the localization of TRA-2 on the ER. We found that when co-expressing the split-GFP1-10 fused with the signal peptide of ER BiP that is targeted to the ER lumen and the 7X-split-GFP11 fused to the N-terminus of TRA-2 (Figure 1A-B in the response letter), we observed the presence of the GFP signal. In contrast, when co-expressing 7X-split-GFP11 fused to the N-terminus of TRA-2 with cytosolic split-GFP1-10, no GFP signals were detected (Figure 1C-D in the response letter). These results demonstrate that the N-terminus of TRA-2 faces the lumen of the ER. Moreover, our data indicates that the C-terminus of ER-membrane localized TRA-2 faces the cytosol, which is evidenced by the presence of the GFP signal upon co-expression of 7X-split-GFP11 fused to the C-terminus of TRA-2 with cytosolic split-GFP1-10 (Figure 1E-F in the response letter), but not with ER lumen-localized split-GFP1-10 (Figure 1G-H in the response letter). Therefore, this revealing of ER membrane localization of TRA-2, in conjunction with our data demonstrating the co-precipitation of TRA-2 and ER BiP protein HSP-4 (Fig. 4D in the revised manuscript), provides further support for the interaction between BiP and TRA-2 in the ER.

In summary, TRA-2 does indeed exhibit ER localization. In the revised manuscript, we have included references to further substantiate this point (lines 254-275).

2) The authors need to test knockdown of the other HSP70 orthologues to show specificity to BiP.

Response 3:

We appreciate the reviewer's valuable suggestions, and our data regarding the knockdown of the other HSP70 orthologs indeed support the specific role of BiP in regulating germline sex. Specifically, in our original manuscript, the RNAi screening for identifying potential thermosensitive genes that regulate germline sex has included the orthologs of the HSP70 family in *C. elegans*, including F11F1.1, HSP-110, HSP-1, HSP-3, HSP-4, HSP-6, STC-1, T14G8.3, and T24H7.2 (Table 2 in the revised manuscript). Our data indicated that only *hsp-4(RNAi)*, that knocks down both *hsp-3* and *hsp-4* (Fig. EV5G-L in the revised manuscript), resulted in a change in germline sex determination (Fig. 3A-D, Fig. EV5B and Table 2 in the revised manuscript). Therefore, knockdown of the other HSP70 orthologs did not result in a change in germline sex determination, further supporting the specificity of BiP's role in mediating TGSD.

3) BiP expression under heat stress is regulated by *xbp-1*. IS *xbp-1* also involved?

Response 4 :

We thank the reviewer for posing this insightful question. As the reviewer mentioned, previous research has shown that *xbp-1* is essential for the induction of BiP expression when exposed to heat stress (Hirota, Kitagaki et al., 2006). Therefore, it is reasonable to hypothesize that the *xbp-1(lf)* mutation, which is unable to induce BiP during temperature elevation, may make the worms more prone to altering germline sex determination when exposed to warmer temperatures. Indeed, when comparing the 30 °C treatment of *xbp-1(lf)* mutants to the control treatment at 20 °C, we observed the Mog (see Fig. EV6P in the revised manuscript). Thus, this finding further supports the conclusion that decreased BiP levels mediate the warmer temperature-induced Mog effect. This result has been included in the revised manuscript (lines 436-443).

It is important to note that to identify the temperature sensor regulating germline sex determination irrespective of temperature, the RNAi screen in Table 2 in the manuscript was conducted at 20 °C. Therefore, the result showing the *xbp-1(lf)* mutant displaying Mog with 30 °C treatment is not conflicting with the results from our RNAi experiments performed at 20 °C, which did not reveal the Mog phenotype upon knocking down *xbp-1*. Therefore, the *xbp-*

I(lf) mutant, which is unable to induce BiP level at warmer temperatures, provides a sensitized background for worms to acquire TGSD.

4) Cdc-48/p97 is a sledgehammer of a tool to look at ERAD and affects MANY other processes. The authors need to more precisely target ERAD and demonstrate the same effects. Suggestions include HRD1.

Response 5:

We appreciate the reviewer's insightful question regarding the role of ERAD in germline sex determination. We also thank the reviewer for suggesting further testing of the role of HRD1 E3 ligases in TGSD to confirm the function of ERAD in TGSD. However, since there are three ERAD-related E3 ubiquitin ligases (HRD-1, HRDL-1, and MARC-6) (Sasagawa, Yamanaka et al., 2007), and considering the potential redundancy of these genes, instead, we chose to test the role of RPN-10, which is known as a receptor for recruiting client proteins of ERAD for downstream proteasome degradation. The new data further supports the role of ERAD in mediating TGSD.

Specifically, to further confirm the function of ERAD in TGSD, we investigated whether RPN-10 is required for the reduction in TRA-2 levels and Mog induced by *hsp-4* (*RNAi*) or warmer temperature treatment. Indeed, we found that the depletion of *rpn-10* reversed the reduction in TRA-2 levels and impeded germline masculinization induced by *hsp-4* (*RNAi*) and 30 °C treatment, respectively (Fig. EV8F-I in the revised manuscript). Notably, this observation is consistent with a previous study that showed the involvement of RPN-10 in germline sex determination by regulating TRA-2 expression (Shimada, Kanematsu et al., 2006). Therefore, the role of *rpn-10* in germline sex determination further supports the notion that the ERAD process regulates germline sex by facilitating the degradation of TRA-2. These new data have been incorporated into the revised manuscript (lines 540-566).

In summary, our findings, which include the role of P97, CUL-2, and RPN-10 in the post-translational regulation of TRA-2 (Fig. 5A, B, D, E and Fig. EV8F-I in the revised manuscript), demonstrate that the ERAD process mediates the warmer temperature-induced germline sex determination change in *C. elegans*.

5) Minor, some of the writing has extra "a, in, the, etc"

Response 6:

Thank you for your careful review. We have gone through the entire manuscript to improve the descriptions.

Referee #2:

General summary and significance of the study, along with questions and answers

This manuscript presents a novel study investigating the impact of high temperatures on germline sex reversal in the nematode *C. elegans*. The authors demonstrate that exposing animals to heat treatment during their late larval stages triggers a masculinization of the germline (MOG) phenotype, primarily due to an increased tendency of germ cells to adopt sperm fates. This phenomenon is shown to be dependent on the heat shock chaperone hsp-4, which operates in conjunction with hsp-3, both of which are BiP homologs. The study further elucidates that changes in BiP expression influence the expression of key sex determination pathway genes, including *tra-2*, *fem-3*, and *fog-3*, thereby promoting spermatogenesis.

A particularly compelling finding is the demonstration that genes within the sex determination pathway can respond to environmental cues such as temperature changes, affecting germ cell fate. This is the first report in any animal model of specific genes involved in germ cell fate determination that are modulated by thermal conditions.

Response 7:

We appreciate the reviewer's comments acknowledging the novelty of our discovery.

Researchers have been studying the relationship between temperature and offspring ratios in different systems for quite some time. We have learned that while sex determination of offspring depends on the genotype of the parent, environment also plays an important role in the process.

In this regard, the identification of BiP as a potential thermal sensor offers intriguing insights into how environmental factors intersect with genetic determinants to influence sex determination.

While the manuscript is well-organized and presents data clearly, there are several areas where further clarification or additional data could significantly strengthen the conclusions drawn.

Major concerns to be addressed

Genetic Background and Chromosomal Aberrations:

The reliance on a single transgenic strain (DE90) with multiple integrated transgenes raises concerns about the potential influence of chromosomal aberrations. Incorporating analyses of BiP and sex determination pathway genes in the identified wild isolates could validate the findings from DE90, enhancing the manuscript's robustness.

Response 8:

We thank the reviewer for their insightful comments regarding the potential role of BiP in germline sex regulation in wild-type *C. elegans* strains. We highly value the reviewer's suggestion, and thus we have performed additional experiments to address this point. Our new data, included in the revised manuscript (Fig. EV5C-F), further confirm that the role of BiP in germline sex determination is not exclusive to DE90 strains.

Specifically, we evaluated the role of BiP in promoting female germline sex in the ED3017 and EG4725 wildtype strains. We found that upon *hsp-4(RNAi)* treatment, both ED3017 and EG4725 worms grown at 20 °C displayed masculinization of the germline (Fig. EV5C-F in the revised paper). This result provides support for the involvement of BiP in the regulation of germline sex determination in these wild-type worms, extending beyond the DE90 strains.

Moreover, it is worth noting that in our original study, we also investigated the effect of *hsp-4(RNAi)* on the wild-type N2 worms (Fig. 4B in the manuscript). Similar to what was observed in the ED3017 and EG4725 strains, we found that downregulating BiP levels induced the Mog phenotype in N2 worms. Furthermore, we also observed a reduction in TRA-2 levels following *hsp-4(RNAi)* treatment in N2 worms (refer to Fig. 4F and 4G in the manuscript). Additionally,

the *hsp-4(lf)* mutation, which results in reduced but not complete depletion of BiP due to redundancy with *hsp-3*, sensitized the N2 worms to exhibit TGSD (refer to Fig. 6 A-G in the manuscript). These findings further support the role of BiP in regulating germline sex determination in N2 wild-type worms.

In conclusion, the new data as well as the original data in the manuscript demonstrate that the role of BiP in germline sex determination discovered in the DE90 strain is also present in the wild-type worms. Once again, we would like to express our gratitude to the reviewer for the thought-provoking question, as it reinforces the strength of our conclusion.

Selection of Wild Isolates:

The criteria for selecting the wild isolates from among hundreds available are unclear. Authors should clarify whether geographical factors influenced the selection of strains. Do ED3017 and EG4725 come from the same region? Is it expected to find ~20% of isolates in the wild to show a Mog phenotype? That seems rather high!

Response 9:

We would like to express our gratitude to the reviewer for raising an interesting point. We apologize for omitting the description how the wild isolates were selected. Actually, no specific selection process was employed; rather, we randomly chose the wild-type strains that were readily available from CGC (Caenorhabditis Genetics Center). This random selection has now been included in our revised manuscript (lines 145-149).

The reviewer's suggestion is inspiring, prompting us to analyze the regions that the ED3017 and EG4725 strains are from. These two strains do not originate from the same region, with ED3017 originating from the United Kingdom, while EG4725 is from Portugal. Presently, we are unsure whether the geographical regions that the strains derive from are related to their ability in TGSD. Furthermore, due to the limited sample size (Fig. 1B in the revised manuscript), we are unable to determine the influence of various geographical locations on the occurrence of TGSD. Nevertheless, the reviewer's suggestion regarding the potential impact of geographical factors on TGSD in worms is an intriguing idea, which demands further exploration in a separate project in the future.

Temperature Specificity:

The exclusive induction of MOG phenotype at 30 °C, with no effects at slightly lower or higher temperatures, requires an explanation. In other words, what is special about the 30C? A discussion of this specific temperature sensitivity and the absence of a graded response across a wider temperature range is needed.

Response 10:

We thank the reviewer for raising this interesting point. Indeed, as the reviewer observed, except for the ones subjected to the 30 °C treatment, the sperm quantity of each germline arm in the worms with 25 °C, 28 °C, and 32 °C treatments was not significantly higher than that in the 20 °C control group. Per the reviewer's suggestion, we have included a discussion on this specific temperature sensitivity in the revised manuscript (lines 768-778).

Notably, previous studies have indicated that *C. elegans* possesses the ability to detect temperature changes as small as 0.1 °C (Ramot, MacInnis et al., 2008). This precise temperature monitoring suggests that animals may employ specific coping strategies for specific temperatures. For instance, in response to a temperature of 30 °C, *C. elegans* utilizes a regulated germline sex alteration strategy to enhance reproductive fitness. However, this noticeable change of germline sex is not observed at other warmer temperatures, implying that worms may have evolved alternative strategies to cope with these levels of temperature. The molecular mechanisms underlying this specific temperature (30 °C) sensitivity, as well as its physiological significance, require further investigation from an ecological perspective in the future.

BiP Expression and Functionality (lines 428-429):

The evidence supporting an increase in total BiP following heat treatment in the Figure 3F is not very convincing. Authors wrote that total BiP was up following warmer temperature treatment . However, there may not be an increase in total BiP, considering that the Actin control band shows higher intensity (Figure 3F). This part needs to be addressed.

Response 11:

We thank the reviewer for the thorough review. We apologize for omitting the statistical data concerning the total BiP levels analyzed using western blot in the initial manuscript. Through quantitative analysis of the grayscale value of the bands on the gel, we have confirmed an increase in the levels of total BiP normalized to the actin loading control under the 30 °C treatment. The corresponding bar graph has been included in the revised manuscript (Fig. 3G; lines 400-405).

Furthermore, we highly value the reviewer's question and thus we have performed new experiments to further demonstrate the increase in total BiP upon warmer temperature treatment. By utilizing strains with wrmScarlet knock-in at the C-terminus of the native HSP-3 and HSP-4 BiP, our microscopic analyses also observed a significant increase in the levels of both endogenous HSP-3 and HSP-4 in the worms following the 30 °C treatment (Fig. EV6A-F in the revised manuscript). These new analyses have been incorporated into the revised manuscript (lines 400-405).

In summary, our data provide strong evidence for the induction of total BiPs by the 30 °C treatment.

Related to this, I am unconvinced by the argument that warmer temperature treatment triggered a decrease in available BiP in worms. While it is true that BiP is more associated with lysate and less free, it does not show there is less 'free' BiP in the germline. The argument presented in the section ('ER chaperone BiPs sense...') should accommodate this fact.

Response 12:

We thank the reviewer for the comment regarding the potential decrease in available BiP in the germline with 30 °C treatment. To address this concern, we conducted new experiments, and the results provides evidence for a decrease in available BiP in the germline upon warmer temperature treatment.

It is important to note that performing gel cross-linking analyses of free BiP in the germline is technically challenging, as it requires dissecting approximately 20,000 worms to collect their germline arms. Therefore, we utilized an alternative strategy to analyze the potential decrease in the available BiP in the germline. Based on previous reports (Carrara, Prischi et al., 2015) and

our data (Fig. EV6G-L), a decrease in available BiP could trigger BiP transcription through negative feedback regulation, thus the activation of BiP transcription indicates insufficient available BiP (Carrara, Prischi et al., 2015). Thus, after warmer temperature treatment, we examined the transcription of *hsp-4* in the germline by extracting mRNA from the dissected intact germline arms (refer to the Method section of the revised manuscript for details, lines 1280-1288). Indeed, the results indicate that under the 30 °C treatment, the mRNA level of *hsp-4* in the germline significantly increases compared to the 20 °C treatment (Fig. EV6M; lines 414-418 in the revised manuscript), supporting that warmer temperature treatment leads to a decrease in available BiP levels in the germline.

Furthermore, in the original manuscript, we demonstrated a decrease in available BiPs within worms upon warmer temperature treatment through gel crosslinking analysis (Fig. 3F and 3G) and by analyzing BiP transcription levels (Fig. EV6G-L in the revised manuscript). Moreover, tissue-specific analyses showed that BiPs specifically function in the germline to regulate germline sex (Fig. EV7 in the revised manuscript). These original data, along with our new data suggesting a decrease in free BiP in the germline (Fig. EV6M in the revised manuscript), support that the decrease in available BiP mediates the warmer temperature-induced germline masculinization. In the revised manuscript, we have incorporated this new data (lines 400-422) and also improved the description as the reviewer suggested.

Additionally, how could reduced BiP in germ cells trigger the ERAD pathway to specifically degrade TRA-2 and not affect other proteins?

Response 13:

We thank the reviewer for this inquiry. We sincerely apologize for any confusion that may have been caused by our description. We would like to clarify that our data did not indicate that ERAD specifically degrades TRA-2. Actually, it is totally possible that other ER-localized proteins can also be degraded by ERAD during warmer temperature treatment, maybe for regulating other processes. Since our interest lies in the transition of germline sex and our genetic data support that warm temperature and BiP likely function upstream of TRA-2, we thus analyzed the role of ERAD in regulating the levels of TRA-2 during its localization at the ER. On the other hand, other known proteins involved in sex determination, such as FOG-3, FEM-3,

and TRA-1, are believed to be localized in the cytoplasm or nucleus (Lee, Kim et al., 2011, Noble, Aoki et al., 2016, Starostina, Lim et al., 2007), and therefore, their expression is unlikely to be regulated by the BiP-mediated ERAD pathway.

Taking these factors into consideration, our investigation focused on exploring the role of ERAD in mediating the downregulation of TRA-2 during warm temperature treatment and BiP knock-down. As a result, we discovered that a decrease in BiP levels and warm temperature triggers ERAD-mediated TRA-2 degradation to promote male germline sex in *C. elegans*.

There is some confusion about data presented in Figure 4B,C and 6G. If *hsp-4* RNAi causes a massive increase in Mog phenotype in N2 worms, then should we not expect sperm numbers to go up as well without heat treatment?

Response 14:

Thank you for your thorough review. We would like to provide clarification regarding the data presented in Figure 4B-C and Figure 6G. The data in Figure 4B-C pertains to *hsp-4(RNAi)* experiments, while the data in Figure 6G was obtained from *hsp-4(lf)* mutant experiments.

As stated in our manuscript, *hsp-4 RNAi* treatment results in the knockdown of both *hsp-3* and *hsp-4*, leading to the Mog phenotype (Fig. EV5G-L and lines 372-390 in the revised manuscript). In contrast, the *hsp-4(lf)* single mutation does not induce obvious Mog due to the functional redundancy between HSP-3 and HSP-4 BiPs (Fig. EV5G and lines 372-390 in the revised manuscript). However, the partial reduction in BiP caused by the *hsp-4(lf)* single mutation sensitizes the worms to display Mog when exposed to warmer temperatures, which is supported by our data showing an increase in Mog in the *hsp-4(lf)* mutant compared to the N2 control under 30 °C conditions (Fig. 6G in the revised manuscript).

Therefore, the data presented in Figure 4B,C and Figure 6G align with our research conclusions. Once again, we appreciate the reviewer's careful analysis of our data.

Mechanism of HSP-4 Response to Temperature:

The finding that *hsp-4* responds to temperature increase is exciting and puzzling at the same time. Is it the protein itself or the pathway regulating its expression that senses temperature changes?

Mechanistically, how could HSP-4 sense the temperature? Given that it is a chaperone regulated by the ER-UPR, one would expect the pathway to respond to temperature changes. However, none of the other ER-UPR components showed the Mog phenotype in RNAi experiments.

Response 15:

We thank the reviewer for the questions regarding the mechanism of HSP-4 response to temperature elevation. Based on our experimental data, the levels of available BiP decrease in response to elevated temperatures to promote male germline sex (Fig. 3E-G and Fig. EV6G-M in the revised manuscript). Moreover, a previous study indicated that high temperatures can directly induce protein unfolding (Hata, Ishiwata-Kimata et al., 2022, Liu & Chang, 2008), which could reduce the availability of BiPs due to their increased binding to the unfolded proteins. Therefore, our findings, along with previous research, support that BiPs sense warmer temperatures maybe by detecting the resultant increase in the protein folding burden within the ER, which leads to a reduction in the availability of BiP and ultimately results in the masculinization of the germline.

Regarding the activation of UPR^{ER}, upon temperature elevation, protein unfolding in the ER is increased (Hata et al., 2022, Liu & Chang, 2008), resulting in the allocation of more BiP to bind to their client proteins in the ER lumen to facilitate their folding. This leads to a decrease in the availability of BiP, which then results in the activation of UPR^{ER} and the subsequent induction of BiP expression to maintain ER proteostasis. Therefore, the activation of UPR^{ER} is a result of the insufficient availability of BiP caused by the increased unfolded proteins in the ER (Amin-Wetzel, Saunders et al., 2017, Bakunts, Orsi et al., 2017, Lai, Aronson et al., 2010). In turn, UPR^{ER} is required for the induction of BiP during ER stress conditions, such as warmer temperature treatment. This functional relationship between BiP and UPR^{ER} suggests that inhibiting UPR^{ER} to suppress the induction of BiP expression during temperature elevation should render worms more prone to exhibit Mog upon warmer temperature treatment, like that observed in the *hsp-4(lf)* mutant. To test this hypothesis, we conducted additional experiments with *xbp-1(lf)* that inhibits the induction of BiP under ER stress. Indeed, we found that *xbp-1(lf)* mutants displayed masculinization upon 30 °C treatment, in contrast to the ones maintained at 20 °C (Fig. EV6P and lines 436-443 in the revised manuscript). These results further support the role of decreased BiP levels in mediating the masculinizing effect of warmer temperature.

It is important to note that to identify the temperature sensor regulating germline sex determination irrespective of temperature, the RNAi screen in Table 2 in the manuscript was conducted at 20 °C. Therefore, the result showing the *xbp-1(lf)* mutant displaying Mog with 30 °C treatment is not conflicting with the results from our RNAi experiments performed at 20 °C, which did not reveal the Mog phenotype upon knocking down *xbp-1*. Collectively, the *xbp-1(lf)* mutant, which is unable to induce BiP at warmer temperatures, provides a sensitized background for worms to acquire TGSD.

In summary, our findings support the idea that BiP may sense elevated temperature by detecting the resultant increase in ER-protein unfolding burden.

Minor concerns

Lines 112-114: Refine the sentence to more accurately reflect that reduced BiP levels lead to decreased TRA-2 levels.

Response 16:

Thank you for your valuable suggestion, and we have improved the description in the revised manuscript (lines 108-111).

Since fog-1 functions at the same level as fog-3. Did authors test fog-1 levels following heat treatments? Was that affected as well?

Response 17:

Figure 2

The qPCR analysis revealed that the mRNA level of *fog-1* in DE90 worms with 30 °C treatment is significantly higher compared to those with 20 °C treatment. DE90 worms subjected to the indicated temperature treatments were collected for qPCR analysis. Three independent biological replicates were conducted. The data are presented as means \pm SEMs, and an unpaired t-test was performed to determine statistical significance.

We thank the reviewer for raising this question. To address this point, we performed additional experiments to analyze *fog-1* transcription and the data has been included in this response letter. As the reviewer predicted, our qPCR analyses indicated that the mRNA level of *fog-1* in DE90 worms with a 30 °C treatment is significantly higher than those with a 20 °C treatment (Figure 2 in the response letter). This result further supports our conclusion that warmer temperatures promote sperm fate through altering the activity of germline sex determination pathway.

Discussion on the implications of differing *fog-3* levels between DE90 and N2 worms would be insightful. What could >50% *fog-3* do mechanistically that ~25% *fog-3* cannot?

Response 18:

We appreciate the reviewer's suggestion. Firstly, we would like to acknowledge that due to an unintentional mistake in data analysis, the values from the technique replicates, instead of the values from the biological replicates, were mistakenly used when generating Figure 2A in the original manuscript. We sincerely apologize for this oversight. We would like to clarify that the correct data has now been used in Figure 2A in the revised manuscript, which still supports the original conclusion. In response to the reviewer's advice, we have included a discussion on the impact of varying *fog-3* levels on the ability of DE90 and N2 worms to exhibit TGSD in our revised manuscript (lines 790-795).

Specifically, the DE90 worms exhibit a significantly higher basal level of *fog-3* transcription compared to the N2 worms, which may suggest that the germline sex determination pathway in the DE90 worms is more readily able to reach the necessary threshold for adopting the sperm fate under elevated temperatures (Fig. 2A in the manuscript). Indeed, during the 30 °C treatment, the *fog-3* transcription level in DE90 worms was further induced and substantially higher than that in N2 worms. This observed discrepancy in *fog-3* transcription levels between the two strains upon warmer temperature treatment supports the possibility that the *fog-3* transcription level in DE90 worms exceeded the threshold required for adopting a sperm fate, thereby resulting in partial masculinization of the germline. In contrast, upon warmer temperature treatment, the *fog-3* transcription level in N2 worms was not high enough to reach the threshold necessary for adopting the sperm fate, thus explaining the lack of germline masculinization. To determine the precise dosage of *fog-3* required for the transition from adopting oocyte fate to adopting sperm fate is an intriguing question, which deserves to be explored in a separate project involving extensive quantitative investigations.

A lot of the details in Section 'Warmer temperature reverses...' are about *tra-2* strain construction. Consider moving some of that to a supplementary data file. This would help keep focus on the main story, i.e., the effect of temperature on germline sex determination.

Response 19:

Thank you for your valuable suggestion. Based on your advice, we have made revisions to the manuscript in order to address potential distractions and to focus more on the main story in the revised manuscript (lines 252-277).

Lines 236-238: While *fem-3* RNAi data is supportive, I would have liked to see *fem-3* levels in those worms. Was that affected as well?

Response 20:

Figure 3

The qPCR analysis revealed that the level of the *fog-3* transcript in DE90 worms with 30 °C treatment is significantly decreased upon *fem-3*(RNAi). The RNAi was performed by feeding in DE90 worms. Three biological replicates were conducted. The results are presented as means \pm SEMs, and the statistical significance was determined using an unpaired t-test.

We would like to express our gratitude to the reviewer for this inquiry. Due to the lack of antibodies and the unknown difficulties associated with introducing a fluorescent tag into the *fem-3* locus in DE90, we have been unable to examine the effect of temperature on FEM-3 protein levels. Nonetheless, we greatly appreciate the reviewer's question and have conducted additional experiments to support the necessity of *fem-3* for temperature-induced changes in germline sex determination activity. These new results have been included in Figure 3 of the response letter.

Specifically, we investigated whether the increase in *fog-3* mRNA levels at higher temperatures is dependent on *fem-3*. Remarkably, our findings demonstrate that knocking down *fem-3* significantly suppresses the induction of *fog-3* by the 30 °C treatment (Figure 3 in the response letter). Furthermore, our previous results in the original manuscript indicate the

suppression of warmer temperature-induced Mog by *fem-3 RNAi* (Fig. 2B in the revised manuscript). Collectively, these data provide compelling evidence demonstrating that the masculinizing effect of warmer temperatures is dependent on *fem-3*. We hope that the reviewer will agree with us regarding this point. We have made the necessary modifications to the description to align it with the data in the revised manuscript.

Line 304: change 'promoted' to 'prompted'

Response 21:

Thank you for your valuable suggestion. We have replaced the word "promoted" with "prompted" in the revised manuscript (line 288).

Line 334: Did authors test *tra-2* mRNA levels?

Response 22:

Thank you for your advice. As you suggested, we conducted further investigations on the *tra-2* transcription by qPCR analyses. The results reveal that, under warmer temperature conditions, there was no significant difference in the mRNA level of *tra-2* in DE90 worms compared to the 20 °C treatment (Fig. EV8J in the revised manuscript). This lack of change in *tra-2* transcription is consistent with our research showing TRA-2 regulation by ERAD, aligning with previous studies indicating that *tra-2* is primarily regulated through post-transcriptional and post-translational mechanisms.

Therefore, the decrease in TRA-2 protein levels caused by warmer temperature treatment is not attributable to a reduction in *tra-2* mRNA levels, further supporting the crucial role of ERAD-mediated post-translational regulation of TRA-2 under warmer temperature treatment. This result has been included in the revised manuscript (lines 562-566).

Lines 623-624: How were Mog and non-Mog animals identified?

Response 23:

We sincerely apologize for initially omitting a detailed description showing the method used to identify Mog and non-Mog animals in our manuscript. This information has now been included in the revised manuscript (lines 1256-1262). Specifically, following the warmer temperature treatment, we conducted a count of sperm in the live DE90 worms, whose sperm had been labeled with *Is[*spe-11p*::*mCherry*::*histone*]* transgene, using a fluorescent microscope to assess whether the worms displayed the Mog or non-Mog phenotype. Subsequently, we carefully transferred both the Mog and non-Mog worms from the agarose pad to NGM dish with one hermaphrodite per plate, ensuring minimal harm to the organisms throughout the process. These worms were then cultivated in order to determine their brood sizes. We appreciate the reviewer's question, which helps to improve the clarity of our paper.

Lines 630-632: Not sure what this means. If you are dealing with adult worms and their germ cells then there is no issue with changing the sex of somatic cells.

Response 24:

We apologize for any confusion caused by our description. The original intention of this sentence is to discuss why warmer temperatures only affect the germline sex in hermaphrodites. We emphasize that altering the germline sex is sufficient to adjust the brood size of hermaphrodites and the reproductive fitness, thus there is no necessity for hermaphrodites to modify the somatic sex upon warmer temperature treatment. To avoid any potential confusion, we have removed this discussion in the revised manuscript.

Lines 643-644: Were these females able to produce progeny without mating?

Response 25:

Thank you for your insightful question. Despite both sperm and oocytes being generated in females of the *C. remanei* species after *hsp-4(RNAi)* (Fig. 6M-S in the revised manuscript), these nematodes are unable to produce progenies through self-fertilization. One possible reason for their inability to generate self-fertilized offspring may be due to a defect in sperm activation. In supporting this postulation, previous research has indicated that the successful transformation of females into hermaphrodites requires not only sperm production but also subsequent sperm activation that does not originally exist in the females (Ellis, 2017). To enable the transformation

of *C. remanei* females into self-fertilizable hermaphrodites, after *hsp-4(RNAi)* to alter the female germline sex to induce sperm production, another genetic manipulation for activating these sperm may be required. Once again, we would like to express our gratitude to the reviewer for posing this thought-provoking question, as it highlights an important area for future research.

Lines 755-756: I don't think that worms are at the L3 stage. A 12 hrs heat shock starting the late-L3 stage will take the animals to early L4 stage. Also, I don't understand what it means by 'when its germ cells switch from sperm fate to oocyte fate'. Germ cells adopt a specific fate!

Response 26:

Thank you for your valuable suggestions, and we sincerely apologize for any confusion caused by the unclear expression in our initial manuscript.

Firstly, regarding the point "worms are at the L3 stage," since the phenotype of Mog appears after exposing hermaphrodites at the late L3 stage to a 30 °C treatment for 12 hours, we stated that "*C. elegans* is sensitive to the effect of temperature at the L3 stage". Following your suggestion, we have revised the description in the manuscript to "*C. elegans* is sensitive to temperature effects during the L3-L4 stage (line 749)" and in other lines (174, 184-185)

Secondly, we apologize for any confusion caused by our inaccurate description regarding 'when its germ cells switch from sperm fate to oocyte fate'. We agree with the reviewer's comment. In the initial manuscript, we had aimed to convey that bipotential germ cells transition from adopting the sperm fate to adopting the oocyte fate, rather than switching their adopted cell fate. In the revised manuscript, we have already amended the description accordingly (lines 749-750). Thanks once again for your suggestion that makes our descriptions more accurate.

FIGURES and TABLES

The figure and table legends could be improved. Often, important details have been left out, forcing the reader to go over the results sections carefully to find details.

Response 27:

Thank you very much for your comments, which help enhance the clarity of our paper. In response to your feedback, we have reviewed all the figure and table legends and made necessary

changes to enhance the level of detail, such as specifying the RNAi method employed, providing a clear definition of the y-axis and adding essential information about the treatments.

Figure 1 mentions $n > 29$ germline arms per group. Does this mean > 29 animals? Or, both arms were counted in each animal? A similar issue applies to other figure legends as well.

Response 28:

Thank you for your valuable inquiry. In accordance with previous research (Sasagawa, Otani et al., 2009, Tang & Han, 2017), we conducted the statistical analysis by quantifying the number of sperm per germline arm. Each germline arm is considered as an independent “unit” that produces sperm, thus, “n” denotes the number of germline arms scored.

To determine the sperm count, z-stacks of the germline arms were captured using spinning disk confocal microscopy and subsequently reconstructed in 3D to facilitate counting. As long as the images of a germline arm are clear, they are used for quantifying the sperm count. It is worth mentioning that, it can be sometimes challenging to obtain clear images of both germline arms in the same worm due to autofluorescence signal interference from the intestine. Consequently, there are instances where only one germline arm is counted for certain animals. Nonetheless, this does not affect our results, as “n” represents the scored germline arm.

We greatly appreciate your questions and we have included this clarification in the methods section (lines 1131-1136).

Figure 2A Y-axis: Is it in percentage or fold change?

Response 29:

Thank you for your thorough review. Firstly, we would like to acknowledge that due to an unintentional mistake in data analysis, the values from the technique replicates, instead of the values from the biological duplicates, were mistakenly used when generating Figure 2A in the original manuscript. We sincerely apologize for this oversight. We would like to clarify that the correct data has now been used in Figure 2A in the revised manuscript, which remains consistent with the original conclusion.

Regarding the reviewer's question, we would like to clarify that in Figure 2A, the Y-axis represents the fold change. In order to improve clarity, we have modified the description of the Y-axis to clearly indicate the fold change in *fog-3* mRNA levels for different treatments by normalizing to the *fog-3* mRNA levels of N2 cultured at 20 °C (Fig. 2A in the revised manuscript). This increase in *fog-3* transcription in Figure 2A is consistent with prior studies showing that the *fog-3* mRNA level is obviously increased when germline appears Mog phenotype (Li & Kelly, 2014, Tang & Han, 2017).

Figure 4: It appears that hsp-4 RNAi was done by injection. If so, then it should be made clear in the legend.

Response 30:

Thank you for your suggestion, and we have included this information in the legends of figures 4 as well as other figures throughout the revised manuscript (Fig.3A-D, Fig. 4A-C, Fig. 6M-S, Fig. EV5C-F, Fig. EV7 and Fig. EV8G).

Table 1: mention that RNAi was done in DE90 animals.

Response 31:

Thank you for your suggestion, and we have included this strain information in Table 2 in the revised manuscript (Table 1 in the original manuscript).

Additional suggestions, at author/editor's discretion

The manuscript offers valuable insights into the complex interplay between environmental factors and genetic pathways in determining sex in *C. elegans*. Addressing the above concerns and questions would strengthen the evidence for the proposed mechanisms and broaden the impact of the findings.

Response 32:

We thank the reviewer for acknowledging the importance of our discovery. We highly appreciate the reviewer's thoughtful questions and comments, which have been thoroughly addressed.

Referee #3:

The *C. elegans* sex determination pathway has been extensively studied and shown to be driven by a genetic cascade (GSD). Shi et al. identify a layer of temperature control on the sex determination pathway suggesting that GSD can be influenced by the environment. Using a large array of approaches, the authors identify *C. elegans* strains that have a Mog (Masculinization of germline) phenotype upon temperature shift. They systematically examine the germ line sex determination pathway and find decreased levels of the oocyte driving TRA-2 at high temperatures. They go on to show that this is driven by the ER chaperone, BiP, and provide evidence that this may be conserved in male/female worms. This is a very interesting study that provides mechanistic insight into how germ line sex determination can be modulated by the environment (temperature) with broad implications for understanding sex determination controlled at both the genetic and environmental levels.

Response 33:

We thank the reviewer for acknowledging the importance of our work.

The manuscript is unnecessarily lengthy and would benefit from extensive editing. The authors should ensure that a non-worm audience doesn't get bogged down in worm terminology.

Response 34:

Thank you for your valuable suggestions. We highly valued your feedback and made revisions to the description throughout the entire paper to enhance its readability for non-worm readers. Additionally, we have improved the writing to increase clarity and reduce unnecessary length.

The following points should also be addressed:

1. If there is no shift back to 20C, will the strains continue to make sperm?

Response 35:

Thank you for your inquiry. Based on our observations, when extending the cultivation time at 30 °C to approximately 24 hours, the worms begin to generate oocytes and produce offspring. Therefore, *C. elegans* do not continue to make sperm when constantly maintained at warmer temperatures. As we know that 20 °C is more optimal than 30 °C for worms to grow, we thus have included this temperature switch back to 20 °C in our assay.

Currently, the reason why worms cultivated constantly at 30 °C do not continuously generate sperm remains unknown, which is worth further exploration in a separate paper in the future.

2. I recommend changing EV3 - seems like a lot of space to say that there are no DNA changes in the sex determination genes in the sensitized worm strain.

Response 36:

Thank you for your invaluable suggestion. We agree with the reviewer and we have converted EV3 into an Excel format and included it in the supplementary data (Table 1 in revised manuscript).

3. Given that there is WGS available of the sensitized temperature-induced Mog strain, did the authors examine BiP or components of ERAD to see if they are altered?

Response 37:

gene	wormbase ID	Analysis of SNP and indel			
		Upstream(5kb)	Exonic	Intronic	Downstream(5kb)
cdc-48.1	C06A1.1	/	/	/	/
cdc-48.2	C41C4.8	/	/	/	/
npl-4.1	F59E12.4	/	/	/	/
npl-4.2	F59E12.5	/	/	/	/
ufd-1	F19B6.2	/	/	/	/
sel-11	F55A11.3	/	/	/	/
hrdl-1	F26E4.11	/	/	/	/
marc-6	F55A3.1	/	/	/	/
ufd-2	T05H10.5	/	/	/	/
ufd-3	C05C10.6	/	/	/	/
rad-23	ZK20.3	/	/	/	/
rpn-10	B0205.3	/	/	/	/
rpt-5	F56H1.4	/	/	/	/
hsp-3	C15H9.6	/	/	/	/
hsp-4	F43E2.8	/	/	/	/

Figure 4

No mutations were detected in the ERAD and BiP genes in the DE90 strain. "/" is used to indicate that there is no difference in the sequence of the indicated region of the BiP and ERAD-related genes between DE90 and the reference sequence (WS288). The mutation analyses of DE90 were conducted using whole-genome sequencing.

Thank you for your suggestion. We have performed additional analyses and found that there are no differences in the genomic sequences of BiP and ERAD-related genes between the DE90 worms and the reference sequence (Figure 4 in the response letter). These results support the idea that the ability of DE90 worms to exhibit TGSD is unlikely due to mutations in BiP and ERAD-related genes. It is worth mentioning that although we do not know the reason for the DE90 strain acquiring TGSD, the role of BiP in germline sex determination initially discovered from DE90 worms (Fig. 3A-D in the revised manuscript) can be extended to N2, ED3017, and EG4725 wildtype worms (Fig. 4B-C and Fig. EV5C-F in the revised manuscript).

4. In Figure 3, please quantify the change in total BiP levels.

Response 38:

Thank you for your suggestion. We apologize for omitting the quantitative data regarding the change in total BiP levels. We have performed additional quantitative analyses and observed an elevation in the total BiP/actin ratio following the 30 °C treatment, further reinforcing the conclusions drawn in the original paper. This information has been included in Figure 3G in the revised manuscript.

5. Need to explain the significance of *rpn-10*.

Response 39:

We thank the reviewer for the suggestion. In response to this input, we have elaborated on the importance of *rpn-10* in our revised manuscript (lines 540-543). Notably, as RPN-10 plays a crucial role in the ERAD pathway, we have conducted additional experiments and showed the involvement of *rpn-10* in mediating the impact of warmer temperatures and BiP on germline sex determination (Fig. EV8F-I and lines 543-548, 552-558). The results further support our original conclusion regarding the role of ERAD in the warmer temperature-induced masculinization of the germline (Fig.5 in the revised manuscript). Once again, we sincerely appreciate the reviewer's suggestion, as it has greatly contributed to enhancing the overall integrity of our paper.

6. I recommend making *C. remanei* part of Figure 6 rather than in the extended figures. I would also like to know whether the *C. remanei* are self-fertilizing when Mog is induced.

Response 40:

Thank you for your valuable advice. We agree with the reviewer's suggestion. We have relocated the data pertaining to *C. remanei* from its previous location as Figure EV10 to Figure 6 M-S in the revised manuscript, which serves to highlight the crucial role of BiP in regulating germline sex determination not only in hermaphrodites but also in a male/female species.

Despite both sperm and oocytes being produced in females of the *C. remanei* species after *hsp-4(RNAi)* (Fig. 6M-S in the revised manuscript), these nematodes are unable to reproduce

through self-fertilization. One possible explanation for this inability could be due to a defect in sperm activation. Supporting this possibility, previous research found that the successful transformation of females into hermaphrodites requires not only the induction of sperm production but also subsequent sperm activation, which is initially absent in females (Ellis, 2017). To enable the transformation of *C. remanei* females into self-fertilizable hermaphrodites, after *hsp-4(RNAi)* to alter the female germline sex to induce sperm production, another genetic manipulation for activating these sperm may be required. Once again, we would like to express our gratitude to the reviewer for raising this thought-provoking question, as it sheds light on a significant area for further exploration in future research.

References

- Amin-Wetzel N, Saunders RA, Kamphuis MJ, Rato C, Preissler S, Harding HP, Ron D (2017) A J-Protein Co-chaperone Recruits BiP to Monomerize IRE1 and Repress the Unfolded Protein Response. *Cell* 171: 1625-1637 e13
- Bakunts A, Orsi A, Vitale M, Cattaneo A, Lari F, Tade L, Sitia R, Raimondi A, Bachi A, van Anken E (2017) Ratiometric sensing of BiP-client versus BiP levels by the unfolded protein response determines its signaling amplitude. *Elife* 6
- Carrara M, Prischi F, Nowak PR, Kopp MC, Ali MM (2015) Noncanonical binding of BiP ATPase domain to Ire1 and Perk is dissociated by unfolded protein CH1 to initiate ER stress signaling. *Elife* 4
- Ellis RE (2017) "The persistence of memory"-Hermaphroditism in nematodes. *Mol Reprod Dev* 84: 144-157
- Hata T, Ishiwata-Kimata Y, Kimata Y (2022) Induction of the Unfolded Protein Response at High Temperature in *Saccharomyces cerevisiae*. *Int J Mol Sci* 23
- Hirota M, Kitagaki M, Itagaki H, Aiba S (2006) Quantitative measurement of spliced XBP1 mRNA as an indicator of endoplasmic reticulum stress. *J Toxicol Sci* 31: 149-56
- Hubert A, Anderson P (2009) The *C. elegans* sex determination gene *laf-1* encodes a putative DEAD-box RNA helicase. *Dev Biol* 330: 358-67
- Jaffe LF (1995) Calcium waves and development. *Ciba Found Symp* 188: 4-12; discussion 12-7
- Kohno S, Katsu Y, Urushitani H, Ohta Y, Iguchi T, Guillette LJ, Jr. (2010) Potential contributions of heat shock proteins to temperature-dependent sex determination in the American alligator. *Sex Dev* 4: 73-87
- Lai CW, Aronson DE, Snapp EL (2010) BiP availability distinguishes states of homeostasis and stress in the endoplasmic reticulum of living cells. *Mol Biol Cell* 21: 1909-21
- Lee MH, Kim KW, Morgan CT, Morgan DE, Kimble J (2011) Phosphorylation state of a Tob/BTG protein, FOG-3, regulates initiation and maintenance of the *Caenorhabditis elegans* sperm fate program. *Proc Natl Acad Sci U S A* 108: 9125-30
- Li T, Kelly WG (2014) A role for WDR5 in TRA-1/Gli mediated transcriptional control of the sperm/oocyte switch in *C. elegans*. *Nucleic Acids Res* 42: 5567-81
- Liu Y, Chang A (2008) Heat shock response relieves ER stress. *EMBO J* 27: 1049-59

Mapes J, Chen JT, Yu JS, Xue D (2010) Somatic sex determination in *Caenorhabditis elegans* is modulated by SUP-26 repression of *tra-2* translation. *Proc Natl Acad Sci U S A* 107: 18022-7

Noble DC, Aoki ST, Ortiz MA, Kim KW, Verheyden JM, Kimble J (2016) Genomic Analyses of Sperm Fate Regulator Targets Reveal a Common Set of Oogenic mRNAs in *Caenorhabditis elegans*. *Genetics* 202: 221-34

Ramot D, MacInnis BL, Goodman MB (2008) Bidirectional temperature-sensing by a single thermosensory neuron in *C. elegans*. *Nat Neurosci* 11: 908-15

Sarge KD, Cullen KE (1997) Regulation of hsp expression during rodent spermatogenesis. *Cell Mol Life Sci* 53: 191-7

Sasagawa Y, Otani M, Higashitani N, Higashitani A, Sato K, Ogura T, Yamanaka K (2009) *Caenorhabditis elegans* p97 controls germline-specific sex determination by controlling the TRA-1 level in a CUL-2-dependent manner. *J Cell Sci* 122: 3663-72

Sasagawa Y, Yamanaka K, Ogura T (2007) ER E3 ubiquitin ligase HRD-1 and its specific partner chaperone BiP play important roles in ERAD and developmental growth in *Caenorhabditis elegans*. *Genes Cells* 12: 1063-73

Shimada M, Kanematsu K, Tanaka K, Yokosawa H, Kawahara H (2006) Proteasomal ubiquitin receptor RPN-10 controls sex determination in *Caenorhabditis elegans*. *Mol Biol Cell* 17: 5356-71

Sokol SB, Kuwabara PE (2000) Proteolysis in *Caenorhabditis elegans* sex determination: cleavage of TRA-2A by TRA-3. *Genes Dev* 14: 901-6

Spiess M, Junne T, Janoschke M (2019) Membrane Protein Integration and Topogenesis at the ER. *Protein J* 38: 306-316

Starostina NG, Lim JM, Schvarzstein M, Wells L, Spence AM, Kipreos ET (2007) A CUL-2 ubiquitin ligase containing three FEM proteins degrades TRA-1 to regulate *C. elegans* sex determination. *Dev Cell* 13: 127-39

Tang H, Han M (2017) Fatty Acids Regulate Germline Sex Determination through ACS-4-Dependent Myristoylation. *Cell* 169: 457-469 e13

Weber C, Zhou Y, Lee JG, Looger LL, Qian G, Ge C, Capel B (2020) Temperature-dependent sex determination is mediated by pSTAT3 repression of *Kdm6b*. *Science* 368: 303-306

Dear Prof. Tang,

Congratulations on a great revision! Overall, the referees have been positive. However, there remain a few editorial items that we ask you to attend to in a revised manuscript. When you submit your revised version, please also take care of the following editorial items and add this also to your point-by-point response:

1. Please reduce the number of keywords to 5.
2. Please update the reference format. Et al should follow after 10 authors, whereas you have this after 20.
3. Please provide a Data Availability section per our website guidelines.
4. Please rename the conflict of interest section to "Disclosure and Competing Interests Statement"
5. Please remove the author contribution section from the main manuscript.
6. Please provide the appendix file in PDF format.
7. Please provide the Source Data checklist. We've uploaded the blank form to your account, please fill this in and reupload.
8. Source data files need to be saved in a one figure/folder and then uploaded as .zip files. E.g. all the Source data files for figure 1 need to be saved in a single folder and this needs to be zipped and then uploaded as "SD figure 1.zip" file.
9. We include a synopsis of the paper (see <http://emboj.embopress.org/>). Please provide me with a general summary statement and 3-5 bullet points that capture the key findings of the paper.
10. We also need a summary figure for the synopsis. The size should be 550 wide by 200-440 high (pixels). You can also use something from the figures if that is easier.
11. Please note that the exact p values are not provided in the legends of figures 1b, g, i; 2a-b, k; 3c, j; 4b-c; 5a-b, d-e; 6g, j, l; EV 1b-c; EV 2; EV 4a-b; EV 5c, e, l; EV 6c, f, i, l, p; EV 7; EV 8f-i. Please update.
12. There are 9 EV figures, there should be up to 5, with the nomenclature Figure EV1-EV5 and appropriate callouts, and the others should be compiled in Appendix PDF renamed to Appendix Figure S1-S4 with the corresponding callouts in the main manuscript.
13. Section order should be corrected: title page with complete author information, abstract, keywords, introduction, results, discussion, materials & methods, data availability section, acknowledgements, disclosure and competing interests statement, references, main figure legends, tables, expanded figure legends.

Thank you for the opportunity to consider your work for publication. I look forward to your revision.

Kind regards,

Kelly

Kelly M Anderson, PhD
Editor, The EMBO Journal
k.anderson@embojournal.org

Referee #2:

The revised manuscript has been significantly improved. I am satisfied with the changes the authors have made in response to the concerns I raised earlier. The additional experiments and revised text further strengthen the study's conclusions. I support its

publication.

Referee #3:

The *C. elegans* sex determination pathway has been extensively studied and shown to be driven by a genetic cascade (GSD). Shi et al. identify a layer of temperature control on the sex determination pathway suggesting that GSD can be influenced by the environment. Using a large array of approaches, the authors identify *C. elegans* strains that have a Mog (Masculinization of germline) phenotype upon temperature shift. They systematically examine the germ line sex determination pathway and find decreased levels of the oocyte driving TRA-2 at high temperatures. They go on to show that this is driven by the ER chaperone, BiP, and provide evidence that this may be conserved in male/female worms. This is a very interesting study that provides mechanistic insight into how germ line sex determination can be modulated by the environment (temperature) with broad implications for understanding sex determination controlled at both the genetic and environmental levels. The authors have done an excellent job addressing the previous reviews. I remain excited about this study and recommend publishing it. My only comment is that the writing is still difficult to wade through.

Referee #2:

The revised manuscript has been significantly improved. I am satisfied with the changes the authors have made in response to the concerns I raised earlier. The additional experiments and revised text further strengthen the study's conclusions. I support its publication.

Response: We appreciate the reviewer's positive feedback regarding our revision.

Referee #3:

The *C. elegans* sex determination pathway has been extensively studied and shown to be driven by a genetic cascade (GSD). Shi et al. identify a layer of temperature control on the sex determination pathway suggesting that GSD can be influenced by the environment. Using a large array of approaches, the authors identify *C. elegans* strains that have a Mog (Masculinization of germline) phenotype upon temperature shift. They systematically examine the germ line sex determination pathway and find decreased levels of the oocyte driving TRA-2 at high temperatures. They go on to show that this is driven by the ER chaperone, BiP, and provide evidence that this may be conserved in male/female worms. This is a very interesting study that provides mechanistic insight into how germ line sex determination can be modulated by the environment (temperature) with broad implications for understanding sex determination controlled at both the genetic and environmental levels. The authors have done an excellent job addressing the previous reviews. I remain excited about this study and recommend publishing it. My only comment is that the writing is still difficult to wade through.

Response:

We thank the reviewer for the positive feedback on our revision. In response to the comment on the writing, we would like to mention that prior to our initial submission, the paper had already undergone professional editing through the Springer Nature author service. We highly value the reviewer's suggestion and have implemented the necessary modifications to enhance the readability of our manuscript during the revision process.

Dear Dr. Tang,

I am pleased to inform you that your manuscript has been accepted for publication in the EMBO Journal.

Your manuscript will be processed for publication by EMBO Press. Some sections will copy edited and you will receive page proofs prior to publication. Please note that you will be contacted by Springer Nature Author Services to complete licensing and payment information.

It has been a pleasure to work with you to get this to the acceptance stage, I will be in touch throughout the editorial process. In the meantime, I hope you find time to celebrate!

Yours sincerely,

Kelly M Anderson, PhD
Editor, The EMBO Journal
k.anderson@embojournal.org

*